# Membrane remodeling by FAM92A1 during brain development regulates neuronal morphology, synaptic function, and cognition

Liang Wang[1,2,9], Ziyun Yang[1,9], Fudo Satoshi[3], Xavier Prasanna[4], Ziyi Yan[2], Helena Vihinen[5], Yaxing Chen[1], Yue Zhao[1], Xiumei He[1,6,7], Qian Bu[1], Hongchun Li[1], Ying Zhao[1], Linhong Jiang[1], Feng Qin[1], Yanping Dai[1], Ni Zhang[8], Meng Qin[1], Weihong Kuang[8], Yinglan Zhao[1], Eija Jokitalo[5], Ilpo Vattulainen[4], Tommi Kajander[3], Hongxia Zhao[2,6,7] ✉ & Xiaobo Cen[1] ✉

The Bin/Amphiphysin/Rvs (BAR) domain protein FAM92A1 is a multifunctional protein engaged in regulating mitochondrial ultrastructure and ciliogenesis, but its physiological role in the brain remains unclear. Here, we show that FAM92A1 is expressed in neurons starting from embryonic development. FAM92A1 knockout in mice results in altered brain morphology and age-associated cognitive deficits, potentially due to neuronal degeneration and disrupted synaptic plasticity. Specifically, FAM92A1 deficiency impairs diverse neuronal membrane morphology, including the mitochondrial inner membrane, myelin sheath, and synapses, indicating its roles in membrane remodeling and maintenance. By determining the crystal structure of the FAM92A1 BAR domain, combined with atomistic molecular dynamics simulations, we uncover that FAM92A1 interacts with phosphoinositide- and cardiolipin-containing membranes to induce lipid-clustering and membrane curvature. Altogether, these findings reveal the physiological role of FAM92A1 in the brain, highlighting its impact on synaptic plasticity and neural function through the regulation of membrane remodeling and endocytic processes.

Bin/Amphiphysin/Rvs (BAR) domain protein families are known for their roles in membrane remodeling in an extraordinary diversity of cellular processes, including fission of synaptic vesicles, endocytosis, secretory vesicle fusion, and regulation of the actin cytoskeleton[1–4]. FAM92A1, also named CIBAR1, is a BAR domain protein that functions in important processes involving membrane remodeling, such as mitochondrial crista ultrastructure and ciliogenesis[5–7]. FAM92A1 preferentially interacts with the negatively charged phospholipids PI(4,5)$P_2$ and cardiolipin to induce extensive membrane curvature[7]. This membrane-remodeling role of FAM92A1 in human osteosarcoma cells is indispensable for the formation of crista, mitochondrial dynamics, and oxidative phosphorylation[7].

FAM92A1 is highly conserved in vertebrates and is widely expressed during embryonic development and in tumors[8–10]. Interestingly, studies have revealed that deletions in the chromosomal region 8q21.3–8q22.1, resulting in the FAM92A1 depletion, are associated with a range of developmental issues, including global developmental delay, autism, and microcephaly[11].

Furthermore, a nonsense variant (c.478 C > T, p.[Arg160*]) in the *FAM92A* gene, located within the mapped region 8q21.13–q24.12, has been identified in a family with autosomal recessive polydactyly[8]. These observations collectively imply that FAM92A1 may have significant roles in maintaining normal brain structure and function, with potential implications for various developmental and neurological conditions.

The brain is composed of distinct functional cell types, each of which is characterized by a complex array of structures primarily originating from their plasma membranes. Notably, cells within the brain, such as neurons and oligodendrocytes, exhibit clear membrane polarization, which is delineated by distinct morphological and functional subcompartments. The establishment, reinforcement, and proper functioning of these subcompartments necessitate a remarkable level of lateral organization within the membrane[12,13]. Integral to these processes are membrane-remodeling proteins, which play an indispensable role in regulating and preserving specialized structures and functions within synapses[14–17]. Within the context of synapses, BAR domain proteins have emerged as pivotal regulators of endocytic recycling processes[18,19]. This intricate regulation involves sequential recruitments of several BAR domain proteins, each performing distinct roles in inducing membrane curvature during synaptic membrane recycling. These include the F-BAR domain proteins FCHo1, FCHo2, and syndapin II, the N-BAR proteins endophilin A2, BIN1, and BIN2, the PX-BAR protein SNX9, and the BAR-PH protein APPL1[19–22]. Their orchestrated efforts culminate in precise membrane curvature, thereby facilitating the recycling of synaptic membranes. Presently, it remains completely unknown whether FAM92A1 participates in these processes and what specific functions it fulfills within the brain. Additionally, uncertainties persist regarding whether the various neurological disorders triggered by FAM92A1 deletion or mutation in humans are linked to its membrane remodeling activity. Furthermore, FAM92A1 is predicted to feature a BAR domain at its N-terminus and an unstructured short tail at its C-terminus. Despite the AI-based AlphaFold2 (AF2) tool predicting the structure of FAM92A1 BAR domain[23], its structural characteristics based on the experimental data have not been elucidated.

In this study, we conducted experiments using homozygous FAM92A1$^{-/-}$ and heterozygous FAM92A1$^{+/-}$ mice to investigate the physiological role of FAM92A1 within neurons. Our data revealed a prominent enrichment of FAM92A1 in the brain, particularly at synapses in addition to its presence within mitochondria. Depletion of FAM92A1 in mice resulted in detrimental impairments across multiple facets of neuronal morphology and function. Specifically, FAM92A1 knockout mice displayed alterations in brain and neuron morphology. The absence of FAM92A1 impaired neuronal complexity, compromised synaptic transmission, and disrupted synaptic plasticity. Notably, the deficiency of FAM92A1 led to the emergence of abnormal membrane architectures and disturbances in neuronal ultrastructure. This encompassed multiple components such as mitochondrial cristae, synapses, and axonal myelin sheaths. These perturbations in membrane architecture had far-reaching consequences, influencing brain regions such as the entorhinal and hippocampal areas. Additionally, these structural deficiencies were accompanied by deficits in multiple cognition-associated tasks, highlighting the importance of FAM92A1 in maintaining proper cognitive functions. By delving deeper into the molecular mechanisms, we determined the crystal structure of the FAM92A1 BAR domain and employed molecular simulation to elucidate its precise membrane interaction dynamics. In summary, by dissecting its impact on neuronal architecture and function, we revealed the underlying mechanism of specific membrane-remodeling activity of FAM92A1 in neurons and the potential risk associated with FAM92A1 mutation in the development of neurological disorders in humans.

## Results
### FAM92A1 is highly expressed in the brain and FAM92A1 knockout results in alterations in brain morphology

Previous studies have demonstrated that FAM92A1 localizes to the mitochondrial inner membrane and the mother centrioles/basal bodies of cilia[5–7]. However, the expression pattern and physiological function of FAM92A1 within the brain remain to be defined. Herein, we first analyzed the expression profile of FAM92A1 in the mouse brain using *Fam92a1* in situ hybridization (ISH) data of the adult mouse brain from the Allen Brain Atlas (ABA) database (https://mouse.brainmap.org/experiment/show/587419). Analysis of the ISH data revealed robust expression of FAM92A1 across various brain regions, including the isocortex, hippocampus (HPC), and olfactory areas (OLF) (Fig. 1a). To substantiate these findings, we conducted immunofluorescence (Fig. 1b) and western blotting assays (Fig. 1c) utilizing wild-type mice. Both approaches consistently confirmed the presence of FAM92A1 in multiple brain regions, such as the HPC and cerebral cortex (CC) (Supplementary Fig. 1a). Moreover, FAM92A1 was expressed in the hippocampus during embryonic development and maintained a consistent level in the early postnatal period (Fig. 1d). By detecting FAM92A1 expression in cultured hippocampal neurons, we observed that, like the mitochondrial protein SDHA, FAM92A1 exhibited constant expression levels during the early stages of neuronal differentiation and maintained a gradual increase throughout the maturation process, akin to the synaptic protein synaptophysin (Fig. 1e and Supplementary Fig. 1b). Collectively, these findings reveal that FAM92A1 is prominently expressed from the early stages of brain development. This conspicuous expression pattern points to a potentially role for FAM92A1 in the developmental processes of the central nervous system (CNS).

To explore the physiological role of FAM92A1 in the brain, we engineered a transgenic knockout mouse strain using the CRISPR/Cas9 system with two single guide RNAs (sgRNA). These sgRNAs targeted the upstream region of exon 1 and the downstream region of exon 6 in the *Fam92a1* gene, respectively, resulting in effectively eliminating the first six exons of *Fam92a1* gene (Supplementary Fig. 1c). The genotype of each mouse was identified by PCR with two pairs of primers, one pair flanking the deleted region (P1, F1 + R1) and the other pair targeting sequences inside the deleted region (P2, F2 + R2) (Fig. 1f and Supplementary Fig. 1c). Subsequent RT-qPCR affirmed the successful knockout of FAM92A1 at the transcriptional level in various tissues (Fig. 1g and Supplementary Fig. 1d). By aligning the immunogen sequences of two commercial anti-FAM92A1 antibodies, we noticed that FAM92A1 knockout mice preserved the immunogen sequences of one antibody from Sigma (HPA034760) (FAM92A1 181–259aa encoded by genes spanning exon 7 to exon 8) while depleting most of immunogen sequences of the antibody from Proteintech (24803-1-AP) (FAM92A1 115–259aa encoded by genes spanning exon 4 to exon 8) (Supplementary Fig. 1e). Hence, we employed the FAM92A1 antibody from Proteintech to evaluate the efficiency of FAM92A1 knockout at the translation level. In line with the transcriptional findings, the protein expression of FAM92A1 was markedly diminished and absent in heterozygous and homozygous FAM92A1 knockout mice, respectively (Fig. 1g and Supplementary Fig. 1f). Immunofluorescence with the FAM92A1 antibody (HPA034760) further confirmed the evident depletion of endogenous FAM92A1 both in brain slices and primary fibroblasts from homozygous FAM92A1 knockout mice (Supplementary Fig. 1g, h). However, the FAM92A1 antibody (24803-1-AP) showed unspecific staining due to the presence of multiple non-specific bands (Supplementary Fig. 1g-i). Taken together, the homozygous FAM92A1 knockout mice (FAM92A1$^{-/-}$) displayed a complete absence of the FAM92A1 protein in contrast to the level of FAM92A1 in wild-type mice (FAM92A1$^{+/+}$).

During the process of mouse breeding, we crossed adult homozygous FAM92A1 male mice with partners that were either wild-type,

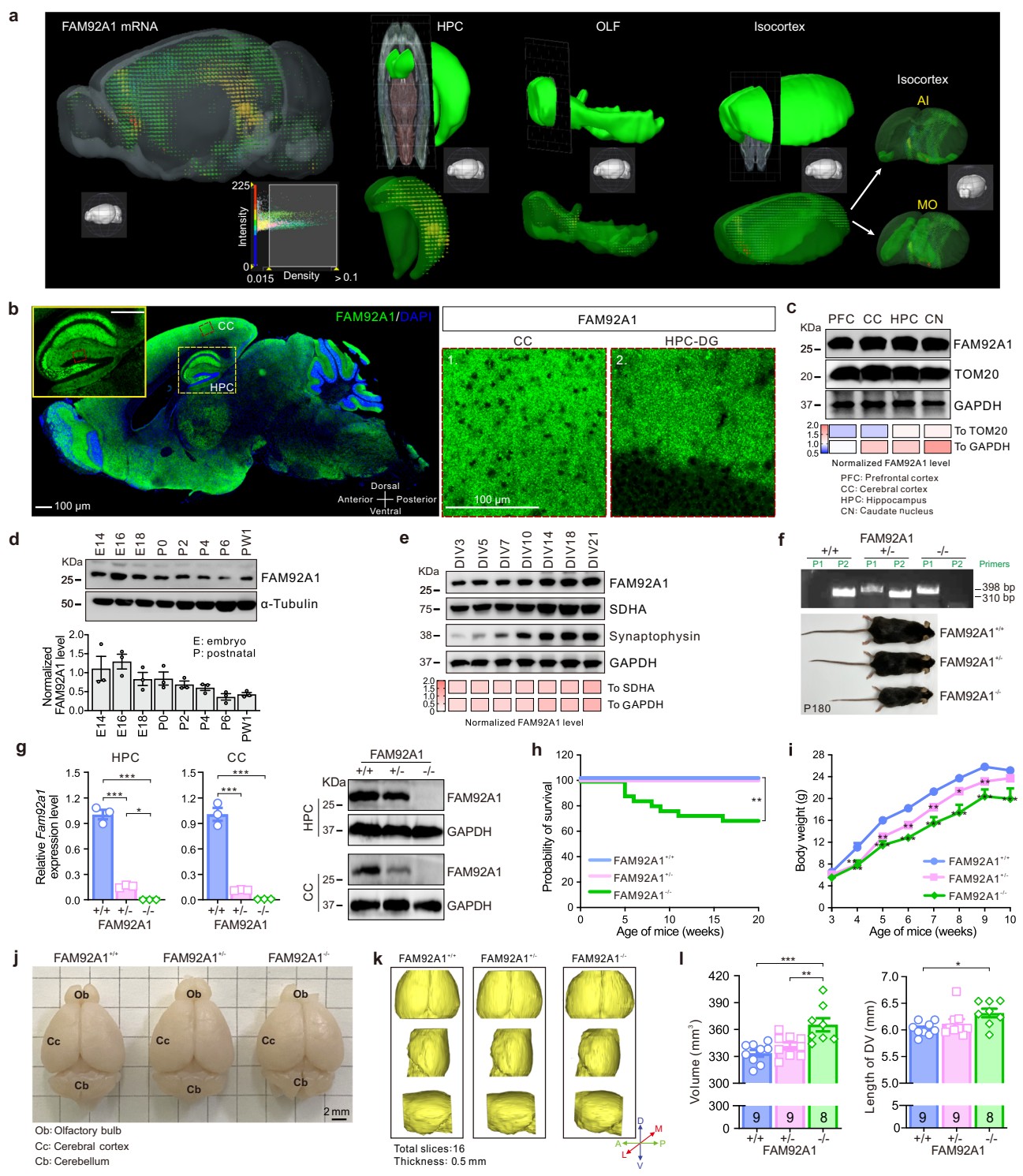

heterozygous, or homozygous for FAM92A1. Unfortunately, none of these crosses yields viable embryos. Conversely, when adult homozygous FAM92A1 female mice were mated with wild-type or heterozygous partners, viable pregnancies were established, leading to the birth of embryos. These results indicated that the FAM92A1 knockout resulted in male infertility. To understand the role of FAM92A1 in embryonic development and postnatal growth, we closely examined the progeny resulting from intercrosses of heterozygous mice. By counting the number of each genotype in 25 litters, the frequencies of three genotypes in each litter were calculated. Compared to the theoretical probabilities of FAM92A1+/+ ($q^2 = 0.25$), FAM92A1+/− ($2pq = 0.5$),

and FAM92A1−/− ($p^2 = 0.25$), the observed frequencies of FAM92A1+/+, FAM92A1+/−, and FAM92A1−/− genotypes were 0.33, 0.64, and 0.08, respectively (Supplementary Fig. 1j, k). Some litters even exhibited no viable FAM92A1−/− offspring (Supplementary Fig. 1l). This observation suggests a departure from the expected Mendelian inheritance pattern for the birth of FAM92A1 homozygotes. Distinctly, a proportion of homozygous pups (~30%) experienced premature mortality between 5 and 15 weeks of age (8 out of 26 FAM92A1−/− mice). In contrast, no lethality was observed among wild-type or heterozygous mice (FAM92A1+/−) during the postnatal period (Fig. 1h). In addition to the observed difference in survival rate, FAM92A1−/− mice displayed slower

**Fig. 1 | Loss of FAM92A1 results in alterations in brain morphology. a** ISH images analyzed from the ABA database showing *Fam92a1* expression in mouse brain. The colored spheres represent the amount of expression, with color bar ranging from blue to green indicating low to high expression. AI, agranular insular area; MO, somatomotor area. **b**–**e** FAM92A1 expression in sagittal sections of the mouse brain (**b**), various mouse brain regions (**c**), mouse hippocampus at different stages from embryonic development to postnatal days (**d**), primary hippocampal neurons on different culture days (**e**). Data are representative images and Western blots from three independent experiments. Heatmap (**c**, **e**) shows the corresponding normalized FAM92A1 levels. Bar graph (**d**) represents mean ± SEM. Scale bar, 100 μm. **f** Representative agarose gel electrophoresis showing PCR-based genotyping of mice (top) using two primer pairs, along with representative images of each mouse genotype at 6 months of age (bottom), derived from at least three independent experiments. **g** RT-qPCR (left) and western blot (right) analysis of FAM92A1 expression within the HPC and CC areas. Data present mean ± SEM of three technical replicates over one independent experiment (one-way ANOVA). **h** Kaplan–Meier plot showing survival of mice (log-rank test). **i** Growth curves of mouse body weight from FAM92A1[+/+] ($n = 5$ mice), FAM92A1[+/−] ($n = 6$ mice), and FAM92A1[+/−] ($n = 5$ and 3 mice for the before and after 8 weeks, respectively) groups. Data represent mean ± SEM; two-way ANOVA. **j**, **k** Representative macroscopic (**j**) and 3D reconstructions (**k**) of mouse brains from each genotype. Sixteen continuous slices with a thickness of 0.5 mm were used for 3D reconstruction with Mimics software (**k**). Scale bar, 2 mm. AP, anteroposterior axis; DV, dorsoventral axis; ML, mediolateral axis. **l** Quantification of brain volume and DV length for the reconstructed mouse brains of FAM92A1[+/+] ($n = 9$), FAM92A1[+/−] ($n = 9$), and FAM92A1[−/−] ($n = 8$). Data present mean ± SEM; one-way ANOVA. *$p < 0.05$, **$p < 0.01$, and ***$p < 0.001$. Source data and exact *p*-values are provided as a Source Data file.

growth rates and smaller body sizes compared to their wild-type littermates (Fig. 1f, i). These findings collectively indicate the essential role of FAM92A1 in embryonic development, growth, and overall viability.

To explore the effects of FAM92A1 knockout on the brain, we first macroscopically examined the adult mouse brains (20 weeks of age). Notably, a majority of FAM92A1[−/−] mice brains exhibited a notable swelling, particularly evident from the transverse sinus to the bregma area (Fig. 1j). Subsequently, we applied T2-weighted magnetic resonance imaging (MRI) and 3D reconstruction to examine the brains of mice using 16 fixed continuous slices (with a thickness of 0.5 mm) spanning from the direction of the transverse sinus to the bregma. This analysis revealed a significant increase in the brain volume upon FAM92A1 loss (FAM92A1[+/+], 334 ± 4 mm³; FAM92A1[+/−], 342 ± 4 mm³; FAM92A1[−/−], 365 ± 7 mm³) (Fig. 1k, l). Furthermore, the length of the dorsoventral (DV) axis also exhibited an increase in the FAM92A1[−/−] mice (Fig. 1l). Taken together, these data indicate that the deletion of FAM92A1 led to distinct alterations in brain morphology, specifically manifesting as swelling around the transverse sinus area. This finding underscores the intricate role of FAM92A1 in maintaining the normal morphology of the brain.

## FAM92A1 depletion leads to abnormal brain structure and degeneration of hippocampal neurons

The pathology of brain swelling is multifaceted and can be associated with various conditions, such as brain injury, severe neuroinflammation, brain tissue edema, or age-related changes in the brain[24–27]. To investigate the cause of brain anomalies, the neuroanatomical features of mice were examined. By quantifying the ventricular area using T2-weighted MRI of the brains, we revealed a considerable increase in total ventricular area in FAM92A1[−/−] mice compared to their wild-type littermates (Fig. 2a, b and Supplementary Fig. 2a, b). At the regional level, ventricular dilatation in FAM92A1 homozygotes was primarily attributed to enlargement of the lateral ventricles (LVs), with no evident difference observed in the third, cerebral, and fourth ventricles (Supplementary Fig. 2c, d). Additionally, a deformation-based morphometry (DBM) analysis was conducted to examine the volumetric differences between wild-type and FAM92A1[+/−] mouse brains using MRI scans from perfused brains. As expected shrinkage of brain regions surrounding the enlarged LVs, FAM92A1[+/−] mice exhibited a diminished volume around the central nucleus of the amygdala (CeA) and entorhinal areas compared to wild-type mouse brains. In contrast, there was an increase in volume around the caudoputamen (CPu) and globus pallidus (GPe) (Supplementary Fig. 2e). These findings suggest that ventricular dilatation, along with alterations in the volumetric morphology of brain areas, may contribute to brain swelling in FAM92A1-depleted mice.

To explore the effect of FAM92A1 deletion on the volumetric differences of the mouse brain, we applied whole-brain voxel-based morphometry (VBM) analysis to detect morphological alterations in the gray matter. The results revealed that the regional gray matter volume (rGMV) in two regions outside of the LVs was lower in FAM92A1-deficient mice compared to FAM92A1[+/+] mice (Fig. 2c and Supplementary Fig. 2e). Quantifying the gray matter density (GMD) in both areas, area 1 (encompassing multilayers of entorhinal area, medial part, dorsal zone-ENTm) and area 2 (encompassing multilayers of the retrosplenial area-RSP, corpus callosum, and cingulum bundle), revealed a notable decrease in FAM92A1[−/−] mice compared to FAM92A1[+/+] mice (Fig. 2d, e). Despite the differences between FAM92A1[+/+] and FAM92A1[+/−] mice being less pronounced, a tendency toward decreased GMD was evident in both areas. Contrary to the expected rise in gray matter density within the diminished brain area, the reduced gray matter in these regions of FAM92A1-deficient mice highlights the intricate role of FAM92A1 in preserving the structural integrity of the brain.

The potential correlates of gray matter density may include factors such as the number and size of neurons, as well as the level of synaptic density[28,29]. To investigate the potential causes behind the diminished gray matter volume observed in FAM92A1-deficient mice, we performed a histological analysis of neurons using cresyl violet (CV) staining. Based on the Franklin and Paxinos mouse brain atlas, we selected five reference planes ranging from anterior (a) to posterior (e) for comparative analyses (Fig. 2f). In line with the macroscopic brain morphology, the width of the planes along the mediolateral (ML) direction was smaller in the anterior area while larger in the posterior area in FAM92A1[−/−] brains compared to wild-type brains. Although no discernible differences were noted in the distribution pattern of CV-stained cells, many vacuoles were observed in the hippocampal dentate gyrus (DG) region, particularly in the subgranular zone (SGZ), of the FAM92A1-deficient mouse brain (Fig. 2g). This vacuolation was linked to the reduced thickness of the DG region (Fig. 2h), indicating neuronal damage of loss in this area. Moreover, given the intricate intrinsic connectivity between the entorhinal cortex and the hippocampus across species[30], we also conducted a VBM analysis of the hippocampal region. In contrast, the difference in the changed ratio of gray matter density was aggravated with the loss of FAM92A1, primarily manifesting as decreased GMD in hippocampal subregions (Fig. 2i). Furthermore, histofluorescence staining using Fluoro-Jade B (FJB) or Fluoro-Jade C (FJC), commonly employed for the targeted detection of degenerating mature neurons, revealed an elevated presence of FJB- and FJC-positive cells in the hippocampus following FAM92A1 depletion (Fig. 2j and Supplementary Fig. 2f, g). Taken together, these data suggest that the enlargement of brain gross morphology in the FAM92A1 knockout mice may be associated with the enlargement of the lateral ventricles, neuronal loss, and degeneration.

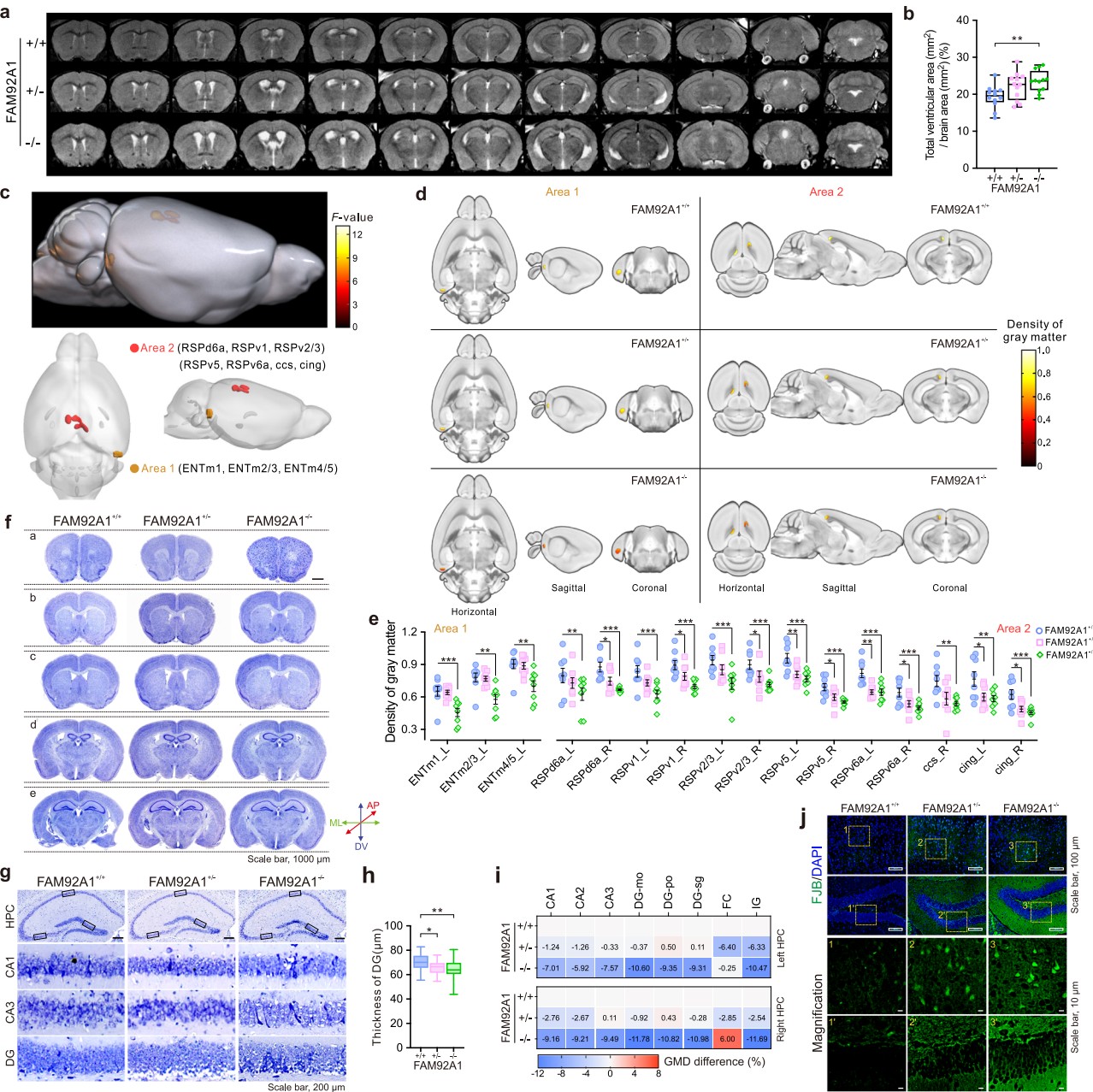

**Fig. 2 | FAM92A1 depletion leads to abnormal brain structure and degeneration of hippocampal neurons. a**, **b** Representative MRIs of brains (**a**) and quantification of total ventricular areas (**b**) for each FAM92A1 genotype. $n = 12$ mice per group; one-way ANOVA. **c** Visualization of the anatomical localization of significant changed rGMV among wild-type and FAM92A1-deficient groups. The red and orange colors represent two significant changed areas. Color bar represents the $F$-value of one-way ANOVA; $n = 8$ mice per group. ENTm, medial entorhinal area; RSPd, dorsal retrosplenial area; RSPv, ventral retrosplenial area; cc, corpus callosum; cing, cingulum bundle. **d** Visualization of GMD in horizontal, sagittal, and coronal planes, respectively. Color bar represents GMD. **e** Scatter plots showing the GMD of each group. Data present as mean ± SEM; $n = 8$ mice per group; one-way ANOVA. **f** Representative images from four mice per group showing CV staining of five selected coronal planes (**a**–**e**) along the rostro-caudal axis of adult mouse brains, according to the Franklin and Paxinos's Mouse Brain atlas. **a** anterior section

without the corpus callosum; (**b**) anterior section including the corpus callosum; (**c**) medial section with a visible anterior commissure; (**d**) medial section including the just appeared hippocampus; (**e**) medial section including the clear DG area. Scale bar, 1000 μm. **g** Magnified CV staining of hippocampus and its subregions (CA1, CA3, and DG) from (**f**). Scale bar, 200 μm. **h** Box-and-whisker plot showing the thickness of the DG. $n = 28$ measurements from four mice per group; one-way ANOVA. For (**b**, **h**) data are presented as minimum to maximum values (whiskers) and the 25th to 75th percentiles (box), with the mean value within the box. **i** Heatmap showing the average change ratio of GMD within hippocampal subregions relative to the FAM92A1$^{+/+}$ group. $n = 8$ mice per group, two-way ANOVA. **j** Representative images from three mice per group showing FJB staining of the hippocampus. Scale bar, 100 μm (top two panels) and 10 μm (magnification). *$p < 0.05$, **$p < 0.01$, and ***$p < 0.001$. Source data and exact $p$-values are provided as a Source Data file.

## FAM92A1$^{-/-}$ mice exhibit age-associated memory decline and cognitive deficits

The entorhinal-hippocampal circuitry, which evolves across species, underpins both spatial and non-spatial memory functions, ultimately culminating in the intricate semantic and episodic memory observed

in humans[31]. Thus, to assess the impact of the abnormalities in the entorhinal area and hippocampus on learning and memory, both young (1–2 months) and adult (5–6 months) FAM92A1 knockout mice, as well as mice with neuronal FAM92A1 knockdown in the hippocampal region, were subjected to a series of hippocampal-dependent tasks

(Fig. 3a). After identifying the silencing efficiency of FAM92A1 siRNA in HT22 cells (Supplementary Fig. 3a, b), two sequences (FAM92A1 siRNA #1 and #4) targeting mouse FAM92A1 were designed as shRNA and individually cloned into a neurotropic AAV vector (pAAV-hSyn-EGFP-3xFlag), generating pAAV-hSyn-EGFP-3xFlag-miR30shRNA (*Fam92a1*) (abbreviated as sh*Fam92a1* #1 and sh*Fam92a1* #4). The virus was bilaterally injected into the hippocampus. After 3 weeks, the robust expressed EGFP in the hippocampus demonstrated the precise viral injection and expression (Fig. 3b). Additionally, the protein level of FAM92A1 in the viral injection site was assessed by immunoblotting. Compared to the control shRNA, the FAM92A1 protein level was reduced by both FAM92A1-targeted shRNAs, indicating the successful knockdown of FAM92A1 in the hippocampus (Fig. 3c, d).

Next, we employed the Morris water maze (MWM) test to evaluate spatial learning and memory in mice (Fig. 3e)[32]. Throughout the training phase, the escape latency was similar across the wild-type and FAM92A1-deficient mice, suggesting that the absence of FAM92A1 did not influence the learning performance of mice at both young and adult ages (Fig. 3f, g). However, adult FAM92A1-deficient mice (FAM92A1$^{+/-}$ and FAM92A1$^{-/-}$) displayed diminished accuracy and shorter duration in the target quadrant (SW) during the probe trial of the MWM test. Moreover, FAM92A1$^{-/-}$ mice spent a longer period of time in the opposite quadrant (NE) compared to their wild-type littermates (Fig. 3h, j). Similar to FAM92A1 knockout mice, FAM92A1 knockdown in hippocampal neurons also resulted in spatial memory errors at probe trial. Mice with FAM92A1 knockdown also spent more time in the opposite quadrant rather than the target quadrant (Fig. 3i, k). The reduced time spent by FAM92A1-deficient mice in the target quadrant did not appear to be attributed to decreased swimming speed (Supplementary Fig. 4a, b) or locomotor impairments, as indicated by the open field test (Supplementary Fig. 4c-f). Furthermore, we assessed short-term spatial working memory through spontaneous behavior monitoring using the T-maze test (Supplementary Fig. 4g). Although no significant differences were noted in the total number of entries or choice latency among the three groups, adult FAM92A1$^{-/-}$ mice exhibited a trend of decreasing the total number of altered entries compared to wild-type mice (Supplementary Fig. 4h). Similar subtle deficits in short-term spatial working memory were also observed in the FAM92A1 knockdown mice (Supplementary Fig. 4i). Together, these findings provide evidence that the absence of FAM92A1 contributes to spatial memory deficits, particularly in adulthood.

To assess hippocampus-dependent contextual fear memory and hippocampus-independent cued fear memory, FAM92A1-deficient mice and their wild-type littermates were subjected to a contextual and cued fear conditioning (FC) test. During this test, mice were conditioned with three repetitions of a tone (the cue stimulus) followed by a foot shock in the FC chamber (Fig. 3l). Interestingly, adult FAM92A1$^{-/-}$ mice displayed a failure to recall the conditioning context when compared to wild-type littermates (Fig. 3m). This was evident from the decreased proportion of freezing time observed in FAM92A1$^{-/-}$ mice. In contrast, FAM92A1 depletion did not seem to impact the memory of cued fear conditioning (Fig. 3m). Similar to FAM92A1 knockout mice, FAM92A1 knockdown caused deficits in contextual memory rather than cued memory (Fig. 3n). These results collectively suggest that deficiency in FAM92A1 compromises the recall of contextual fear memory. Moreover, the novel object recognition (NOR) test was employed to assess recognition memory (Supplementary Fig. 4j). Both FAM92A1$^{+/-}$ and FAM92A1$^{-/-}$ mice, as well as FAM92A1 knockdown mice, exhibited reduced exploration time for the new object compared to their respective wild-type littermates and control mice. Although a negative discrimination index was observed across all groups, there was a slight decline in object recognition memory upon FAM92A1 loss (Supplementary Fig. 4k, l). Taken together, these data provide evidence linking FAM92A1 depletion to an age-related decline

in memory functions, indicating the vital role of FAM92A1 in preserving cognitive capabilities.

Next, we further explored whether the observed memory deficit resulting from the reduction of FAM92A1 implies an increased risk of neurological disorder. To address this question, we first examined the expression of FAM92A1 in aged mice (10 months old) and the APP/PS1 transgenic mouse model for early-onset Alzheimer's disease (AD). Intriguingly, both aged mice and AD mice exhibited decreased levels of FAM92A1 in the brain (Fig. 3o–r). Moreover, we performed a genome-wide RNA-seq analysis to identify genes with differential expression upon FAM92A1 depletion. In contrast to FAM92A1$^{+/+}$ mice, the expression of 472 genes (168 downregulated and 304 upregulated genes) was significantly changed in FAM92A1$^{+/-}$ mice, while 1,125 genes (629 downregulated and 496 upregulated genes) showed profound alteration in the hippocampus of FAM92A1$^{-/-}$ mice (Fig. 3s and Supplementary Fig. 3c). Subsequently, an ingenuity pathway analysis (IPA) was utilized to explore the implications of these differentially expressed genes (DEGs) in various diseases and disorders. The resulting bubble chart highlighted the enrichment of numerous neurological disorders, including progressive encephalopathy, Alzheimer's disease, and dementia (Fig. 3t). This intriguing observation underscores a potential link between decreased FAM92A1 expression and the onset of various neurological conditions. Taken together, FAM92A1 deficiency resulted in age-associated memory decline and cognitive deficits, suggesting a possible connection between reduced FAM92A1 expression and the development of neurological disorders.

## FAM92A1 depletion leads to neuronal morphological abnormalities

To gain insight into the precise mechanisms underlying the contribution of FAM92A1 deficiency to neurological dysfunction, we investigated whether FAM92A1 depletion affected neuronal complexity by tracing the dendrites of granule neurons in the hippocampal DG area, visualized through Golgi-staining. The extent of dendritic arborization of each neuron was quantified by Sholl analysis. The results unveiled a significant reduction in dendritic branching complexity in neurons lacking FAM92A1, as observed in both FAM92A1$^{+/-}$ and FAM92A1$^{-/-}$ mice, compared to neurons from the wild-type group (Fig. 4a, b). Moreover, the average total dendritic length of individual neurons decreased from 773 μm in the wild-type neurons to around 450 μm in FAM92A1-deficient neurons (Fig. 4c). These findings suggest a role for FAM92A1 in orchestrating the complexity of dendritic arbors within hippocampal neurons.

Dendritic spines are small protrusions found on neuronal dendrites, playing a pivotal role in synaptic function and inter-neuronal communication. To probe into the contribution of FAM92A1 to dendritic spine formation, maintenance, and structural plasticity, we examined the effects of FAM92A1 knockdown on spine density and morphology in both hippocampus (in vivo) and cultured primary hippocampal neurons (in vitro). The fluorescent images of EGFP-labeled hippocampal neurons showed that the depletion of FAM92A1 in neurons resulted in a substantial loss of spines and reduced spine density (Fig. 4d, e). Given the diversity in dendritic spine shapes and their distinct functional properties, we classified the spines into four categories (stubby, mushroom, long thin, and filopodia/dendrites) using the "Imaris Spines Classifier" extension (Fig. 4f). The analysis results revealed that the loss of FAM92A1 led to pronounced changes in both spine number and morphology (Fig. 4g, h). Across three types of dendritic spines (excluding filopodia/dendrite-like spines), FAM92A1-deficient neurons consistently displayed reduced numbers of spines compared to wild-type neurons (Fig. 4h). Notably, some mushroom-like spines in FAM92A1 knockout neurons exhibited a slight enlargement of head volume in contrast to those in the wild-type neurons (FAM92A1$^{+/+}$: 0.04 μm$^3$; FAM92A1$^{-/-}$: 0.06 μm$^3$) (Fig. 4i). This suggests that FAM92A1 knockout may influence the structural

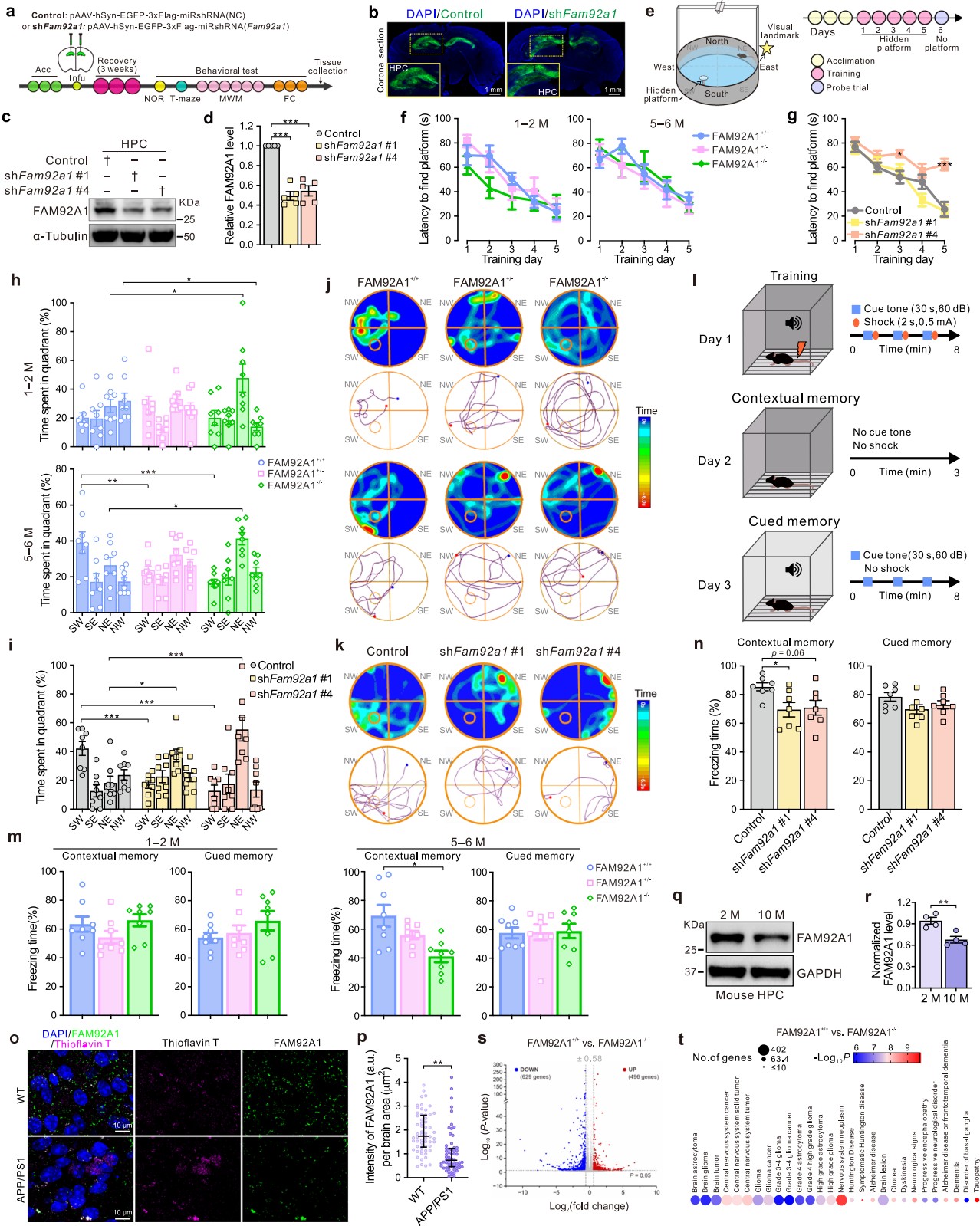

plasticity of dendritic spines. Our in vitro experiments with FAM92A1-silenced hippocampal neurons corroborated these in vivo findings, where the depletion of FAM92A1 by shRNA resulted in reduced spine density and a pronounced loss of mushroom spines (Fig. 4j). Together, these findings indicate that FAM92A1 is involved in the development, maintenance, or stability of dendritic spines, particularly those associated with mature and stable synaptic connections.

## FAM92A1 depletion perturbs synaptic function

Having revealed the altered neuronal morphology following the depletion of FAM92A1, we next sought to monitor its effect on neuronal function. We began by using fiber photometry to record the in vivo physiological calcium ($Ca^{2+}$) signals from the hippocampal DG area in freely moving mice. Monitoring $Ca^{2+}$ signals, as an indicator of neuronal activity, was chosen because heightened intracellular

**Fig. 3 | FAM92A1 deficient mice present age-associated memory decline and cognitive deficits. a** Experimental procedure for behavioral tests with neuron-specific FAM92A1 knockdown mice. **b** Representative images showing EGFP expression in mouse HPC. Scale bar, 1 mm. **c, d** Western blot (**c**) and quantification (**d**) of FAM92A1 expression in mouse HPC. Data represent mean ± SEM; $n = 5$ mice per group; one-way ANOVA. **e** Experimental diagram for the MWM test. **f, g** Latency to find the platform during 5 days of training for FAM92A1 knockout (**f**) and knockdown mice (**g**). **h, i** Percentage of time spent in the four quadrants during the probe trial by FAM92A1 knockout mice (**h**) and knockdown mice (**i**). For (**f–i**) data represent mean ± SEM; $n = 8$ mice per group; two-way ANOVA. **j, k** Representative heatmaps and swimming trajectories of FAM92A1 knockout (**j**) and knockdown mice (**k**) during the probe trial. **l** Experimental diagram for fear conditioning test. **m, n** Percentage of freezing time in FAM92A1 knockout mice (**m**, $n = 8$ mice per group) and knockdown mice (**n**, $n = 7$ mice per group) during contextual and cued

memory tests. Data represent mean ± SEM; one-way ANOVA. **o, p** Representative images (**o**) and quantification (**p**) of FAM92A1 expression of two wild-type ($n = 60$ measurements) and APP/PS1 ($n = 62$ measurements) mice per group. Scale bar, 10 μm. **q, r** Representative western blot (**q**) and quantification (**r**) of FAM92A1 expression in the HPC of four young (2 month-old) and old (10 month-old) mice per group. For (**p**) and (**r**) data represent mean ± SEM; unpaired two-tailed Student's *t*-test. **s** Volcano plot of DEGs (fold change > 1.5 and $p < 0.05$) from RNA-seq analysis upon loss of FAM92A1. **t** Bubble chart showing the enriched human diseases and disorders in IPA analysis using the DEGs in panel (**s**). Statistical tests for (**s**) and (**t**) were based on a two-sided *t*-test with *P*-value adjustment for multiple comparisons (Benjamini–Hochberg method) by DESeq2 ($n = 3$ biologically independent samples per group). *$p < 0.05$, **$p < 0.01$, and ***$p < 0.001$. Source data and exact *p*-values are provided as a Source Data file.

calcium levels correlate with increased neuronal activation[33]. To achieve this, we employed the genetically encoded calcium indicator GCaMP6m, which facilitated the monitoring of calcium signals in excitatory neurons. We stereotactically infused the rAAV9-CaMKIIα-GCaMP6m virus into the hippocampus of FAM92A1-deficient mice and wild-type littermates, and implanted the optic fiber in the same place (Supplementary Fig. 5a). After 3 weeks of recovery from surgery, immunohistochemical assays confirmed the expression of GCaMP6m in the hippocampal DG area (Fig. 5a). We then recorded Ca²⁺ transients in a population of infected cells by monitoring the fluorescence intensity of GCaMP6m. The results unveiled that the frequency of spontaneous Ca²⁺ transients in hippocampal excitatory neurons was notably lower in FAM92A1 knockout mice than in wild-type littermates (Fig. 5b, c), indicating reduced neuronal excitability and firing rates in the absence of FAM92A1.

Next, to examine the impact of FAM92A1 on synaptic activity, we performed whole-cell recordings to compare the AMPA receptor-mediated excitatory postsynaptic current (EPSCs) between adult FAM92A1 knockout and wild-type littermates. Spontaneous EPSCs (sEPSCs) and miniature EPSCs (mEPSCs) were recorded in hippocampal DG granule neurons in acute slices. The absence of FAM92A1 resulted in a reduction in both sEPSC and mEPSC frequencies, indicating the impaired excitatory synaptic inputs following the loss of FAM92A1. Additionally, the amplitude of mEPSC was diminished upon FAM92A1 depletion, whereas the amplitude of sEPSC remained less affected (Fig. 5d, e). These data suggest that FAM92A1 knockout might affect the efficacy of neurotransmitter release during miniature synaptic events while having a lesser effect on the overall strength of synaptic transmission onto the postsynaptic neuron. The reduction in the frequency of sEPSC indicates that FAM92A1 depletion dampens overall synaptic activity (Fig. 5d), further supported by the decreased probability of glutamate release sites per neuron and the weakened strength of each site (as inferred from the frequency and amplitude of mEPSCs) (Fig. 5e). Overall, these results suggest that FAM92A1 knockout results in the dysfunction of excitatory synaptic transmission.

To further examine the impact of FAM92A1 deficiency on overall cellular excitability and network activity in larger cell populations, we conducted a comparative analysis of spontaneous electrical activity using microelectrode arrays (MEAs). Given the rarity of obtaining FAM92A1⁻/⁻ embryos, we primarily examined whether heterozygous loss of FAM92A1 could affect the neuronal network. Primary hippocampal neurons derived from FAM92A1⁺/⁺ and FAM92A1⁺/⁻ embryos were plated on MEA plates, and their spontaneous electrical activity was monitored at different days of culture (days in vitro (DIV) 7, 10, and 14) (Fig. 5f). Compared to FAM92A1⁺/⁺ neurons, a heatmap of spike rate (spikes/sec) presented a consistently lower spike rate in FAM92A1⁺/⁻ neurons across all time points, suggesting a decrease in regional connectivity among the neurons (Fig. 5g). The raw traces of voltage from the single electrode displayed a decreased spontaneous activity

of FAM92A1⁺/⁻ neurons at each time point (Fig. 5h). Both the quantified results of spikes per minute and mean firing rate also revealed reduced electrical activity in FAM92A1⁺/⁻ neurons during the neuronal maturation process (Fig. 5i, j). To confirm that the reduced electrical activity was the primary cause of these differences, but not due to other technical issues or poor neuron attachment, the number of active electrodes, defined by the detection of at least 5 spikes per minute, was counted. The results revealed that nearly all 16 electrodes were active at DIV14 in both groups (Fig. 5k), suggesting that the reduced spike rate and voltage resulted from the reduced electrical activity.

The coordinated electrical activity of neurons often organizes into neuronal bursts as well as network bursts, characterized by rhythmic and synchronous firing events. These network bursts signify the establishment of functional neuronal networks that are thought to facilitate activity-dependent development[34–36]. Having revealed the lower electrical activity, we subsequently sought to examine the impact of FAM92A1 depletion on the formation of synchronized network bursts. Compared to the FAM92A1⁺/⁺ neuronal activity in a 30 sec time frame, FAM92A1⁺/⁻ neurons showed sparse electrical activity, which slowly organized into bursts (blue line) as well as synchronized network bursts (marked with gray background) (Fig. 5l). Both the representative plots of synchrony and synchrony index, depicted by the area under the normalized cross-correlation graph, also revealed a reduced synchronous firing among FAM92A1⁺/⁻ neurons (Fig. 5m). Although synchronized neuronal activity was similar between the two groups at DIV14, the number of electrodes participating in the network bursts was still lower in FAM92A1⁺/⁻ neurons (Fig. 5k, m). These data indicate that following the absence of FAM92A1, some neurons could form synchronized network activity, while the electrical activity of some neurons is still defective in organizing bursts. Interestingly, compared to neuronal networks formed by FAM92A1⁺/⁺ neurons at DIV14, those FAM92A1⁺/⁻ neurons involved in synchronized network bursts were revealed to generate a high frequency of bursts and network bursts (Fig. 5l, n and Supplementary Fig. 5b). However, the number of spikes within each burst and network bursts, as well as the duration of bursts and network bursts, were decreased in FAM92A1⁺/⁻ neurons (Fig. 5o, p and Supplementary Fig. 5c). Combined with the minor extended interval time between two bursts in FAM92A1⁺/⁻ neurons (Supplementary Fig. 5d), these data collectively suggested that FAM92A1 depletion leads to the dysregulated formation and function of excitatory synapses.

## FAM92A1 depletion results in aberrant membrane remodeling in neurons

To investigate the underlying mechanisms through which FAM92A1 influences neural plasticity, the impact of FAM92A1 depletion on mitochondrial morphology and function was examined using electron microscopy and a Seahorse XFp Extracellular Flux analyzer, respectively. In line with our earlier observations in cultured osteosarcoma cells, the absence of FAM92A1 prompted evident alterations in the

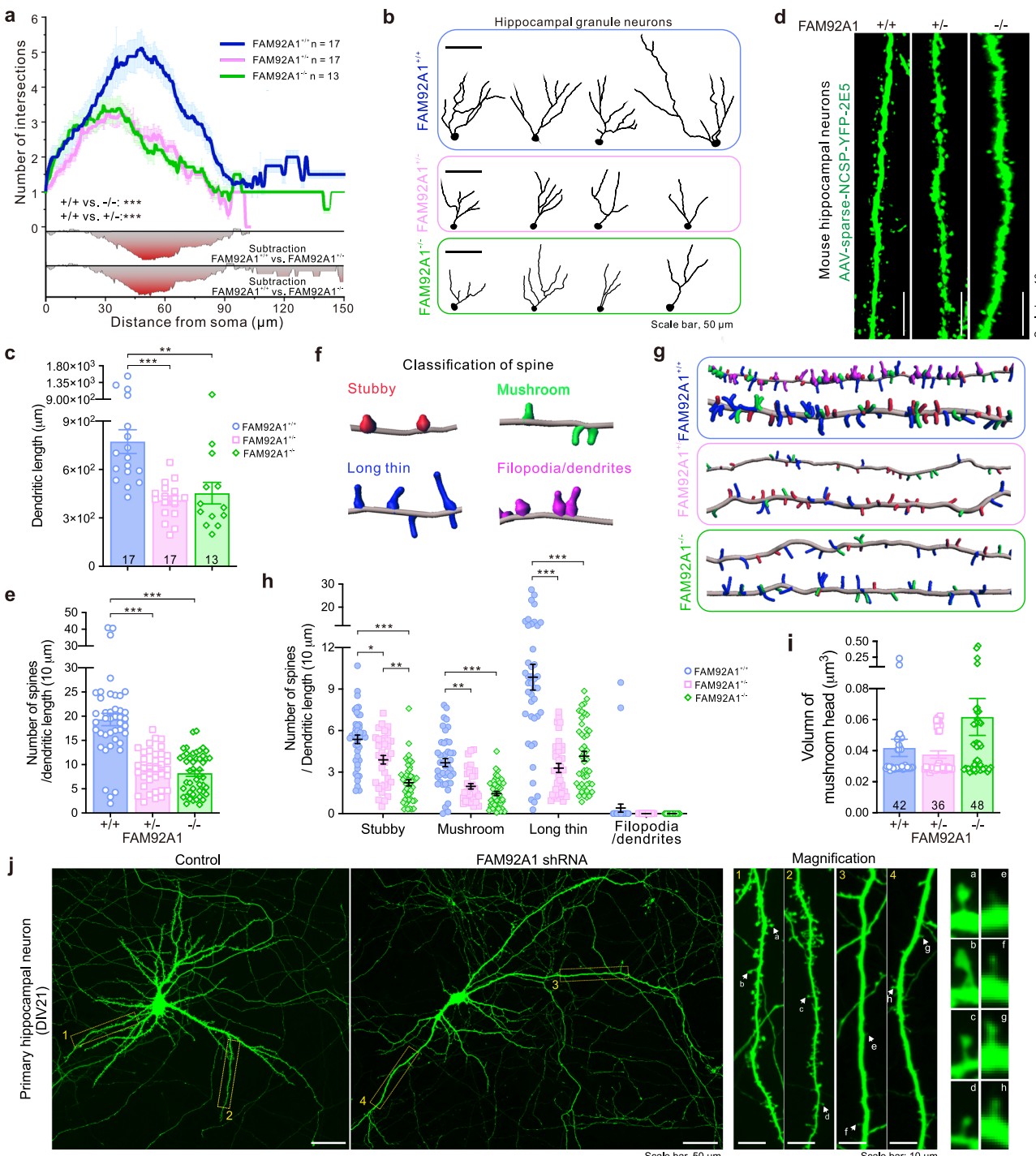

**Fig. 4 | FAM92A1 depletion leads to reduced neuronal complexity and altered spine morphology. a** Sholl analysis of the number of intersections at each radial increment. Heatmaps below show the difference in the number of intersections between the two indicated groups. **b** Representative images showing the traced neuronal dendrites. Scale bar, 50 μm. **c** Bar graph showing the total length of each traced neuron. For (**a**) and (**c**) data represent mean ± SEM; $n = 17$ neurons per group (FAM92A1$^{+/+}$ and FAM92A1$^{+/-}$ group) and $n = 13$ neurons for FAM92A1$^{-/-}$ group from three mice per group; two-way ANOVA (**a**) and one-way ANOVA (**c**). **d** Representative images of dendritic segments from five mice per group injected with AAV-sparse-NCSP-YFP-2E5 viruses into the HPC. Scale bar, 10 μm. **e** Bar graph showing the number of spines. **f** Representative images showing four types of reconstructed spines by the Imaris Spines Classifier. Spines classified as stubby, mushroom, long thin, and filopodia/dendrite were shown in red, green, blue, and

magenta colors, respectively. **g** Representative reconstructed spine morphology of each experimental group. Each class of spine was shown with the same color in panel (**f**). **h, i** The number of each class of spine per 10 μm of dendritic length (**h**) and the head volume of mushroom spines (**i**). For (**e, h,** and **i**) data represent mean ± SEM; dendritic segments ($n = 42$ for FAM92A1$^{+/+}$ group, $n = 36$ for FAM92A1$^{+/-}$ group, and $n = 48$ for FAM92A1$^{-/-}$ group) from five mice per group; one-way ANOVA. **j** Representative images of three biologically independent experiments showing the morphology of primary hippocampal neurons at DIV18 after transfected with control and FAM92A1 shRNA for 48 h. Scale bar, 50 μm. The spine ultrastructure was visualized in the magnified images of the right panel. Scale bar, 10 μm. *$p < 0.05$, **$p < 0.01$, and ***$p < 0.001$. Source data and exact $p$-values are provided as a Source Data file.

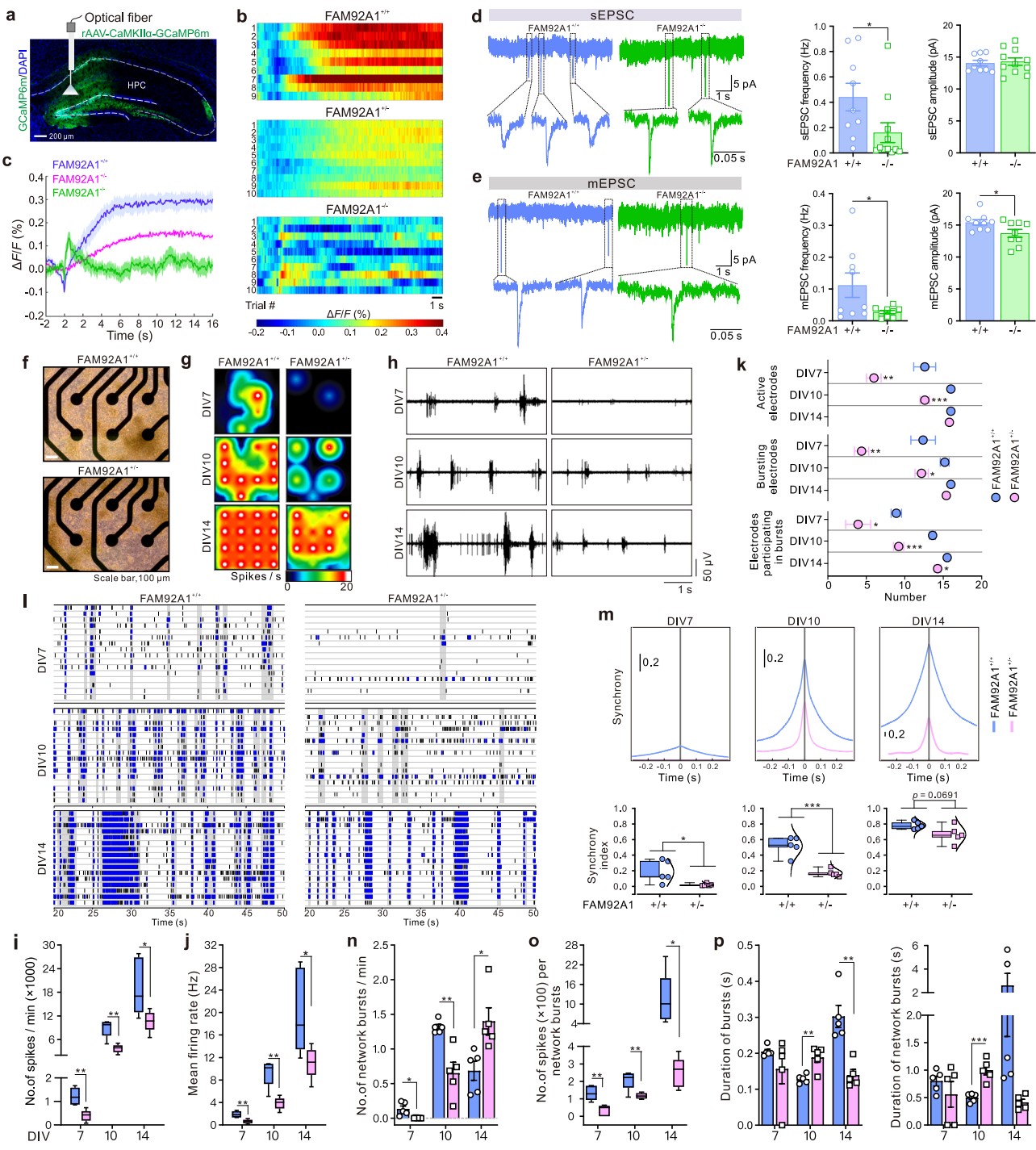

**Fig. 5 | FAM92A1 depletion perturbs neuronal function. a** Representative image showing GCaMP6m expression in mouse HPC. **b**, **c** Heatmaps and line graph showing the calcium signals in hippocampal neurons of freely behaving mice. Data represent mean ± SEM; recordings (*n* = 9 for FAM92A1+/+ group, *n* = 10 per group for FAM92A1+/− and FAM92A1−/− groups) from two to five mice per group. **d**, **e** (Left) representative traces of sEPSC (**d**) and mEPSC (**e**) in granule neurons of hippocampal DG region. (Right) Bar graph showing the changes in the frequency and amplitude during the recording of sEPSC (**d**) and mEPSC (**e**). Data present mean ± SEM; *n* = 9 neurons for FAM92A1+/+ group and *n* = 11 neurons for FAM92A1−/− group (**d**), *n* = 9 neurons per group (**e**) from three mice per group; unpaired two-tailed Student's *t*-test. **f** Representative bright-field images of primary hippocampal neurons grown on MEA plate at DIV10 from five MEA wells per group. Scale bar, 100 μm. **g** Activity heatmaps showing spike rate (spikes per second). **h** Traces of voltage showing neuronal spontaneous activity. **i, j** Box plots showing the number of spikes per minute (**i**) and mean firing rate (Hz) (**j**). **k** Mean plot showing the number of active electrodes, bursting electrodes, and electrodes participating in bursts. **l** Raster plots showing neuronal network activity in a 30 sec time frame. The gray bars represent the detected network bursts. **m** (Top) Cross-correlogram showing synchronous activity. (Bottom) Box plot showing the synchrony index. **n**–**p** The number of network bursts per minute (**n**) and spikes per network bursts (**o**), and the duration (sec) of bursts (left) and network bursts (right) (**p**). Data presented as minimum to maximum values (whiskers) and the 25th to 75th percentiles (box), with the mean value within the box (**i, j, m**, and **o**) and data represent mean ± SEM (**k, n**, and **p**); *n* = 5 MEA wells per group; multiple unpaired two-sided *t*-tests. **p* < 0.05, ***p* < 0.01, and ****p* < 0.001. Source data and exact *p*-values are provided as a Source Data file.

architecture of the mitochondrial inner membrane, with some mitochondria even exhibiting a lack of membrane invaginations (Fig. 6a). Quantitative analysis of mitochondrial morphology and inner membrane architecture profiles revealed an increase in mitochondrial diameter in FAM92A1$^{-/-}$ mitochondria, with less impact on its perimeter, indicating the appearance of swollen mitochondrial morphology. Additionally, the number of lamellar cristae inside mitochondria decreased with the loss of FAM92A1. These findings in mouse brains further confirm that FAM92A1 is required for maintaining mitochondrial morphology and inner membrane ultrastructure (Fig. 6b). Having revealed the changes in mitochondrial morphology, we subsequently examine the consequences of FAM92A1 depletion on mitochondrial function. Mitochondria were isolated from the fresh hippocampus and the oxygen consumption rate (OCR) of mitochondria was measured. Compared to mitochondria from the wild-type group, FAM92A1 depletion caused a notable reduction in OCR (Fig. 6c–e). Consistent with these findings, the hippocampal ATP content was also diminished in mice lacking FAM92A1 (Fig. 6f). The decline in OCR and ATP levels implies that mitochondria in FAM92A1-depleted neurons are less proficient in utilizing oxygen to generate ATP through oxidative phosphorylation. Taken together, the impaired mitochondrial architecture, reduced OCR, and lower ATP content collectively underscore the essential role of FAM92A1 in upholding proper mitochondrial structure and function.

Furthermore, within the hippocampus of mice lacking FAM92A1 (FAM92A1$^{+/-}$ and FAM92A1$^{-/-}$), several axons displayed abnormal myelin sheaths (Fig. 6g). Unlike the well-organized and closely aligned myelin sheaths observed in the hippocampus of wild-type mice (FAM92A1$^{+/+}$), the myelin sheaths in FAM92A1-deficient mice exhibited characteristics such as looseness, rupture, and distortion. Some axons even had double myelin sheaths (Fig. 6g, indicated with orange arrowheads). Additionally, the ratio between the inner and outer diameters of the myelin sheath, known as the g-ratio[37], was reduced in FAM92A1 heterozygous mice (Fig. 6h, i). Typically, a decreased g-ratio can be attributed to either a reduction in axon caliber or an increase in the amount of myelin surrounding each axon. To determine the underlying cause of the decreased g-ratio, we conducted additional measurements of myelin thickness. Compared to the wild-type group, FAM92A1 heterozygous mice exhibited an increase in myelin sheath thickness (Fig. 6i), indicating aberrations in the myelin membrane. In contrast, FAM92A1 homozygous mice displayed a more pronounced reduction in axon diameter (Fig. 6h, inset). Considering the sequential changes in the axonal morphology during neuronal degeneration, progressing from swelling to eventual breaking at the thinned part[17], these findings suggest heightened neuronal damage following FAM92A1 knockout. Collectively, the atypical appearance of the myelin sheath, reduced g-ratio, and altered relationship between g-ratio and axon diameter imply that the absence of FAM92A1 likely disrupts in myelin formation and maintenance.

Importantly, we observed a reduced thickness of the postsynaptic density (PSD) (Fig. 6j-l), a protein-rich region in the postsynaptic compartment that is essential for the neurotransmitter reception and signal transduction[38]. The reduced thickness of the PSD in synapses of FAM92A1-deficient mice indicates structural and compositional alterations in the postsynaptic region. Along with the reduced thickness, the PSD length was extended in FAM92A1-deficient mice (Fig. 6k, l), suggesting an elongated arrangement of postsynaptic components. Additionally, the area of synaptic vesicles (SVs) was enlarged in the FAM92A1-deficient mice (Supplementary Fig. 5e, f). To precisely visualize the changes in the ultrastructure of SVs in hippocampal neurons in their native state, we applied a focused ion beam (FIB) and a scanning electron microscope (SEM) (FIB-SEM) imaging system for three-dimensional (3D) reconstruction of SVs[39]. Following the 3D reconstruction, all SVs were categorized into three groups based on their diameter (<60 nm, 60–80 nm, and >80 nm) and

visualized in different colors accordingly. The morphology of SVs in pre-synaptic boutons of wild-type hippocampal neurons was homogeneous, with the majority exhibiting a typical spherical shape. In contrast, some large and pleomorphic SVs (>80 nm, yellow color) appeared in the presynaptic boutons of FAM92A1-deficient neurons (Fig. 6m). By analyzing micrographs from every 50th section, the area of all SVs in synapses with distinguishable pre- and post-synaptic structures was quantified across a total of 4500 nm depth (Fig. 6n). Compared to SVs in wild-type neurons, the total number of SVs was clearly higher in FAM921$^{-/-}$ neurons (Fig. 6o). Although the majority of SVs had a diameter <60 nm, there was a higher proportion of SVs with a diameter >80 nm in FAM921$^{-/-}$ neurons (Fig. 6p). Additionally, consistent with the measured increase in SV area using 2D electron micrographs, the average SV area was also found to be larger in FAM92A1$^{-/-}$ neurons compared to FAM92A1$^{+/+}$ neurons (Fig. 6q). The abnormal morphology and heterogeneity in the size and shape of SVs point to disturbances in the formation and dynamics of SVs in the absence of FAM92A1.

Overall, the disruption of synaptic architecture and alterations in the morphology of SVs suggest that FAM92A1 is essential for maintaining the precise organization and function of synapses. The abnormalities in multi-membrane morphologies, including crista, myelin sheath, and synapses, indicate that FAM92A1 plays an important role in membrane remodeling and maintaining proper membrane organization in the brain.

## FAM92A1 localizes to synapses and loss of FAM92A1 leads to defects in endocytosis

To investigate the mechanism underlying FAM92A1 deficiency-induced abnormalities in membrane morphology, the ranked gene lists acquired from the RNA-seq data were subjected to the Gene Set Enrichment Analysis (GSEA) referring to the curated gene sets (c2.all.v7.5.1.symbols.gmt). The GSEA revealed several significantly enriched pathways associated with FAM92A1, including cell surface interaction, mitochondrial protein import, and E-cadherin stabilization pathway (Supplementary Fig. 6a). The enrichment of these pathways suggests a potential involvement of FAM92A1 in the remodeling of other cellular membranes beyond the mitochondrial inner membranes. Thus, we investigated the subcellular localization of FAM92A1 in hippocampal neurons using immunofluorescent staining, immuno-electron microscopy, and subcellular fractionation analysis. Consistent with our previous findings, a significant portion of endogenous FAM92A1 was found in mitochondria, as evidenced by its colocalization with the mitochondrial protein TOM20 in HT22 cells (Fig. 7a). Moreover, a small fraction of endogenous FAM92A1 was observed at the cell periphery, separate from its mitochondrial localization (Fig. 7a, b). This peripheral localization suggests a potential role for FAM92A1 in membrane trafficking at the cell edge. Similar localization patterns were observed when FAM92A1-GFP was introduced exogenously in HT22 cells (Supplementary Fig. 6b). In primary hippocampal neurons, both endogenous and overexpressed FAM92A1 exhibited granular puncta along the neuronal dendrites (Fig. 7c and Supplementary Fig. 6c). These puncta exhibited partial colocalization with synaptic proteins vGlut1, SV2, and PSD95, in addition to colocalization with mitochondrial protein VDAC (Fig. 7d and Supplementary Fig. 6d), indicating the presence of FAM92A1 at synaptic terminals. Furthermore, when exogenously expressed as FAM92A1-EGFP, the protein was distributed along dendrites, showing colocalization with both vGlut1 and PSD95 (Supplementary Fig. 6d), supporting its presence at synapses. Additionally, subcellular fractionation analysis revealed an enrichment of FAM92A1 in the synaptosome membrane fraction (LP1) (Fig. 7e). This fraction contains synaptosomes, which are vesicles derived from both presynaptic terminals and postsynaptic densities, further confirming the synaptic localization of FAM92A1.

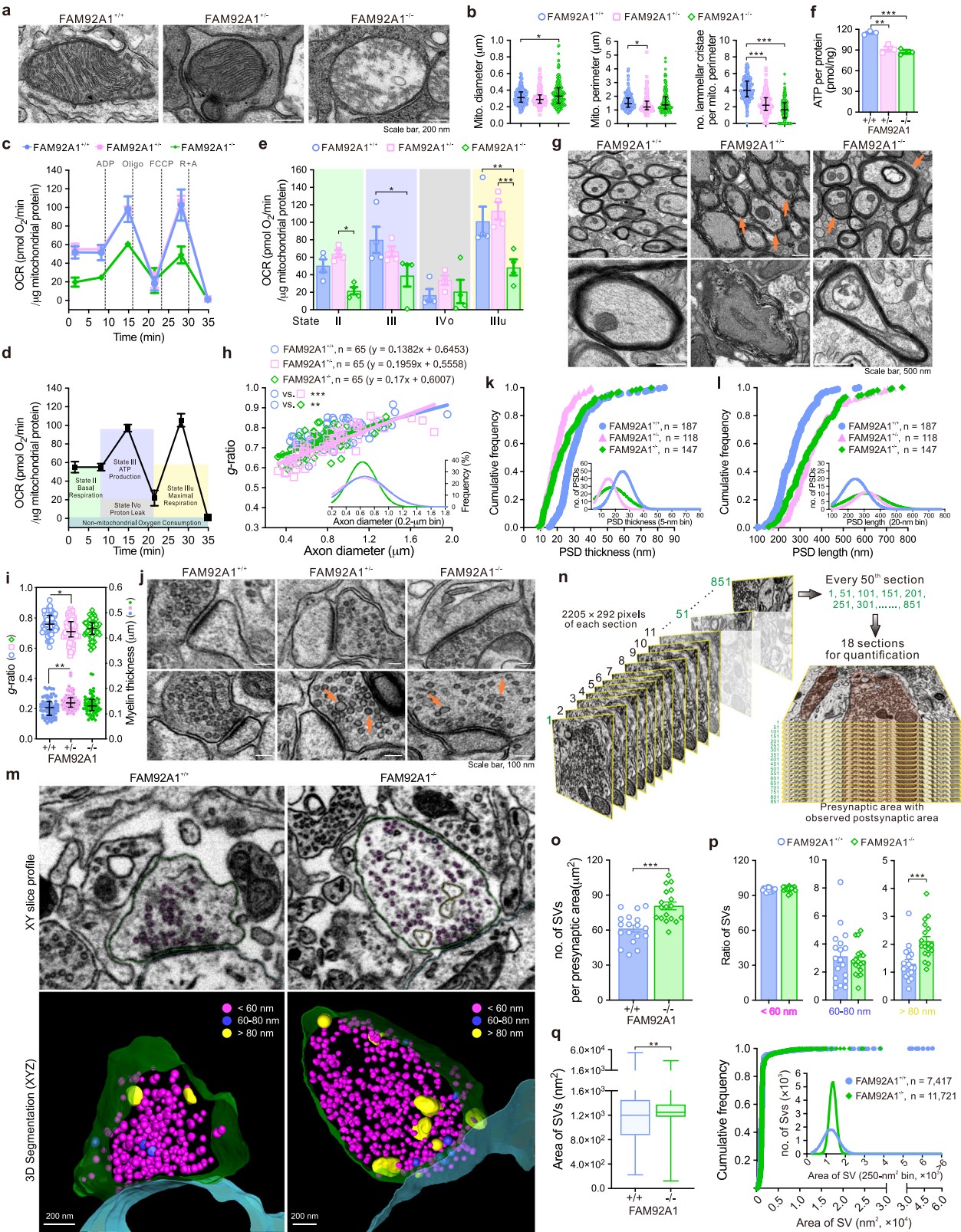

The presynaptic membrane is dynamically involved in circular exo- and endocytosis, which are necessary for synaptic transmission and brain function[40]. To delve into the role of FAM92A1 localized at synapses, we conducted endocytosis analysis using primary fibroblasts isolated from the ears of both wild-type and FAM92A1-deficient mice (Supplementary Fig. 6e–g). The fluorescent dye FM1–43, commonly employed to investigate endocytosis, was used to assess the endocytic

activities of the fibroblasts. Notably, primary FAM92A1$^{-/-}$ fibroblasts exhibited reduced FM1–43 uptake in comparison to fibroblasts from FAM92A1$^{+/-}$ and FAM92A1$^{+/+}$ mice (Fig. 7f). This observation suggests that the absence of FAM92A1 led to an endocytic impairment in FAM92A1$^{-/-}$ fibroblasts. The mean fluorescence intensity (MF, a.u.) of FM1–43 in individual FAM92A1$^{-/-}$ fibroblasts decreased by ~32% when compared to wild-type fibroblasts (FAM92A1$^{+/+}$: 47 ± 2 a.u.; FAM92A1$^{-/-}$:

**Fig. 6 | FAM92A1 depletion results in aberrant membrane remodeling in neurons. a** Representative electron micrographs showing mitochondrial morphology. Scale bar, 200 nm. **b** Quantification of mitochondrial diameter, perimeter, and the number of lamellar cristae in the FAM92A1$^{+/+}$ ($n = 168$), FAM92A1$^{+/-}$ ($n = 167$), and FAM92A1$^{-/-}$ ($n = 177$) groups. Data present mean ± SEM; one-way ANOVA. **c** OCR of hippocampal mitochondria. **d, e** Schematic depicting OCR (**d**) and quantified OCR (**e**) in each state. For **c** and **e**, data represent mean ± SEM of four biologically independent experiments; two-way ANOVA. **f** ATP content in the HPC. Data represent mean ± SEM of three biologically independent experiments; one-way ANOVA. **g** Representative electron micrographs showing myelin sheath, with abnormal myelin indicated by orange arrowheads. Scale bar, 500 nm. **h, i** Scatter plot with best-fit lines (**h**) and quantification of g-ratio and myelin thickness (**i**). Gaussian curve (**h**, inset) showing the reduced axon diameter upon loss of FAM92A1. Data represent mean ± SEM; $n = 65$ axons per group; Kruskal–Wallis $H$ test (**h**), ANCOVA (**i**, g-ratio), and one-way ANOVA (**i**, myelin thickness). **j** Representative electron micrographs showing synaptic contacts, with enlarged SVs indicated by orange arrowheads. Scale bar, 100 nm. **k, l** Cumulative frequency showing changes in PSD thickness (**k**) and length (**l**). Gaussian curve showing the reduced PSD thickness (inset, **k**) and elongated PSD length (inset, **l**) upon loss of FAM92A1. For (**a, b**) and (**g–l**) data from three mice per group. **m** Representative electron micrographs (top) and 3D reconstruction (bottom) of presynaptic compartments. Scale bar, 200 nm. **n** Schematic illustrating the method for quantifying SVs using FIB-SEM data. **o–q** Quantification of SV number (**o**), ratio (**p**), and area (**q**). Data represent mean ± SEM; $n = 18$ sections per group (**o, p**) and data presented as minimum to maximum values (whiskers) and the 25th to 75th percentiles (box), with the mean value within the box; $n = 7417$ SVs for FAM92A1$^{+/+}$ and $n = 11,721$ SVs for FAM92A1$^{-/-}$ (**q**); unpaired two-tailed Student's t-test. *$p < 0.05$, ** $p < 0.01$, and *** $p < 0.001$. Source data and exact p-values are provided as a Source Data file.

33 ± 2 a.u.) (Fig. 7g). The decreased uptake of FM1–43 was not attributed to a decrease in cell area (Supplementary Fig. 6h), indicating that the endocytic defect is not due to alterations in cell size. The MF intensity per cell area also exhibited similar trends (Fig. 7h). To gain further insights into the endocytic process, live cell imaging was employed. The data revealed a much slower uptake rate of FM1–43 in fibroblasts from FAM92A1-deficient mice when compared to wild-type fibroblasts (Fig. 7i and Supplementary Movies 1–3). While FM1–43 was rapidly internalized by FAM92A1$^{+/+}$ fibroblasts within 150 s, the uptake was delayed to 180 s in FAM92A1$^{+/-}$ fibroblasts, and notably, FM1–43 uptake was not observed within the 450 s observation period in FAM92A1$^{-/-}$ fibroblasts. To further quantify the endocytic defect, a fluorescent latex bead uptake assay was conducted. After 8-h incubation with latex beads (1 μm), the phagocytic activity decreased from 63% in fibroblasts from wild-type mice to 52% in fibroblasts from FAM92A1$^{-/-}$ mice (Fig. 7j, k and Supplementary Fig. 6i). These findings collectively illustrate that FAM92A1 knockout results in defective endocytosis in fibroblasts, suggesting the potential role of FAM92A1 in regulating endocytic processes at synapses.

To investigate the link between aberrant synaptic morphology and reduced endocytosis in FAM92A1-deficient mice, we examined the expression levels of several proteins involved in various stages of endocytosis (Fig. 7l, m). Interestingly, both the transcriptional and translational levels of sorting nexin 9 (SNX9) decreased in FAM92A1-deficient mice. In contrast, clathrin, a key component of clathrin-coated pits formed during endocytosis, exhibited increased expression in the hippocampus of these mice. Despite these changes in SNX9 and clathrin abundance, the levels of Dynamin 2 (Dnm2 gene), which is involved in the final fission step of endocytosis, remained unaltered in the hippocampus of FAM92A1-deficient mice (Fig. 7l, m). These data further support the role of FAM92A1 in endocytosis, which is essential for the biogenesis and dynamics of synaptic vesicles. Intriguingly, depletion of FAM92A1 has a notable effect on the shape of synaptic vesicle intermediates. Specifically, FAM92A1 deficiency led to a pronounced increase in the frequency of U-shaped vesicle intermediates, accompanied by a reduction in the occurrence of Ω-shaped constricted vesicle intermediates (Fig. 7n, o), indicating the requirement of FAM92A1 for constriction of the endocytic membrane. The retrieval of endocytic membrane relies on both clathrin-dependent and -independent endocytosis[41,42]. To reveal whether FAM92A1 was specifically involved in the endocytic pathway, we examined the colocalization of plasma membrane-resident FAM92A1 with the clathrin-dependent and -independent endocytic proteins clathrin and caveolin[43], respectively, using total internal reflection microscopy (TIRFM). The results revealed that FAM92A1 colocalized with both proteins (Fig. 7p), indicating the involvement of FAM92A1 in both endocytic pathways, a function likely executed by another BAR domain proteins, such as endophilin[44,45]. These findings imply that FAM92A1 is required for effectively narrowing the neck of vesicles during the endocytic process.

## FAM92A1 is a classical BAR domain protein and interacts with membranes through a specific concave surface

FAM92A1 possesses a BAR domain at the N-terminus and a relatively short C-terminal region[23]. Despite this known domain arrangement, the crystal structure of FAM92A1 has not been characterized yet. Thus, to elucidate the structural basis for its membrane remodeling activity, we purified and crystallized the FAM92A1 N-terminal region (residues 1–219) (Supplementary Fig. 7a). The crystal structure of FAM92A1 BAR domain was solved at 2.03 Å resolution (Fig. 8a and Supplementary Fig. 7b). Based on the crystal structure, the region spanning residues 2–211 was coiled-coil composed of three α helices, designated as the FAM92A1 BAR domain (Supplementary Fig. 1e). In the asymmetric unit of the crystals, two FAM92A1 BAR domain monomers assembled via a two-fold symmetric interface, yielding a banana-shaped homo-dimer characteristic for BAR-domain proteins (Fig. 8a). Similar to other BAR domains, the monomer of FAM92A1 BAR domain consists of three long α-helices connected via loops. The first helix (α1) encompasses residues 2–61, followed by a long second helix (α2) spanning residues 64 – 136. Notably, α2 features a noticeable bend at Pro98, a positioning analogous to other BAR domains[46]. This kink is situated at the interface of the six-helix bundle dimerization core and the extending helical arm. The BAR domain is finalized by a last long helix (α3) covering amino acids 142–211 (Fig. 8a). In the second chain α2 helix, the loop between α2 and α3 is partially disordered from Lys125 to His142, indicating some flexibility in the structure at the distal ends.

The overall dimensions of the dimeric BAR domain are 150 Å in total length and ca. 25 Å in diameter around the center of the six-helical dimer bundle (Fig. 8a, b). The dimer interface is extensive with a total interface area of 2532 Å$^2$ involving 65 hydrophobic residues with >70% buried surface area at the interface and 11 hydrogen-bonded interactions across the interface (Fig. 8c and Supplementary Fig. 7c–e). The extensive dimer interface consists of a number of buried aromatic and hydrophobic residues aligned along the helical interface between the monomers in the central two-fold symmetrical six-helix dimeric bundle. In addition to the extensive hydrophobic buried interface area, the distal ends of the six-helix bundle also have several hydrogen-bonded interactions that contribute to the dimer interface. These interactions are particularly prominent between residue pairs Glu62-Lys176, His26-Glu54, and Asn22-Thr61 of the two chains at each end of the dimeric six-helical bundle (Fig. 8c and Supplementary Table 1).

To investigate the significance of these residues in the dimerization process of the FAM92A1 BAR domain, we simultaneously mutated three hydrophobic residues at the dimer interface observed in the structure to negatively charged residues, generating a mutant construct (FAM92A1 Leu70Glu-Phe73Glu-Phe184Glu, abbreviated as mutant) (Supplementary Fig. 7f). These mutations aimed to disrupt the dimer interface packing. Following the expression and purification of mutant proteins, the dimerization states of wild-type (WT) and mutant FAM92A1 BAR domains were assessed via SEC-MALS (size-exclusion

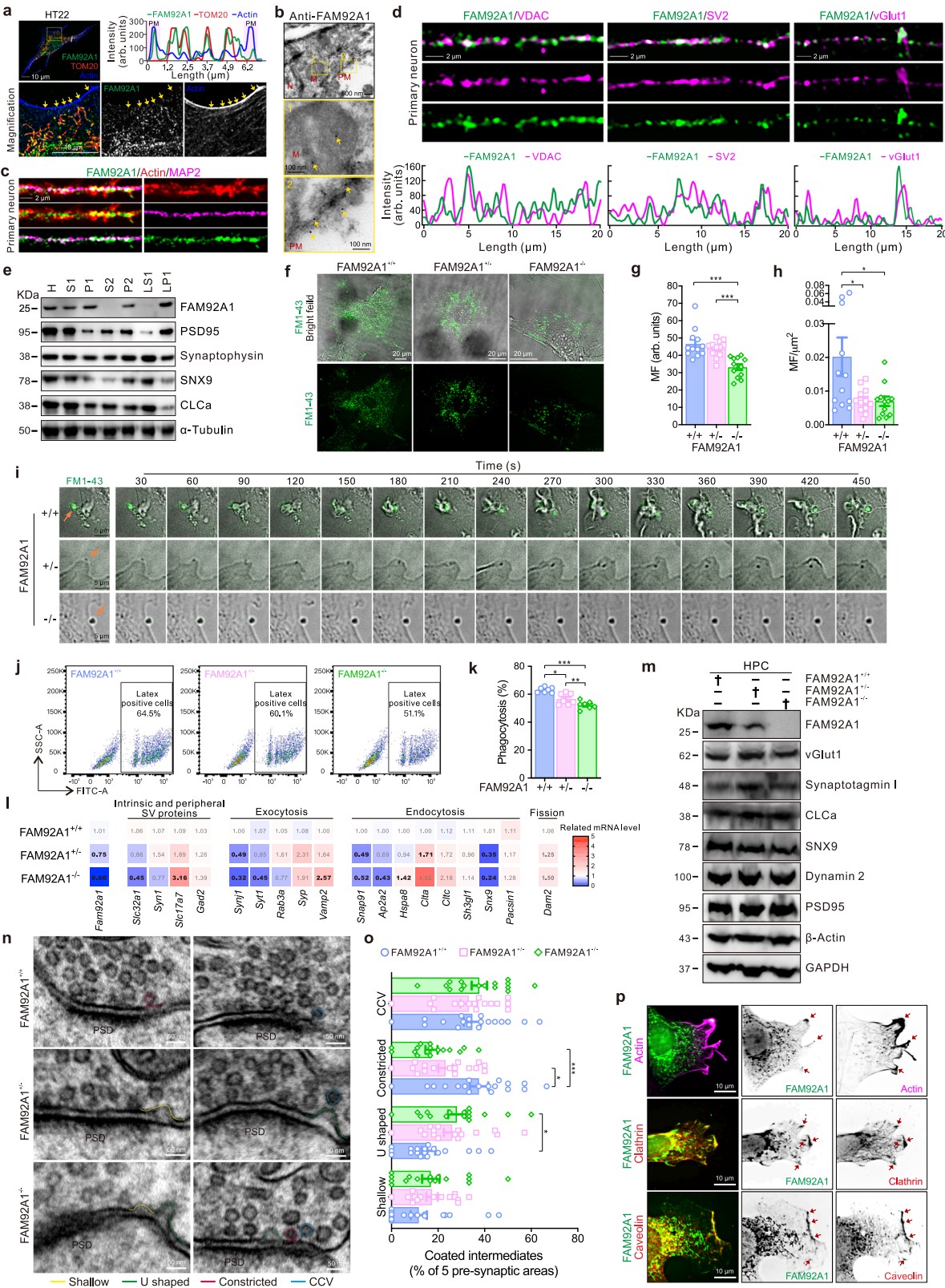

chromatography coupled to multi-angle light scattering) experiment. Consistent with our previous result[7], the wild-type FAM92A1 BAR domain protein primarily existed as dimer. In contrast, the mutant protein primarily existed as monomer in equilibrium (Supplementary Fig. 7g), confirming the involvement of these residues in FAM92A1 BAR dimerization. Although the interface mutant lost its capacity to form dimers compared to WT, co-sedimentation assay revealed that the

mutant preserved its membrane-binding capability with only a slight decrease compared to the WT (Supplementary Fig. 7h). However, despite retaining affinity for liposomes, the mutant lost its ability to sculpt spherical liposomes into narrow tubules (Supplementary Fig. 7i). These findings collectively suggest that dimerization of the FAM92A1 BAR domain is necessary for membrane sculpting but not for membrane binding.

**Fig. 7 | FAM92A1 localizes to synapses engaging in endocytosis.**
**a**, **b** Immunofluorescence (**a**) and immunoEM (**b**) images showing the subcellular localization of FAM92A1 in mitochondria (M), plasma membrane (PM), and nucleus (N). M and PM were visualized by TOM20 and phalloidin, respectively. Line graph showing the fluorescence intensity of the indicated white line (**a**). Scale bars, 10 μm (**a**), 500 nm (up, **b**) and 100 nm (magnifications, **b**). **c**, **d** Representative images showing the expression (**c**) and co-localization of FAM92A1 with mitochondrial and synaptic proteins (**d**) in neurons. Scale bar, 2 μm. Line graphs (**d**) showing the fluorescence intensity of the indicated proteins. **e** Western blots showing subcellular localization of FAM92A1 in the HPC across three biologically independent experiments. **f**–**h** Representative fibroblasts loaded with FM1–43 (**f**) and mean fluorescence (MF) intensity of FM1–43 in single fibroblasts (**g**) and per cell area (μm²) (**h**). Scale bar, 20 μm. Data represent mean ± SEM; n = 12 fibroblasts per group; one-way ANOVA. **i** Representative images showing the endocytic process of FM1–43 by fibroblasts. Scale bar, 5 μm. **j**, **k** Representative images (**j**) and

quantification (**k**) of the phagocytic capacity of fibroblasts using the fluorescence-labeled latex beads. Data represent mean ± SEM; n = 7 assays per group; one-way ANOVA. **l**, **m** Heatmap and western blots showing the mRNA (**l**) and protein (**m**) expression level of genes involved in endocytosis across three biologically independent experiments. Numbers represent the normalized mRNA level, with significance in bold text (**l**). **n**, **o** Representative electron micrographs (**n**) and the frequency of endocytic intermediates (**o**). Scale bar, 50 nm. Data represent mean ± SEM; n = 18 samples from 92 synapses for FAM92A1⁺/⁺ group, n = 16 samples from 82 synapses per group for FAM92A1⁺/⁻ and FAM92A1⁻/⁻ groups; two-way ANOVA. **p** TIRFM images showing the colocalization of FAM92A1 with the indicated proteins. Scale bar, 10 μm. Representative images are from at least three (**a**, **c**, **d**, **i**, **j**, **n**, and **p**) and two (**b**, **f**) biologically independent experiments. *p < 0.05, **p < 0.01, and ***p < 0.001. Source data and exact p-values are provided as a Source Data file.

To determine the subfamily of FAM92A1, we performed a comparative analysis between the dimeric structure of the FAM92A1 BAR domain and representative structures of other BAR domains (Fig. 8d). Through this alignment of dimers, we found that the curvature of the FAM92A1 BAR domain closely resembles that of the classical BAR domain, such as the SNX9 PX-BAR domain (PDB ID 2RAI)[47]. Specifically, the angle between the α2 helices of the two chains in the FAM92A1 BAR domain is 33.2°, a value that is consistent with other classical BAR domains, including N-BAR domain proteins. In contrast, some domains like the APPL1 BAR-PH dimers exhibit a significantly more pronounced curvature (54°) as depicted in Fig. 8d. Moreover, the curvature is significantly less (ca. 10°) for the different flatter F-BARs domains, and more obviously compared to the I-BAR domains which have inverse curvature (Fig. 8d)[48]. Also, the FAM92A1 does not possess any additional helices like some of the BAR domain family proteins, including the N-BAR and F-BAR domains, and the APPL BAR-PH which has a long fourth helix linking to the C-terminal PH domain.

According to previous studies, the concave surface of the BAR domain has been shown to interact with the lipid bilayer of cellular membranes[19]. Consistently, the membrane interacting residues identified in our previous in vitro lipid-binding assay[7], including Lys107, Lys109, and Arg110 and Arg132, Arg134, and Arg136, form two positive charged clusters located on the concave surface of the FAM92A1 BAR domain dimer (Fig. 8a, b). These positively charged clusters on the concave surface play a pivotal role in facilitating interactions with the negatively charged phospholipids present in membranes. Furthermore, an analysis of the electrostatic potential surface of the FAM92A1 BAR domain revealed the characteristic accumulation of positive electrostatic potential/charge on the concave surface (Fig. 8b). This charge distribution encompasses the α1 and α2 helices, as well as the far-end of the monomer arm, which includes the membrane binding residues Arg132 and Arg136 (Fig. 8a, b).

Moreover, the crystal structure elucidates the potential oligomerization of the FAM92A1 BAR domain, unveiling an interface for oligomerization spanning ~850 Å². This involves the head-to-head packing of coiled-coil dimers (Fig. 8e and Supplementary Fig. 8a). This possible oligomerization interface encompasses several polar residues that form a hydrogen-bonded network and has a high confidence score for a protein-protein interface by the PISA-server (https://www.ebi.ac.uk/msd-srv/prot_int/). The residues contributing to this network include Arg9, Arg141, Arg134, Arg120, and Glu148 from one monomer, and Asn10, Arg9, Asn121, and Glu148 from the second partly disordered chain of the other dimer (Supplementary Fig. 8b, c). A second parallel packing interface is observed in the crystal that could contribute to oligomerization but appears less significant (Supplementary Fig. 8a, d). It's noteworthy that BAR domain proteins are recognized for their unique capacity to undergo oligomerization. This property holds significant importance for their function in reshaping membranes and governing lipid clustering[49,50]. The oligomerization of BAR domains is

primarily driven by their characteristic curved, banana-like structure, which facilitates the assembly of multiple subunits into higher-order complexes.

To gain a deeper understanding of the molecular mechanisms and dynamics underlying membrane interaction and curvature generation by FAM92A1, we applied all atom molecular dynamics simulations. Our previous findings demonstrate that FAM92A1 preferentially binds the negatively charged phospholipids cardiolipin and PI(4,5)P₂, which are hallmark lipids in the mitochondrial inner membrane and plasma membrane respectively[7]. To investigate the roles of these negatively charged lipids in facilitating the FAM92A1 BAR domain interaction with membranes, we carried out simulations of a FAM92A1 monomer (residues 1–214) with model lipid bilayers containing PI(4,5)P₂ (PIP₂ bilayer) or cardiolipin (Cardiolipin bilayer) (Supplementary Table 2). The results revealed that the positively charged amino acid residues from the α1 and α2 helices of the FAM92A1 BAR domain showed increased interaction with the negatively charged phospholipids (Fig. 8f and Movie 4). In addition, another membrane binding site on the α2 helix of FAM92A1 protein was identified. This observation is in line with our previous experimental results showing the preferential interactions of FAM92A1 with these negatively charged lipids[7]. The negatively charged lipids appeared to cluster around the binding sites on the FAM92A1 BAR domain (Fig. 8g and Movie 4). Furthermore, we observed that the binding sites showed a relatively higher association with PI(4,5)P₂ compared to cardiolipin, suggesting that PI(4,5)P₂ promotes a more stable interaction of FAM92A1 BAR domain with membranes (Fig. 8f).

While the negatively charged phospholipids facilitate FAM92A1 BAR domain interaction with membranes, the concave surface of the dimer possibly promotes remodeling of the membrane curvature. To validate this hypothesis, we carried out long (μs) timescale simulations of the FAM92A1 BAR domain dimer with a model lipid bilayer having a membrane composition resembling the mitochondrial inner membrane (MIM bilayer) (Supplementary Tables 2 and 3). The dimer was placed on top of the lipid bilayer such that the concave surface faced the bilayer (Fig. 8h). The BAR domain dimer bound to the bilayer shortly after the start of the simulation and remained associated with it for the rest of the simulation period. Interestingly, after binding to the lipid bilayer, the FAM92A1 BAR domain dimer underwent a conformational change resulting in a decrease of overall domain curvature, suggesting that the BAR domain dimer is not a rigid structure (Fig. 8h, i). The dimer, however, induced a stable positive curvature in the membrane leaflet to which it was bound (Fig. 8h, j, upper panel). This was significant compared to the instantaneous local curvature effects in the membrane leaflet in the absence of BAR dimer binding (Fig. 8j, lower panel). These observations indicate that BAR domain-induced membrane curvature is a complementary process in which the membrane undergoes induced-fit bending to match the domain curvature, while simultaneously the dimer domain also

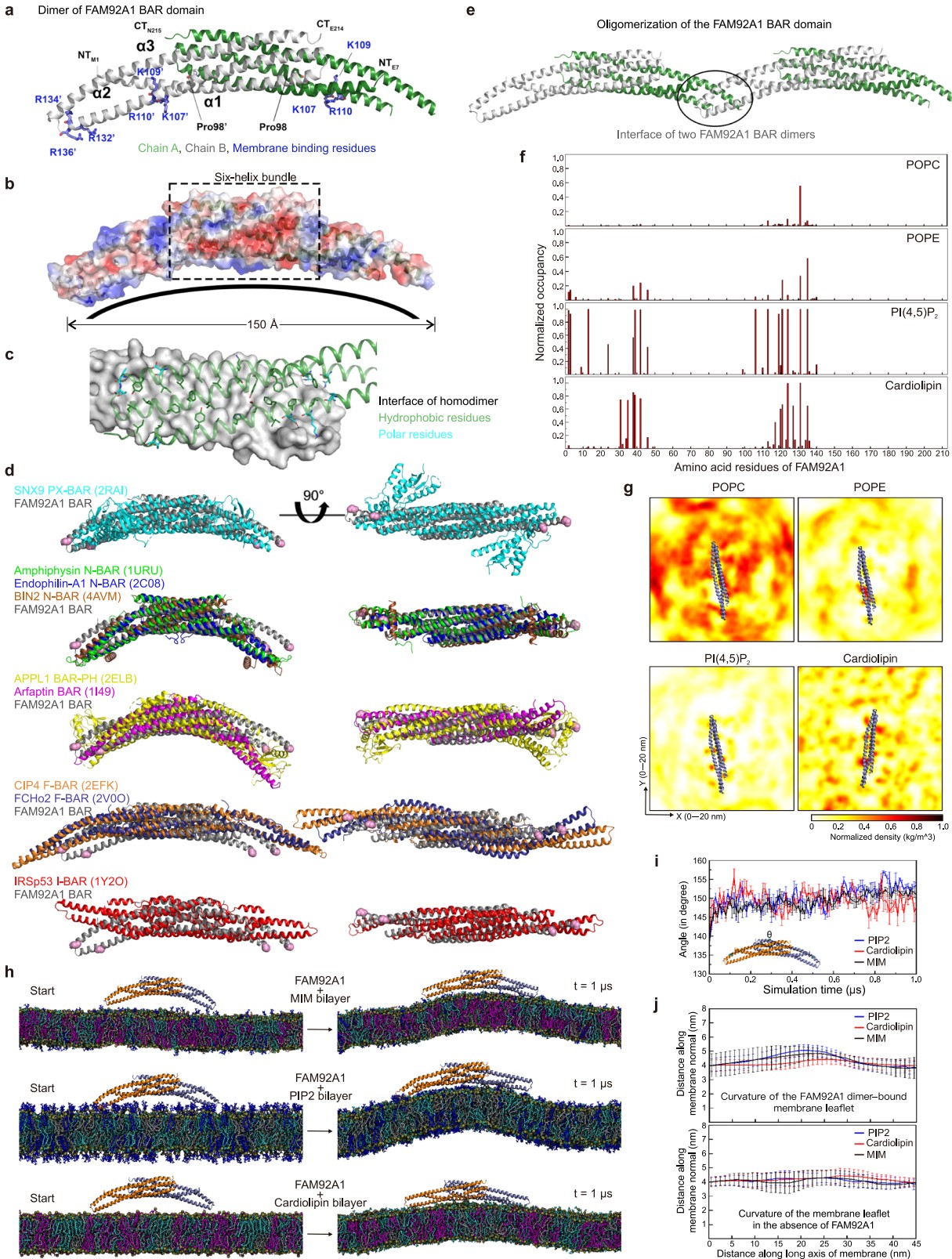

undergoes a small conformational change to facilitate the interaction of positively charged amino acids from the α1 and α2 helices with the membrane.

To investigate the extent to which each of the negatively charged phospholipids contributes towards BAR domain-induced membrane curvature, we carried out simulations of the FAM92A1 BAR domain dimer with PIP2 and cardiolipin bilayers (Supplementary Tables 2 and 3). As with the MIM bilayer, the FAM92A1 BAR domain dimer bound to the membrane and underwent a conformational change while simultaneously inducing membrane bending to fit the resultant dimer domain curvature (Fig. 8h, i). Interestingly, we observed that the curvature induced by the BAR domain dimer in a PIP2 bilayer was more pronounced than that in the cardiolipin bilayer (Fig. 8h, i). This suggests that the increased interaction of FAM92A1

**Fig. 8 | Structure of FAM92A1 BAR domain and molecular simulation of FAM92A1 membrane remodeling mechanism. a** Crystal structure of human FAM92A1 BAR domain (residues 1–219). The three α-helices in chain B were labeled with α1, α2, and α3. NT, N-terminus, CT, C-terminus. **b** The electrostatic potential surface of FAM92A1 BAR domain. The membrane interacting concave surface is indicated with a curve line, box with dashed lines encloses the central 6-helix bundle of the dimer. **c** The structure of FAM92A1 BAR homodimer interface. Hydrophobic residues and polar residues involved in hydrogen-bonded interactions are shown in green and cyan, respectively. **d** Superimposition of the FAM92A1 BAR domain with the existing structures of the BAR domain subfamilies. **e** The hypothesized oligomerization mode and interface at the distal arm tips of two adjacent FAM92A1 dimers, interface indicated with a black sphere. **f** Normalized occupancy of FAM92A1 amino acid residues with membrane phospholipids. **g** 2D density plots showing the distribution of lipids around the FAM92A1 BAR domain

monomer. The values have been averaged over the last 100 ns and across the replicate simulations. **h** Schematic representation of FAM92A1 BAR domain dimer on MIM, PI(4,5)$P_2$, and cardiolipin bilayer, respectively, at the beginning (start time, $t = 0$) and end ($t = 1\,\mu s$) of the simulation. The phospholipid head groups are shown as teal spheres. POPC, cyan; POPE, gray; POPS, pink; PI(4,5)$P_2$, blue; cardiolipin, magenta. **i** Line plot showing the evolution of FAM92A1 BAR domain dimer concave conformation during the simulation period. Inset, residues constituting the angle chosen to estimate the change in FAM92A1 BAR domain dimer concave conformation. **j** Line plot showing the curvature profile of membrane leaflet with (upper) or without (lower) FAM92A1 bound. The position of phosphate atoms of the membrane phospholipids has been binned at intervals of 1 nm along the long axis (x-axis) of the membrane. For (**i**) and (**j**) data represent mean ± SD of three biologically independent simulations. Source data provided as a Source Data file.

with PI(4,5)$P_2$ facilitates a stable and larger induced positive membrane curvature.

Taken together, the BAR domain of FAM92A1 has the capability to form dimers and possibly oligomers. The positively charged residues situated on the concave surface of the FAM92A1 BAR domain dimer allow FAM92A1 to bind to the negatively charged head groups of lipids within the cell membranes. These interactions induce lipid clustering in the membrane that facilitates FAM92A1's role in membrane shaping and bending, contributing to essential cellular processes linked to the formation and remodeling of membrane curvature such as mitochondrial inner membrane morphogenesis and synaptic membrane endocytosis.

## Discussion

FAM92A1 is a BAR domain protein that functions in various cellular processes, including mitochondrial crista ultrastructure and ciliogenesis[5–7]. Although mutations or deletions in the *FAM92A* gene have been linked to neurological disorders and polydactyly in humans[8,11], the precise neurobiological mechanism remains largely unknown. In this study, we show that FAM92A1 is highly expressed in the brain, and its knockout in mice leads to alterations in brain morphology, age-related memory decline, and cognitive deficits. Additionally, the absence of FAM92A1 results in morphological abnormalities in neurons, hippocampal neuron degeneration, and impaired synaptic function, indicating its involvement in brain development, neuronal health, synaptic function, and cognitive processes. Furthermore, we have elucidated the molecular mechanism by which FAM92A1 exerts its effects and its role in these processes. Our findings reveal that FAM92A1 deficiency causes aberrant membrane remodeling and endocytic defects in neurons. To understand the molecular basis of how FAM92A1 regulates neuron morphology and function, we determined the crystal structure of the FAM92A1 BAR domain. Integrating structural insights and molecular simulations, we gained a comprehensive understanding of how FAM92A1 interacts with membranes, inducing lipid clustering and subsequent generation of membrane curvature. In summary, our study utilizing FAM92A1-deficient mice uncovers the role of FAM92A1 in hippocampal synaptic plasticity and neuronal complexity. Moreover, it elucidates the molecular mechanisms underlying its function in neurons.

Additionally, besides its role in membrane remodeling through binding to the lipid membrane, the FAM92A1 BAR domain can interacts with Chibby1, positioned on the convex side of FAM92A1 BAR dimer, inducing deformed membrane-like structures that is engaged in ciliogenesis[5,23]. In the brain, FAM92A1 is expressed in multiple brain regions, including ependymal cells in the ventricles (Fig. 1b). Loss of FAM92A1 might impair ependymal ciliogenesis, leading to reduced dynamics of cerebrospinal fluid and consequently increased ventricle area, and in severe cases, hydrocephalus[51]. If the brain swelling in the FAM92A1-deficient mouse only resulted from the ventricular dilatation, the surrounding brain area adjacent to the enlarged ventricles

would experience compression. Morphometric analysis indeed revealed a decreased volume outside of LVs. However, VBM analysis revealed a reduction in gray matter within these regions, rather than the expected increase in gray matter within the compressed brain area. Subsequent neuronal histological and morphometric analysis in the hippocampal region further revealed decreased neuronal complexity and spine density, altered spine morphology, neuronal degeneration, and neuronal loss. These findings collectively suggest that brain swelling in FAM92A1-deficient mice is a combined result of both ventricular dilatation and neuronal degeneration.

FAM92A1-deficient mice exhibited reduced gray matter density in both the medial and lateral entorhinal areas, which are integral components of the brain's memory and spatial navigation systems[52]. In both human and rodent brains, grid cells in the medial entorhinal cortex and place cells in the hippocampus form a well-known entorhinal-hippocampal neural circuit for spatial navigation, memory and response to locations in visual space[30,53]. Corresponding to these diminished gray matter levels, reduced spatial navigation and spatial contextual memory were observed not only in adult mice (5–6 months old) but also in younger counterparts (1–2 months old) with FAM92A1 deficiency. Generally, the entorhinal area serves as a conduit for relaying diverse sensory information from the outer cortex of the brain to the hippocampus, a region responsible for memory consolidation and storage[54]. Considering those spatial memory deficits and the observed vacuoles in the hippocampus of FAM92A1-deficient mice, we further conducted a VBM analysis of the hippocampus. The results revealed that a reduced gray matter density in hippocampal subregions was exacerbated with the loss of FAM92A1. Moreover, the importance of FAM92A1 in the hippocampus may be further verified by the observed neuronal degeneration and loss in this region. The lower ability to detect the reduced gray matter density in the hippocampus might be attributed to the lower magnetic field strength of the MRI instrument and the absence of a contrast medium to enhance contrast during MRI scans. Hence, we investigated the role of FAM92A1 in the CNS by investigating the potential malformation in the hippocampus.

Interestingly, FAM92A1 knockout mice exhibited deficits in contextual fear memory but not in cued (sound) fear memory during the fear conditioning test. This discrepancy suggests a potential dysregulation of the entorhinal-hippocampal circuitry, known to be one of the earliest networks affected in both AD patients and AD animal models[55,56]. The reduced expression of FAM92A1 in both aged mice and AD mice suggests a potential risk of neurological disorders associated with the loss of FAM92A1. Notably, a child with a microdeletion spanning *FAM92A1* gene loci shows the development of autism and microcephaly[11]. Our bioinformatic analysis of DEGs implicated in disease revealed a potential association of FAM92A1 depletion with a clustering of neurological disorders, such as dementia and AD. Moreover, the behavioral deficits observed in adult FAM92A1-deficient mice rather than in their younger counterparts raise the possibility that neurological disorders resulting from FAM92A1 deficiency may

become more pronounced with age. These results collectively imply that the loss of FAM92A1 expression heightens the susceptibility to neurological disorders. However, the exact neurological disease associated with FAM92A1 requires further investigation. Further exploration of this connection holds the promise of unveiling intriguing insights into the specific roles of FAM92A1 in neurological function and its potential involvement in aging-related neurological disorders.

The intricate membrane systems within neurons play a pivotal role in numerous crucial functions, such as cell signaling, neurotransmission, synaptic plasticity, and the preservation of cellular integrity. Dysregulation or impairment of these membrane-associated processes can significantly contribute to the pathogenesis and progression of neurodegenerative diseases. The generation of membrane curvature typically involves the coordinated action of various proteins sequentially recruited to active sites on the membrane[57]. BAR domain proteins are recognized for their critical roles in diverse cellular processes involving membrane sculpting[58,59]. These processes encompass endocytosis, synaptic vesicle fission, secretory vesicle fusion, and the regulation of the actin cytoskeleton[1–3,60]. The inactivation or loss of BAR domain proteins in cells and animals often leads to severe phenotypes stemming from disruptions in membrane dynamics[61–64]. Correspondingly, our study demonstrated that in vivo FAM92A1 knockout resulted in abnormalities in versatile membrane morphologies, including mitochondrial crista, myelin sheath, and synapses. These aberrations consequently led to neuron degeneration in FAM92A1 knockout mice, contributing as one of the main risk factors for reduced rGMV[65,66].

The crystal structure of the FAM92A1 BAR domain reveals a curvature closely resembling that of classical BAR domains, which typically dimerize into a banana-shaped structure. The concave surface of the FAM92A1 BAR domain generates positive membrane curvature. Similar to other BAR domain proteins[59], the dimerization of the FAM92A1 BAR domain is required for membrane sculpting, as mutations in residues at the dimer interface impaired its ability to sculpt membranes, although it still retained its membrane-binding capability. Compared to other BAR domain proteins, FAM92A1 induces membrane curvature regulation in both mitochondria and the plasma membrane. The mitochondrial resident FAM92A1 predominantly localizes to the mitochondrial matrix in close proximity to the inner membrane, clustering mainly along the cristae, with some localization at the base and tip of the cristae[7]. These findings, combined with molecular simulation data, suggest that FAM92A1 interacts with the mitochondrial inner membrane to induce positive curvature, initiating membrane invaginations to form cristae, and contributes to maintaining the cristae structure by binding to the inner membrane. For plasma membrane-localized FAM92A1, co-localization with proteins involved in both clathrin-mediated endocytosis (CME) and clathrin-independent endocytosis (CIE) suggests the involvement of FAM92A1 in both endocytic pathways. This parallels another BAR domain protein, endophilinA2, which is also implicated in membrane remodeling during both CME and CIE[45]. In addition to the BAR domain, many BAR proteins pose an additional conserved domain, such as the src-homology 3 (SH3) domain, which mediates the interaction with proteins having a proline-rich domain (PRD)[59]. However, FAM92A1 only contains a BAR domain at the N-terminus and an unstructured C-terminus. Most likely, FAM92A1 interacts with negatively charged lipids to generate positive membrane curvature during the endocytosis process, rather than associating with other proteins such as N-WASP through an SH3 domain, to coordinate with the actin cytoskeleton during the endocytic process[67].

Loss of FAM92A1 not only resulted in decreased endocytosis but also impaired mitochondrial inner membrane architecture and energy production. Hence, the depletion of FAM92A1 elicits intriguing effects on various facets of neuronal structure and function. It impacts the density, morphology, and head volume of dendritic spines. Moreover, FAM92A1 deficiency leads to alterations in synaptic architecture and the morphology of synaptic vesicles. Neuronal activity, including neurotransmitter release and synaptic membrane retrieval at synapses, relies heavily on energy[68,69]. Depletion of FAM92A1 impaired the FAM92A1-involved step of synaptic vesicle membrane retrieval and inhibited energy-dependent endocytic processes of the synaptic membrane. Compared to FAM92A1 homozygous mice, both mitochondrial architecture and function were less affected in FAM92A1 heterozygous mice, indicating less suffering from the energy shortage. Consequently, in contrast to FAM92A1 homozygotes, the endocytic process of the synaptic membrane might occur more frequently in FAM92A1 heterozygotes. As a result, more SVs might exhibit enlarged areas due to defects in membrane curvature generation during the retrieval of the synaptic membrane. Therefore, we speculate that the aberrant structure of SVs and decreased synaptic activity result from the combined effects of impaired mitochondrial energy provision and the endocytic process of the synaptic membrane.

The neck of an endocytic pit is a critical site where vesicle scission occurs, leading to the detachment of the newly formed vesicle from the plasma membrane. Alterations or reductions in neck size can significantly influence the efficiency and dynamics of this scission process. FAM92A1 binds to membranes due to its intrinsic curved shape within the BAR domain and its clustering with negatively charged phospholipids like PI(4,5)P$_2$ on one leaflet of the lipid bilayer. The asymmetric enrichment of PI(4,5)P$_2$ creates a disparity between the two lipid leaflets, thereby inducing spontaneous membrane curvature. This curvature-inducing effect is further amplified by the oligomerization of FAM92A1 within the membrane and cooperative interplay with other proteins at the fission site. These combined actions synergistically promote high membrane curvature, culminating in the constriction of the membrane at the specific site of endocytosis. The interaction-induced membrane heterogeneity orchestrated by FAM92A1 may facilitate the specific recruitment and activation of other BAR-domain-containing proteins such as SNX9 to the endocytic site for further membrane constriction. This orchestrated action mediates the membrane morphological transitions from U-shaped to Ω-shaped vesicle pits, which serve as the substrate for membrane fission by dynamin. Importantly, FAM92A1 deficiency prominently increases the frequency of U-shaped vesicle intermediates while reducing the occurrence of Ω-shaped constricted vesicle intermediates. This observation underscores the potential role of FAM92A1 in membrane constriction during the biogenesis of synaptic vesicles.

In summary, our research presents evidence positioning FAM92A1 as a fundamental component within the neuronal machinery, playing a substantial role in maintaining optimal brain function. Specifically, FAM92A1 significantly contributes to the preservation of neuronal architecture and synaptic plasticity through its role in membrane scaffolding at synapses. These findings underscore FAM92A1's indispensable role in brain development, synaptic function, and cognitive processes. Its active participation in membrane remodeling through the BAR domain, coupled with its engagement in endocytosis and the biogenesis of synaptic vesicles, reinforces the fundamental significance of FAM92A1 in the brain.

## Methods
### Animals
For the generation of FAM92A1 knockout mice, the FAM92A1 heterozygotes (FAM92A1$^{+/−}$) and homozygotes (FAM92A1$^{−/−}$) were purchased from GemPharmatech (Nanjing, China). In brief, CRISPR/Cas9-mediated gene editing was used to delete the ORF of mouse *Fam92a1*. Two target sequences of sgRNA (upstream sgRNA, 5′-GGTTATCATCTAC CAATGTC-3′; downstream sgRNA, 5′-TAGACTTCCAGCCGCAGTGG-3′) and Cas9 mRNA were transcribed in vitro and injected into the cytoplasm of fertilized eggs of C57BL/6 genetic background. Injected eggs

were then transferred into the oviductal ampulla of pseudopregnant C57BL/6 JGpt females. Pups were born and genotyped by PCR with the following two pairs of primer: F1, 5′-AACAGGAATCTTGAGAGAGGTA TAGGG-3′, R1, 5′-TCATAACCTCCATTAAGTGCCTTC A-3′; F2, 5′-GGAATCCTCACTGGAGAAACAGG-3′, R2, 5′-ATAC CATGTGCCTTGC TATGAGCT-3′. The homozygous (FAM92A1$^{-/-}$) and wild-type (FAM92A1$^{+/+}$) mice were generated from heterozygous (FAM92A1$^{+/-}$) breeding. Offspring were genotyped by genomic PCR. By identifying the genotype of all viable offspring in an individual litter, the number of each genotype (FAM92A1$^{+/+}$, FAM92A1$^{+/-}$, and FAM92A1$^{-/-}$) was recorded. The frequency of each genotype was calculated by dividing the total number of each genotype by the total number of offspring in a single litter. Following Mendelian laws of inheritance, the observed frequencies of FAM92A1$^{+/+}$, FAM92A1$^{+/-}$, and FAM92A1$^{-/-}$ offspring were compared to their theoretical probabilities. For Alzheimer's disease model mice, 18-month-old male APPswe/PS1$^{\Delta E9}$ (abbreviated as APP/PS1 mice) transgenic mice and age-matched wild-type mice were purchased from Beijing HFK Bioscience Co., Ltd. (Beijing, China; certification number SCXK 2014-0004). For FAM92A1 knockdown experiments in vivo, adult C57BL/6J mice (6–8 weeks old, 20–25 g) were purchased from GemPharmatech Co., Ltd. (Nanjing, China). All mice were housed in cages with 4–5 mice per cage in a controlled environment with a 12 h light/12 h dark cycle (lights on at 7:00 a.m. and off at 7:00 p.m.), a constant ambient temperature of $23 \pm 3\,°C$, and humidity of $55 \pm 5\%$. The animal experiments were performed in strict accordance with the People's Republic of China legislation regarding the use and care of laboratory animals. All procedures used in this study were approved by the Institutional Animal Care and Use Committee of Sichuan University (permit number: 20240428001). All the efforts were made to minimize the suffering of the mice.

## Analysis of mouse *Fam92a1* expression

The expression pattern of *Mus musculus Fam92a1* (Gene ID: 68099, NM_026588.2) was explored and visualized by Brain Explorer 2 using the ABA ISH database (Fam92a - RP_040831_01_F02). The *Fam92a1*-highly expressed brain areas were separately presented. Intensity ranged between 0 and >255, and density ranged between 0.015 and >0.1.

## Cell culture and transfection

Primary hippocampal neurons were dissociated from mouse embryos (E16 to E17). In brief, the whole hippocampus was dissected and digested with 20 U/mL papain for 15 min at 37 °C. After being treated with 2000 U/mL DNase I, the hippocampus was triturated by a glass pipette to make a single-cell suspension. The dissociated hippocampal cells were seeded at $5 \times 10^5$ cells per well (6-well plate) in the neurobasal medium supplemented with 2% B27 supplement (Cat# 17504044 Gibco), 0.5 mg/mL primocin, and 0.5 mM glutamine. Neurons were maintained in a humidified 5% $CO_2$ atmosphere at 37 °C, and half of the medium was changed every 3 days.

Primary fibroblasts were isolated from the ear of FAM92A1$^{+/+}$, FAM92A1$^{+/-}$, and FAM92A1$^{-/-}$ adult mice according to previously reported procedures[70]. Briefly, the cut ears were placed in 70% ethanol for 5 min, dried in a laminar flow cabinet, and dropped into small pieces. Digestion was performed in 1640 medium (Cat# 61870036, Gibco) supplemented with 10% FBS, 100 U/mL penicillin, 100 μg/mL streptomycin, 0.25% collagenase (Cat# 9001-12-1, Sigma), and 0.125% pronase (Sigma), pH 7.4 at 37 °C for 90 min with shaking at 200 rpm. After being filtered through a 70 μm cell strainer (Biosharp), cells were collected by centrifugation at $580 \times g$ for 7 min. Cells were cultured in 1640 medium supplemented with 10% FBS, 100 U/mL penicillin, 100 μg/mL streptomycin, and 0.25 μg/mL amphotericin B. Upon reaching confluency, the fibroblasts were dissociated with a trypsin-EDTA solution and passaged at a 1:3 ratio. The fibroblasts could be passaged for up to six passages.

The mouse hippocampus-derived neuronal cell line HT22 cells were obtained from the Cell Bank of Chinese Academy of Science (Serial# GNM47, Shanghai, China). HT22 cells were cultured in DMEM medium (Cat# 11965092, Gibco) supplemented with 10% FBS, 1% Penicillin-Streptomycin-Glutamine (100×) (Cat# 10378016, Gibco) and incubated in a humidified 5% $CO_2$ atmosphere at 37 °C. The efficiency of siRNA sequences targeting mouse FAM92A1 (Cat# 1027416, QIAGEN) was assessed in HT22 cells using Lipofectamine™ RNAiMAX (Cat# 13778150, ThermoFisher) according to the manufacturer's instructions. The silencing efficacy was determined by detecting the FAM92A1 protein levels 72 h post-transfection.

For transient transfection in the primary hippocampal neuron, two FAM92A1 siRNA oligos that mostly depleted the FAM92A1 protein were chosen to construct shRNA. Briefly, siRNA sequences were followed by the reverse complement, separated by a linker containing an *XhoI* site. The acquired shRNA sequences were subsequently cloned into the pLKO.1-EGFP-puro vector. The shRNA sequences targeting mouse *Fam92a1* were listed as follows: FAM92A1 shRNA 1#, 5′-TGGGACCATTGTAAAGATGAA-3′, FAM92 A1 shRNA 4#, 5′-AAGGACA TAAAGAACATATTA-3′. Transient transfection of cultured neurons was performed as previously reported[71]. In brief, the neurobasal medium was separately mixed with Lipofectamine 2000 (Cat# 11618019, Invitrogen) and FAM92A1 shRNA. After 5 min of incubation at RT, the two mixtures were mixed, and Magnetofection CombiMag (Cat# CM20200, OZ Biosciences) was added. The mixture was incubated for 20 min at RT and was subsequently added dropwise to the neuronal culture. The culture dish was then placed on a magnetic plate and incubated for 30 min in a humidified 5% $CO_2$ atmosphere at 37 °C.

## Mice preparation for magnetic resonance imaging (MRI)

For mice perfused with MRI contrast media, mice were anesthetized with sodium pentobarbital (80 mg/kg, *i.p.*) and intracardially perfused with 30 mL of 0.1 M PBS containing 10 U/mL heparin (Sigma) and 2 mM (ProHance, Bracco Diagnostics, Inc.), followed by 30 mL of 4% paraformaldehyde (PFA) containing 2 mM ProHance. After perfusion, mice were decapitated, and the brain with the skull was dissected as described[72,73]. The brain was then immersed in 4% PFA + 2 mM ProHance at 4 °C overnight and subsequently transferred to 0.1 M PBS containing 2 mM ProHance and 0.02% sodium azide. For mice without perfusion, mice were directly fully anesthetized with a 3% isoflurane/oxygen mixture before the MRI scanning.

## MRI acquisition

MRI scans of mice without perfusion were performed on a small animal 7 T magnetic resonance imaging system (BioSpec 70/30USR, Bruker, Ettlingen, Germany). Following anesthesia, each mouse was placed on the animal carrier with a fixation device supplied with gases. After careful fixation, the breathing rate and temperature of mice were monitored throughout the scan. The heating and isoflurane concentrations were adjusted to maintain the core body temperature and breathing rate of mice at 37 °C and 90–100 per minute, respectively. To adjust the mouse position and perform a two-step shim procedure (first global shim, followed by local shim on the brain volume), initial sequences (Localizer, Localizer multislice, and Calcshim) were acquired. A high-resolution rapid acquisition with relaxation enhancement (Turbo-RARE) scan was used for anatomical, T2-weighted MRI (T2W-MRI) with the following parameters as previously reported[74]: matrix size, 256 × 256; slices, 28; slice thickness, 0.4 mm (no gap); field of view (FOV), $17.5 \times 17.5$ mm$^2$; repetition time (TR), 5500 ms; echo time, 32.5 ms; echo spacing, 10.833 ms; rare factor, 8; flip angle, 90; refocusing angle, 180.

MRI scans of perfused mice were performed on a small animal 7 T (MRINOVA 7.0 T/16, Time Medical, China). Each perfused brain with skull was carefully fixed on the animal carrier. A referred T2-weighted 3D fast spin echo (FSE) sequence was used with

minor modification[72]: TR = 2000 ms; ESP = 42 ms; ETL = 8; effective TE = 84 ms; number of averages = 1; FOV = 25 mm × 20 mm × 20 mm; and matrix size = 384 × 192 × 192.

## MRI data processing

For voxel-based morphometry analysis, the MRI data were processed as previously reported with minor modification[75]. The T2-weighted imaging was performed using the statistical parametric mapping toolbox and software (SPM12, Wellcome Trust Centre for Neuroimaging, London, UK) in MATLAB 2013b as follows: (1) Pre-processing of T2-weighted images: High precise whole-brain masks were created using PCNN3D to segment non-brain tissue for in vivo scans. The whole brain tissues were then extracted using an image calculator in the SPM12 toolbox by multiplying the mouse's raw scans by their brain mask. Additionally, ITK-SNAP was applied to decorate a flawed mask. (2) Origin setting and voxel size resizing: Each skull-stripped T2-weighted image was multiplied by 10 before the subsequent analysis. And the resized T2-weighted images were manually reoriented to the origin. (3) Segmentation and spatial normalization: through using a unified segmentation approach, the resized and reoriented T2-weighted images were segmented into tissue probability maps of the gray matter, white matter, and cerebrospinal fluid and normalize to standard space. To add the transformation information of regional or global volume generated by nonlinear transformations (such as warp and deform) in image segmentation, the signal intensities of the normalized images were modulated using the Jacobian determinant. (4) Smooth: The generated images were then smoothed with a 6-mm full-width-at-half-maximum Gaussian kernel for analysis. (5) Visualization of statistical analysis and results: the smoothed images were imported into the statistical analyzer in SPM12, contrast T-maps were generated and compared voxel by voxel with one-way analysis of variance $F$-test with a gray matter mask settled at an absolute threshold mask of 0.2 (with relative total brain volumes entered as a covariate). For multiple comparisons, the stringent family wise error (FWE) correction was applied with a statistical threshold of 0.05. The results of the voxel-based morphometry (VBM) analysis were shown on the original scales. The significantly altered clusters were saved as masks, and their corresponding regional gray matter volume (rGMV) was extracted.

For deformation-based morphometry analysis, the raw diffusion data were first converted to NIFTI format using the dcm2niix tool included in the MRIcroGL package (nitrc.org/projects/mricrogl/) and preprocessed before further analysis. After preprocessing, all MRI scans of individual subjects were orderly linearly and non-linearly registered using ANTS[76]. In linearly registrations, by adjusting orientation and overall width, length, and height, all brains were transformed to the anatomical space of a single template mouse brain from Allen institute. Subsequent non-linearly registrations introduced local deformations to each single brain to match it to the common registered brain. The local Jacobian determinants were then applied to characterize volume difference at each voxel (3D pixel) between two groups. Relative volume measurements were applied to compare total brain volume (TBV) differences between FAM92A1[+/+] and FAM92A1[+/−] groups. Comparisons between two groups were conducted using a student's $t$-test (two-tailed) using the statistical analyzer in SPM12. Voxel $p < 0.05$, cluster $p < 0.05$ and FWE $< 0.05$ were set as the cut-off for statistical significance.

For quantifying the ventricle area, T2-weighted MRI scans were processed using Fiji as described[27]. Images containing the lateral, third, cerebral, and fourth ventricles were generated by creating maximum intensity projections from slices containing each respective ventricle. Manual segmentation of ventricular areas was performed by two independent raters. Through drawing a region of interest (ROI) in the bright areas of the ventricles on the projection image, the area of the ventricles was quantified.

## Immunofluorescence

For cells, primary hippocampal neurons were fixed with 4% PFA / 4% sucrose (v/v) in PBS at RT for 15 min, and primary fibroblasts and HT22 cells were fixed with 4% PFA. After being permeabilized with 0.1% Triton X-100, cells were blocked with 1% BSA in PBS. Then, cells were incubated with primary antibodies overnight at 4 °C. Following three times washing with PBS, cells were incubated with Alexa-conjugated secondary antibody at RT for 1 h (protected from light). Lastly, cells were counterstained with DAPI and mounted with ProLong™ gold antifade mountant (Cat# P36934, ThermoFisher).

For brain tissues, the fixed whole brains were cryoprotected overnight in 30% sucrose in PBS and sliced on a microtome at 10 μm. After being permeabilized, slices were blocked with a blocking buffer (5% BSA and 0.2% Triton X-100 in PBS, pH 7.4) to block nonspecific binding sites. Then, slices were incubated with primary antibodies diluted with blocking buffer overnight at 4 °C. Following three times washing with PBST, slices were incubated with Alexa-conjugated secondary antibody at room temperature for 1 h (protected from light). Lastly, slices were counterstained with DAPI and mounted with ProLong™ gold antifade mountant. Images were acquired on a laser scanning confocal microscope (Leica SP8 TCS) with built-in LAS X software (Leica Biosystem).

## TIRF microscopy

TIRF microscopy was conducted to visualize the localization of FAM92A1 on the plasma membrane using an N-STORM microscope (ECLIPSE Ti-E, Nikon) equipped with an oil immersion TIRF objective lens (Apo TIRF 100× N.A. 1.49) and a high-speed complementary metal-oxide semiconductor (CMOS) camera (ORCA-Flash4.0 V3, Hamamatsu). Fixed cells were imaged with excitation laser light of 488 nm, 561 nm, or 640 nm wavelength. The TIRF angle was adjusted in live mode until most of the cytoplasmic background signal disappeared, leaving only the target plasma membrane visible (usually with a penetration depth of ~150 nm). Images were captured in a single focal plane using NIS-Elements AR software (version 5.21.00, Nikon Instruments). Images were analyzed in Fiji.

## Immuno-electron microscopy (Immuno-EM)

Immuno-EM was performed to visualize the endogenous localization of FAM92A1 as described[77]. HT22 cells were pelleted through centrifugation at 600 × g for 5 min and immediately fixed with 4% PFA supplemented with 0.5% glutaraldehyde (CAS#111-30-8, SPI). The cell pellet was suspended in 12% weight/volume gelatin (Cat# G7041, Sigma) in PBS at 37 °C for 15 min and subsequently placed on ice for 20 min to solidify the gelatin. After solidification, the cell pellet was cut into small blocks and placed in 2.3 M sucrose overnight at 4 °C. The specimen blocks were transferred to aluminum specimen holders (no. 16701950, Leica Microsystems, Germany) and frozen by immersing them in liquid nitrogen. Ultrathin sections were then cut from the frozen specimen blocks using a Leica EM FC7 ultramicrotome (Leica Microsystems, Germany). The sections were blocked with 1% BSA-c (Aurion, The Netherlands) for 30 min and immunolabeled using the anti-FAM92A1 antibody in 1:50 dilution for 2 h. After washing six times with PBS, the sections were immunolabeled by a gold-conjugated (10 nm mean diameter) goat anti-rabbit IgG antibody (GAR-25109, Aurion, The Netherlands) at a 1:60 dilution for 1 h at RT. The immunolabeled samples were placed on drops of uranyl acetate/methyl cellulose (1:9) for 5 min. After drying, the samples were visualized using transmission electron microscopy (FEI, Tacnai, G2 spirit, Hillsboro, Oregon) operating at 100 kV, and images were captured using a CCD camera (MoradaG3; EMSIS) with RADIUS software at RT.

Primary antibodies used for immunofluorescence and Immuno-EM are listed as follows: rabbit anti-FAM92A1 (1:100, Cat# HPA034760, Sigma), rabbit anti-FAM92A1 (1:100, Cat# 24803-1-AP, Proteintech), rabbit anti-TOM20 (1:1000, Cat# 11802-1-AP, Proteintech), mouse anti-

VDAC (1:50, Cat# ab14734, Abcam), mouse anti-Clathrin (1:50, Cat# 610449, BD Bioscience), mouse anti-Caveolin (1:50, Cat# 610406, BD Bioscience), mouse anti-PSD95 (1:500, Cat# MA1-046, Invitrogen), mouse anti-SV2 (1:250, Cat# SV2, DSHB), mouse anti-MAP2 (1:400, Cat# 13-1500, Invitrogen), rabbit anti-vGlut1 (1:1000, Cat# 55491-1-AP, Proteintech), rabbit anti-Vimentin (1:500, Cat# 10366-1-AP, Proteintech). Actin was visualized with Alexa Fluor 647 conjugated to phalloidin (1:200; Cat# A22287, Invitrogen).

## Transmission electron microscopy (TEM)

Mice were deeply anesthetized with an overdose of isofluorane and transcardially perfused with PBS followed by ice-cold 2.5% glutaraldehyde in 4% PFA. The brains were quickly removed and cut into a 1 mm slice. The hippocampus was dissected into $1 \times 1$ mm sizes with the upper left corner cut and post-fixed with 2.5% glutaraldehyde for 3 days at 4 °C. The samples were dehydrated in a gradient of ethanol and propylene oxide after being rinsed in double-distilled water. Next, samples were individually impregnated in 100% epoxy resin. Afterward, the embedded samples were sliced into 80 nm sections, which were further mounted on 200 mesh Metaxaform Copper Rhodium grids. The mounted samples were stained with 3% uranyl acetate and bismuth subnitrate to increase contrast. The final samples were observed and images were acquired on a JEM-1400PLUS (JEOL Ltd, Japan) with an Orius SC 1000B bottom-mounted charge-coupled device camera (Gatan). Images of all samples were taken at accelerating voltages of 80 kV and at magnifications ranging from 8000 × to 100,000×. The pixel value of all images was 4476 × 3280 pixels (length × width).

## Focused ion beam (FIB) and scanning electron microscopy (SEM) (FIB-SEM)

The FIB-SEM samples were processed as described with slight modifications[78]. In brief, mice were deeply anesthetized with an overdose of isoflurane and transcardially perfused with PBS followed by ice-cold 2% PFA in PBS. The brains were quickly removed and cut into coronal sections with 300 μm thickness using a vibratome (Leica VT 1200, Leica Biosystems, Buffalo Grove, IL) in ice-cold 0.1 M sodium phosphate buffer (pH 7.4), supplemented with 2 mM CaCl$_2$. The hippocampus area was dissected from the coronal sections, and further fixed with 2% glutaraldehyde (EM-grade, Sigma) and 2% PFA in 0.1 M sodium phosphate buffer, supplemented with 2 mM CaCl$_2$ at 4 °C for 3 h. The samples were then washed with 0.1 M sodium cacodylate buffer (NaCac, pH 7.4) and osmicated with 2% osmium tetroxide (Electron Microscopy Sciences) in 0.1 M NaCac for 1.5 h, followed by 2.5% ferrocyanide in 0.1 M NaCac for another 1.5 h. After washes with water, the samples were immersed with 1% thiocarbohydrazide (Cat# 2231-57-4, Sigma-Aldrich) at 40 °C for 30 min followed by treatments with 1% unbuffered osmium for 1 h, and 1% UA in water at 4 °C overnight. Between the treatments, the samples were washed with ion-exchanged water 5 times for 5 min. For embedding, samples were dehydrated through a graded ethanol series into pure acetone followed by gradual infiltration into Durcupan™ (Sigma) according to the manufacturer's instructions. After polymerization at 60 °C for 48 h, the embedded samples were trimmed and mounted on aluminum pins with conductive glue, prior to coating the exposed hippocampal surface with a thin layer of platinum (Quorum Q150TS, Quorum Technologies, Laughton, UK).

The target volumes in the hippocampus were imaged by a Zeiss FIB-SEM (Crossbeam 550 with Gemini 2 optics, Carl Zeiss Microscopy GmbH, Jena, Germany) using a 2.5 nm pixel size with 5 nm milling depth. The SEM micrographs were acquired with 0.31 nA electron beam, 1.3 or 1.4 keV, dwell time 2 − 3 μs with 1 − 2 times line averaging. Both backscattered and secondary electron signals were collected using Inlens detectors. The new surface for serial FIB-SEM imaging was generated by FIB milling with a 0.7 nA beam current at an acceleration voltage of 30 kV. The datasets were collected and aligned using Zeiss Atlas 5 software with a 3D tomography module. The representative synapses in the electron microscopy data sets generated were manually segmented using MIB software[79], and the vesicles and endosomal structures which were located mostly in the volume covering the depth of 300 μm were included in the models. Visualization of the models was done by Amira software (Visage Imaging Inc., San Diego, CA, USA).

The number and area of synaptic vesicles were quantified using MIB software with the deep learning tool Deep MIB. Automated segmentation was manually verified. From FIB-SEM data with voxel dimensions of $5 \times 5 \times 5$ nm (available at https://www.ebi.ac.uk/empiar/deposition/logout/), every 50th section was selected, resulting in a total of 18 slices for measurement. Synaptic vesicles within synapses showing distinguishable pre- and post-synaptic areas on micrographs were measured. Statistical analysis was conducted using GraphPad Prism (version 9.5.0).

## Cresyl violet (CV) staining

CV staining was performed using the 0.1% cresyl violet solution (Cat# G1036, Servicebio, China) by referring to the previously reported procedure as well as the manufacturer's instructions[80]. Briefly, the obtained brain sections were collected on glass slides, deparaffinized, and stained with 0.1% cresyl violet solution for 5 min and washed with distilled water. After sequential being dehydrated with ascending grades of alcohol and cleared with xylene, the stained brain sections were mounted with DPX mounting medium (Cat# 06522, Sigma) and observed with a digital slide scanner (Pannoramic MIDI, 3DHISTECH Ltd., Hungary).

## Fluoro-jade B (FJB) and Fluoro-jade C (FJC) histofluorescent staining

FJB staining was performed as described[81]. In brief, after dewaxed and hydrated, the brain slides were incubated with 0.06% potassium permanganate solution for 10 min. The slides were then rinsed with distilled water and incubated with fresh 0.0004% FJB solution (Cat# AG310, Millipore) in 0.1% acetic acid overnight at 4 °C. The brain sections were then stained with DAPI at room temperature for 20 min, rinsed with distilled water, fully air-dried, and mounted with DPX. FJC staining is performed using the FJC staining kit (Biosensis, TR-100-FJ, Thebarton, South Australia) according to the manufacturer's instructions. Briefly, after dewaxed and hydrated, the brain slides were incubated with 0.06% potassium permanganate solution for 10 min. The slides were subsequently rinsed with distilled water and incubated with diluted FJC solution (FJC solution: distilled water = 1: 9) for 10 min. The slides were then washed, dried, and mounted with DPX. Images were acquired on a laser scanning confocal microscope (Leica SP8 TCS) with built-in LAS X software (Leica Biosystem).

## Golgi staining and sholl analysis

Golgi staining was performed using the FD Rapid GolgiStain kit (FD Neurotechnologies) according to the manufacturer's instructions. In brief, the mouse brains were immersed in a Golgi mixture solution (A + B, A/B = 1:1) for 14 days at room temperature, followed by 3 days at RT and darkness in the Golgi solution C. Brains were then serially coronally sectioned at 100 μm slices using a Leica VT100S vibrating microtome (Wetzlar, Germany). Slices ranging from rostral-caudal bregma −1.82 mm−−2.06 mm were chosen for carefully mounting on gelatin-coated microscope slides. After drying overnight, the slides were immersed in the staining solution (solution D/E) for 10 min and rinsed the double distilled water. The sections were then dehydrated sections with a graded ethanol series. Lastly, the sections were cleared in xylene and mounted with Eukitt® Quick-hardening mounting medium. Brightfield confocal photomicrographs were acquired using a digital slide scanner (Pannoramic MIDI, 3DHISTECH Ltd., Hungary).

Images were imported into Fiji and processed for Sholl analysis to assess neuronal complexity. Hippocampal granule neurons in the DG region included in the analysis satisfied the following criteria: (a) the presence of soma within the hippocampal DG region; (b) morphologically, by the presence of multiple short dendrites extending outward in a radial pattern as characteristic of hippocampal granule neurons; (c) an isolated cell body with a clear relationship of the primary dendrite to the soma; (d) complete impregnation of the neuron along the entire length of the dendritic arbor without truncated dendrites; and (e) minimal overlap or relative non-overlap with surrounding impregnated cells[82,83]. All included neurons were manually traced using the NeuronJ plugin[84]. An arborization analysis was then performed using the Sholl analysis plugin applying a modified Sholl method for the best polynomial fit. First and subsequent shells were set at a radius of 0.5 μm and intersections at each Sholl radius were determined. Raw data of concentric ring intersections and dendritic length were compiled and analyzed by GraphPad Prism (version 9.5.0). Two-way ANOVA followed by Tukey's post hoc test was used for statistical analysis.

### FM1–43 uptake assay
The cultured primary fibroblasts were labeled with FM1–43 (Synapto-Green, Cat# 70020, Biotium) to track exocytosis, endocytosis, and recycling of secretory granules or vesicles as previously reported procedure[85]. The cultured fibroblasts were incubated with 4 μM FM1–43 in 1640 complete medium for 7 min at 37 °C. The FM1–43 fluorescence images were taken immediately (excitation, 480 nm; emission, 598 nm). The mean fluorescence (MF, a.u.) of a single fibroblast was measured using Fiji.

### Live cell imaging
The primary fibroblasts were replated on glass-bottomed dishes (35 mm, Cat# 627860, Greiner Bio-One). The dish was placed in a heated sample chamber with controlled 5% CO2. After incubating with 4 μM FM1–43 in 1640 complete medium for 7 min, the time-lapse images were acquired with a laser scanning confocal microscope (Leica SP8 TCS) with built-in LAS X software (Leica Biosystem). Analyses of the frames were performed with Fiji.

### In vitro phagocytosis assay
Latex beads with yellow-green fluorescence (Cat# L1030, Sigma) were used for the phagocytosis assay according to the previously reported procedure[86]. In brief, the primary fibroblasts were harvested after 8 h of phagocytosis using the latex beads. After washing three times with PBS, cells were resuspended with PBS and the phagocytic ability of the primary fibroblast was analyzed by a BD Fortessa X20 flow cytometer (BD Biosciences) using the BD FACS Diva™ software version 8.0 (BD Biosciences). The acquired raw data were analyzed using FlowJo software (version 10.8.1).

### Image acquisition and analysis
For spine morphology analyses, z stacks of secondary dendritic stretches were captured using an oil-dipping 63x objective on a laser scanning confocal microscope (Leica SP8 TCS) with built-in LAS X software (Leica Biosystem). Spines were then automatically reconstructed and manually adjusted using the Filament Tracer module of Imaris (Bitplane). Dendritic spines were classified into four classes using the "Imaris Spines Classifier" extension in the Imaris (version 9.5.1, Bitplane, Zurich, Switzerland) software. with the following morphological criteria: (1) stubby spines were identified first as spines with a length < 1; (2) mushroom spines were then identified as spines with a length < 3 and max_width (head) > mean_width (neck) * 2; (3) long thin spines were identified as spines with mean_width (head) ≥ mean_width (neck); (4) filopodia/dendrite spines were the remaining spines. For each dendritic stretch, spine density was calculated as a ratio in relation to the baseline level, which was defined as the spine density measured during the first imaging session. All steps were carried out blindly to the experimental conditions.

For g-ratio analyses, both the axon diameter and the total diameter of the axon plus the myelin sheath were separately measured using Fiji (version 1.53 t). The g-ratio was calculated as the previously reported equation[87]. Linear regression was done to examine the differences in myelin thickness across the range of axon diameters among the experimental groups.

### Mitochondrial isolation and measurement of hippocampal mitochondrial respiration
Mitochondria were isolated from fresh hippocampus according to our previously reported procedure[88]. Briefly, mice were sacrificed by cervical dislocation and the hippocampus was immediately subjected to mitochondrial isolation after being dissected from the brain. Then, the hippocampus was homogenized in an ice-cold mitochondrial isolation buffer (10 mM Tris-MOPS, 1 mM EGTA, and 200 mM sucrose, pH 7.4) using a Teflon potter (Potter S; Braun). After centrifuging twice at 800 × g for 10 min, the resulting supernatant was collected and centrifuged at 7000 × g for 10 min. The pellet was the crude mitochondria that can be used for subsequent assay.

Mitochondrial oxygen consumption rate (OCR) was measured using a Seahorse XFp Extracellular Flux Analyzer (Seahorse Bioscience, Billerica, MA) as previously reported with minor modifications[89]. The crude fresh mitochondria were diluted in MAS1 buffer (220 mM D-Mannitol, 70 mM sucrose, 10 mM of KH$_2$PO$_4$, 5 mM of MgCl$_2$, 2 mM of HEPES, 1 mM of EGTA, and 0.2% (w/v) of fatty acid-free BSA, pH 7.2). The diluent mitochondria were then plated on Seahorse XF Cell Culture Microplates (Cat# 103022-100, Seahorse Bioscience; Agilent Technologies) at a concentration of 5 μg protein per well. After centrifugation at 2000 × g for 15 min, OCR was measured in the degassed cell culture plates using the Mito Stress Test protocol. Mitochondrial complex inhibitor solutions were freshly prepared and loaded into the hydrated Flux Pak as follows: port A: 20 μL adenosine diphosphate (ADP, [stock] = 40 mM); port B: 22 μL oligomycin (Oligo, [stock] = 32 μM); port C: 24 μL fluoro-carbonyl cyanide phenylhydrazone (FCCP, [stock] = 40 μM), and port D: 26 μL rotenone ([stock] = 40 μM) + antimycin A (R + A, [stock] = 20 μM). Data were analyzed using the Seahorse Wave 2.6.3 software (Seahorse Bioscience).

### ATP assay
Hippocampal ATP levels were detected by a luciferin/luciferase bioluminescence assay using an ATP Determination kit (A22066, ThermoFisher) according to the manufacturer's procedure. In brief, mice were sacrificed by cervical dislocation, and the hippocampus was lysed with a commercial Mammalian Cell & Tissue Extraction Kit (Cat# K269–500, BioVision) after being dissected from the brain. Then, the hippocampus was homogenized using a Polytron-type homogenizer, and the homogenate was centrifuged at 16,000 × g for 10 min at 4 °C. 10 μL supernatant was mixed with 90 μL reaction solution and the luminescence was measured by a multimode plate reader (Varioskan Flash, ThermoFisher). ATP content was calculated based on an ATP standard curve.

### MEA recordings and data analysis
To record spontaneous network activity, the Maestro Edge multi-well MEA system (Axion Biosystems Inc., Atlanta, GA, USA) was used. Each well of 24-well MEA plates embedded with an array of 16 gold electrodes (Axion BioSystems) were used to record. The day before plating cells, MEA wells were pre-coated with 0.01% poly-L-lysine (Cat# 25988-63-0, Sigma) at RT for 1 h. Then, primary hippocampal neurons were isolated from mouse embryos (E16 to E17) as described above. After isolation, cell pellets were suspended in a complete neurobasal medium with Matrigel (Cat# 356231, Corning) to a concentration of 8 × 10⁶

neurons/mL, followed by thorough mixing on ice. 10 µL of cell suspension was gently added to the recording electrode area of each well. MEA plate with the seeded neurons was put into the incubator in a humidified 5% $CO_2$ atmosphere at 37 °C for 1 h. The final culture medium was brought to 500 µL/well and MEA plate was maintained in the incubator. Half of the medium was changed every 3 days. At DIV7, 10, and 14, MEA plate was transferred to the recording chamber, which was maintained at 37 °C with 95% O2 and 5% CO2, and allowed to equilibrate for 30 min before recording. The activity of neuronal networks was recorded for 10–30 min using Axion BioSystems' Integrated Studio software (AxIS, version 3.7.2) (Axion BioSystems) with a Butterworth band-pass filter of 10 Hz and 2.5 kHz cutoff frequencies.

Data analysis was performed off-line by using the manufacturer's standalone tool Neural Metric Tool (Axion Biosystems). A spike detection was computed with an adaptive threshold of six standard deviations of the estimated noise for each electrode. The electrodes that detected at least 5 spikes per minute were classified as active electrodes. Bursts were identified using an inter-spike interval (ISI) threshold requiring a minimum number of five spikes with a maximum ISI of 100 ms. Network bursts were identified as bursts of >50 spikes that occurred in >35% of the active electrodes in the well, with a maximum ISI of 100 ms. Synchrony represented a measure of similarity between two spike trains and the synchrony index was estimated by the area under the normalized cross-correlogram for a time window of 20 ms.

## Viral preparation

Recombinant adeno-associated virus 9 (rAAV9)-CaMKIIα-GCaMP6m (BC-0082) and AAV-sparse-NCSP-YFP-2E5 (BC-SL001) were purchased from Braincase Co., Ltd. (Shenzhen, China). For in vivo FAM92A1 knockdown, both FAM92A1 shRNA #1 and FAM92A1 shRNA #4 were individually cloned into a neurotropic AOV062 AAV vector pAAV-hSyn-EGFP-3xFlag (Obio Technology), referred to as sh*Fam92a1* #1 and sh*Fam92a1* #4. One scrambled shRNA (5'-GAAGTCGTGAGAAGTAGAA-3') with theoretically no effect on any gene was cloned into the same vector, referred to as control. Viral packaging and manipulation were conducted following the manufacturer's recommended protocol. All viruses had titers >$10^{12}$ viral particles per ml, and they were aliquoted and kept at −80 °C until use.

## Stereotaxic surgery

For viral microinjection, the procedure was carried out as previously reported[90]. Mice were head-fixed on a stereotaxic device (RWD, Shenzhen, China) after being anesthetized with sodium pentobarbital (80 mg/kg, *i.p.*). The skull was completely exposed and a tiny hole was drilled above the target brain area. A volume of 100–250 nL of virus was injected bilaterally (slice recordings and knockdowns) or unilaterally (photometry) into the hippocampal DG with the following coordinates: (anteroposterior (AP): −2 mm, mediolateral (ML): ±1.4 mm, dorsoventral (DV): − 1.75 mm). The injection coordinates were derived from the Paxios and Franklin Mouse Brain Atlas, second edition. Virus was infused at a uniform speed over 10 min using a syringe (needle size: 22 gauge, RWD, Shenzhen, China) and a micro-infusion pump (Legato 130 Syringe Pump, KdScientific). The needle was kept in the site for a minimum of 10 min to permit diffusion before being slowly withdrawn. The scalp incision was closed using surgical sutures after injections.

For implantation of optical fiber, following rAAV9-CaMKIIα-GCaMP6m viral injection, an optical fiber (core diameter: 200 µm, numerical aperture (NA): 0.37; Newdoon, Shanghai, China) bound to a ceramic ferrule was slowly and unilaterally implanted with its tip targeting viral injection sit. The optical fiber was then secured to the skull with dental cement. After surgery, all mice were positioned on a thermal blanket to facilitate their recovery until they regained consciousness from anesthesia. Each mouse was individually returned to a

home cage and allowed to recover for 3 weeks for complete healing and maximum viral expression.

## Fiber photometry

The signal of $Ca^{2+}$ in neurons was recorded using a multichannel fiber photometry recording system (Thinker Tech, Nanjing, China) as previously described[91,92]. In fiber photometry, the same fiber was utilized for both the excitation and real-time recording of the GCaMP6m fluorescent signal. The laser intensity at the tip of optical fiber was adjusted to -20 µW to minimize bleaching. Fiber photometry of $Ca^{2+}$ signals was carried out by transmitting a 488 nm laser beam through the optical fiber to activate the genetically coded GCaMP6m. The fluorescent signal generated by the excited GCaMP6m was collected by the optical fiber. After being bandpass filtered (MF525-39, Thorlabs), GCaMP6m fluorescent signal was detected by a sensitive photodetector CMOS (complementary metal-oxide semiconductor) (DCC3240M, Thorlabs) camera of the system. A LabVIEW program (Thinker Tech, Nanjing, China) was applied to control the CMOS camera and record $Ca^{2+}$ signals at a sampling frequency of 50 Hz. During recording, mice were placed in a rectangular open field arena (48 cm × 48 cm × 31 cm, length × width × height) and allowed to move freely.

## Photometry data analysis

The GCaMP6m fluorescence signals were analyzed in MATLAB 2016a software (MathWorks, Cambridge, United Kingdom). Changes in fluorescence intensity ($\Delta F/F$) was calculated as $\Delta F/F = (F − F_0)/F_0$, where $F$ was the test fluorescence signal and $F_0$ was the averaged baseline fluorescence signal recorded before stimulation. The value of $\Delta F/F$ was presented as heatmaps and average plots, with the shaded area indicating the standard error of the mean.

## Whole-cell patch-clamp recording

Electrophysiological recordings were conducted using mice aged 8–9 weeks. In brief, mice were deeply anesthetized and transcardially perfused with 95% O2 and 5% CO2 oxygenated ice-cold N-Methyl-D-glucamine (NMDG) cutting solution (93 mM NMDG, 93 mM HCl, 2.5 mM KCl, 1.2 mM $NaH_2PO_4$, 30 mM $NaHCO_3$, 25 mM D-glucose, 20 mM HEPES, 5 mM Na-ascorbate, 2 mM thiourea, 3 mM Na-pyruvate, 10 mM $MgSO_4$, and 0.5 mM $CaCl_2$, pH adjusted to 7.35 with NMDG or HCl). The mouse brain was quickly removed and immersed in an ice-cold NMDG cutting solution. Horizontal hippocampal slices (300 µm thickness) were dissected with a Leica VT1200s vibratome, recovered at 34 °C for 10–13 min in NMDG cutting solution, and then maintained at 25 °C in oxygenated artificial cerebrospinal fluid (ACSF) (126 mM NaCl, 2.5 mM KCl, 2 mM $CaCl_2$, 2 mM $MgCl_2$, 26 mM $NaHCO_3$, 1.25 mM $NaH_2PO_4$, and 10 mM glucose) for 1 h until electrophysiological recordings. Slices were transferred to the recording chamber and superfused with oxygenated ACSF ( - 3 mL/min). Slices were visualized with infrared optics using an upright microscope equipped with a 60 × water-immersion lens (BX51WI, Olympus). granule neurons in the hippocampal dentate gyrus were identified based on their location and morphology.

For whole-cell recording of excitatory postsynaptic currents (EPSCs), pipettes were filled with a solution containing (135 mM K-gluconate, 5 mM KCl, 0.5 mM $CaCl_2$, 10 mM HEPES, 2 mM Mg-ATP, 0.1 mM GTP, and 5 mM EGTA, 300 mOsm, pH adjusted to 7.3 with KOH). To isolate the glutamatergic currents, GABAA receptors were blocked with 10 µM bicuculline (CAT #BML-EA149-0050, Enzo Life Sciences). In addition, 1 µM TTX (Cat# 1078, Tocris Bioscience) was also included while recording miniature EPSCs (mEPSCs). Peak events were detected automatically using an amplitude threshold of 10 pA. The holding potential was −70 mV during mEPSC. All signals were acquired with a MultiClamp 700B amplifier (Molecular Devices), filtered at 1 kHz, and sampled at 5 kHz with a Digidata 1440 A interface

using Clampex 10.2 (Molecular Devices). Data were accepted when series resistance fluctuated within 15% of initial values (25–35 MΩ). Spontaneous and Mini events were analyzed using MiniAnalysis 6.07 software (Synaptosoft). A 5 min recording duration was used for frequency and amplitude analyses.

## Subcellular fractionation of brain tissues

Subcellular fractionation was conducted as described before[93]. HPC from adult mice was homogenized in ice-cold homogenization buffer containing 4 mM Hepes, pH 7.4, 320 mM sucrose, phosphatase/protease inhibitor, and 1 mM PMSF, using a glass-Teflon homogenizer (Potter S; Braun) with ten strokes at 900 r.p.m. Homogenates (H) were centrifuged at $1000 \times g$ for 10 min at 4 °C. The resulting pellet (P1) was the nuclei and large debris. The supernatant (S1, cell fraction without nuclei) was centrifuged at $15,000 \times g$ for 15 min at 4 °C to obtain a further supernatant (S2) and a crude synaptosomal fraction pellet (P2). P2 was subsequently osmotic lysed with a mixture containing 1 vol of homogenization buffer and 9 vol of ice-cold water. The lysate was adjusted to 25 mM Tris and incubated for 30 min at 4 °C. Thereafter, P2 was centrifuged at $17,000 \times g$ for 15 min at 4 °C to obtain pellet a pre- and post-synaptic membrane fraction (LP1) and supernatant (LS1). After determining protein concentration, an equal amount of proteins from each fraction was subjected to western blot analysis.

## Western blots

For mouse tissue, mice were deeply anesthetized with an overdose of isoflurane and transcardially perfused with PBS. Following dissection, the liver, heart, lung, spleen, kidney, and various mouse brain sub-regions were instantly frozen with liquid nitrogen. The tissue proteins were extracted using the total protein extraction kit (Cat# BC3710, Solarbio) following the manufacturer's instructions. For primary fibroblasts, cells on the cultured plates were directly scraped with a cell scraper in 1% n-dodecyl-β-D-maltopyranoside (DDM, Cat# HY-128974, MedChemExpress) lysis buffer supplemented with phosphatase/protease inhibitor and 1 mM PMSF. After determining protein concentration with Bradford reagent (Cat# E211-01, Vazyme), equal amounts of proteins were loaded and fractionated on SDS-PAGE and transferred onto 0.45 or 0.2 μm PVDF membrane (IPVH00010 or ISEQ00010, Millipore), immunoblotted with antibodies, and visualized by Western ECL Substrate (WBKLS0500, Millipore). Band intensity was quantified using Fiji. The used antibodies and the dilutions of antibodies were listed as follows: rabbit anti-FAM92A1 (1:500, Cat# HPA034760, Sigma), rabbit anti-FAM92A1 (1:1000, Cat# 24803-1-AP, Proteintech), rabbit anti-Dynamin 2 (1:4000, Cat# 14605-1-A, Proteintech), rabbit anti-CLCa (1:1000, Cat# 10852-1-AP, Proteintech), rabbit anti-SNX9 (1:8000, Cat# 15721-1-AP, Proteintech), rabbit anti-Synaptotagmin-1 (1:1000, Cat# 14511-1-AP, Proteintech), rabbit anti-vGlut1 (1:500, Cat# 55491-1-AP, Proteintech), mouse anti-PSD95 (1:2000, Cat# MA1-046, Invitrogen), rabbit anti-SDHA (1:1000; Cat# 14865-1-AP; Proteintech), rabbit anti-Synaptophysin (1:20,000, Cat# ab32127, Abcam), rabbit anti-GAPDH (1:2000, Cat# 2118, Cell Signaling Technology), rabbit anti-β-actin(1:50000, Cat# 81115-1-RR, Proteintech), rabbit anti-α-Tubulin(1:2000, Cat# 2125, Cell Signaling Technology).

## RT-qPCR

The fresh tissues were immediately subjected to RNA isolation with the AxyPrepTM Multisource RNA Miniprep kit (Cat# AP-MN-MS-RNA-50, Axygen) according to the manufacturer's instructions. Total mRNA was reverse transcribed using a PrimeScriptTM RT reagent kit with gDNA Eraser (Cat# RR047A, Takara). Quantitative PCR reactions were carried out in QuantStudio 1 Real-Time PCR System (QS-1) (Thermo-Fisher) using PowerUpTM SYBRTM Green Master Mix (Cat# A25742, ThermoFisher). Primer sequences used in this study are listed in Supplementary Table 4. Mouse β-actin was used as a reference control. Changes in expression levels were calculated with the $2^{-\Delta\Delta Ct}$ method.

## Genome-wide RNA-sequencing

The hippocampus was dissected from the mouse brain and was rapidly frozen in liquid nitrogen. Three biological replicates from three mice were used for each experimental group. Total RNA was extracted using TRIzol RNA isolation reagent (ThermoFisher) following the protocol provided by the manufacturer. After the purity and concentration detection, the library preparation and RNA sequencing were carried out in the Shanghai Meiji Biomedical Technology Co., Ltd. (Shanghai, China). In brief, the prepared cDNA libraries were sequenced as 150 bp paired-end reads with the Illumina HiSeq 4000 sequencer instrument. The sequenced raw data in FASTQ format were filtered out adapter, reads containing ploy-N and low-quality reads from raw data. The acquired clean reads with high quality were used for further analyses.

## Gene set enrichment analysis (GSEA)

GSEA (https://www.gsea-msigdb.org/gsea/index.jsp)[94] was applied to investigate the affection of FAM92A1 deficiency on a priori-defined set of genes. The analyses were conducted in a desktop application (version 4.2.3) as previously reported[95]. The curated gene sets (c2.all.v7.5.1.symbols.gmt) from the molecular signatures database (MSigDB) were used as the reference gene sets. The number of permutations was set at 1000. The nominal $P < 0.01$ and FDR < 0.25 were considered statistically significant. A positive NES indicates enrichment in the morphine, while a negative NES indicates enrichment in the saline. The enrichment score of a single gene set is estimated by nominal $P$.

## Protein expression and purification

Human FAM92A1 residues 1–219 (WT) were cloned into pHAT vector between SpeI and NsiI cloning sites. The mutant construct of FAM92A1 residues 1–219 with mutations Leu70Glu-Phe73Glu-Phe184Glu was generated through site-directed mutagenesis as described[96]. Primer sequences used for generating the mutant are listed in Supplementary Table 4. Both constructs were expressed as his-tagged fusion proteins in *E. coli* BL21 (DE3) at 16 °C for 20 h with 1 mm isopropyl β-D-1-thiogalactopyranoside (IPTG) induction. Cells were harvested by centrifugation and bacterial pellets were resuspended in lysis buffer (20 mM Tris-HCl, pH 8.8, 300 mM NaCl, 5 mM CHAPS, 40 mM Imidazole) supplemented with proteinase inhibitors (PMSF stock 20 mg/mL Pepstatin A 1 μg/mL Leupeptin 1 μg/mL Aprotinin 1 μg/mL) and *DNAse I* (20 μg/mL). Cells were lysed by sonication and the extract was centrifuged for 30 min at $15,000 \times g$. Supernatants were incubated with Ni-NTA Superflow beads (Sigma) for 2 h at 4 °C, followed by elution with an imidazole gradient. The enriched proteins were further purified by gel filtration using Superdex 75 column (GE Healthcare).

## Detection of protein dimerization state

The dimerization states of the wild-type and mutant FAM92A1 BAR domain proteins were assessed using multi-angle laser light scattering with the MiniDAWN light scattering detector and the Optilab dRI detector (Wyatt technology) conjugated with high-performance liquid chromatography (Agilent). The purified proteins were loaded onto a Superdex 200 10/300 column at 4 °C, with a flow rate of 0.5 mL/min. Data analysis was performed using Astra 8.1 software provided by Wyatt technology.

## Crystallization of FAM92A1 BAR domain, data collection, and processing

The purified human FAM92A1 (residues 1–219) was concentrated to 8 mg/mL and crystallized as thin plates from 0.1 M citrate pH 6.0 and various alcohols, the final best diffracting crystals were obtained from 0.1 M citrate pH 6.0 and 6% tert-butanol, flash frozen in 30% tert-butanol, with the drop additionally covered with paratone-N oil.

Data were collected at Diamond Light source beamline I24. The protein crystallized in P1 space group and diffraction data was collected to 2.03 Å. Data were processed with AutoPROC (https://www.globalphasing.com/autoproc/) pipeline using XDS[97] and pointless and aimless for data reduction and scaling[98] (Supplementary Table 1).

## Structure solution and refinement

The structure of FAM92A1 BAR domain was solved with Arcimboldo Lite via the CCP4 GUI with ab initio helical molecular replacement in coiled-coil mode using eight 20 residues helical fragments as the search model and the molecular replacement model was completed with SHELXE[99,100]. After the model was subjected to automated refinement with ARP/WARP and manually corrected with Coot and refined with Phenix[101,102]. The final model had R-factors of $R_{work}/R_{free} = 19.0/23.7\%$. Overall statistics are given in Supplementary Table 1. The crystal structure of FAM92A1 BAR domain protein structure coordinates has been deposited to PDB with the accession code 8CEG (PDB ID 8CEG).

## Molecular dynamics simulations

The crystal structure of the FAM92A1 BAR domain (residues 1–214) was used for the simulation studies. Model bilayers with a lipid composition resembling the mitochondrial inner membrane (MIM) were built using GROMACS tools (Supplementary Table 2). To investigate the interaction of the FAM92A1 BAR domain with negatively charged phospholipids, model lipid bilayers with either phosphatidylinositol bis-phosphate (PIP2 bilayer) or cardiolipin (cardiolipin bilayer) were generated. A monomer unit of the protein was placed at around 3.4 nm above the surface of the lipid bilayer with dimensions of about 18 nm along the x, y, and z-axes. The system was solvated, counter ions added, energy minimized, and equilibrated under NVT and NPT conditions for 500 ps and 1 ns, respectively. The equilibrated systems were then simulated for 500 ns. For membrane curvature studies, lipid bilayers of dimensions 45 nm, 15 nm, and 15 nm along x, y, and z-axes, in respective order, were generated. We also generated PIP2 and cardiolipin bilayers to examine the effect of these negatively charged lipids (phosphatidylinositol bis-phosphate and cardiolipin respectively) on BAR domain-induced membrane curvature. The FAM92A1 BAR domain dimer was placed initially at about 0.2 nm above the surface of the lipid bilayer. The systems were simulated for 1 μs (Supplementary Table 3).

All-atom molecular dynamics simulations were performed using the GROMACS simulation package ver. 2022.5[103]. The Charmm 36/m forcefield with virtual interaction sites was used for proteins, lipids, water, and counter ions[104–107]. For water, we used the TIP3P model. The Nose-Hoover thermostat[108] was used for maintaining the temperature at 310 K with a coupling constant of 0.5 ps, and the Parrinello-Rahman barostat[109] was used for maintaining a semi-isotropic pressure of 1 atm along the membrane surface (xy-plane) and 1 atm along the membrane normal (z-axis) with a coupling constant of 2 ps. Neighbor list was updated every 10 steps using the Verlet cutoff scheme. Covalently bonded hydrogen bonds were constrained using the LINCS algorithm[110]. Short-range electrostatic and van der Waals interactions were cut off at 1.0 nm while long-range electrostatic interactions were treated using the Particle Mesh Ewald method[111].

Analyses were performed using standard GROMACS tools. Simulations were visualized using the VMD software[112]. 2D density plots were generated using g_mydensity and dispgrid.py scripts[113]. To measure membrane phospholipid interaction with FAM92A1 amino acid residues, we calculated the occupancy period of lipids around each amino acid residue. A distance of at least 0.4 nm between amino acid residue and membrane lipid was chosen as a criterion to consider interaction between them. We define occupancy as the longest continuous period of interaction between an amino acid and a membrane lipid. The values were normalized to the length of the simulation period starting from the first point of interaction between any amino acid of FAM92A1 and any membrane phospholipid. A value of 1 indicates that the amino acid always interacts with the membrane phospholipid and a value of 0 indicates that the amino acid never interacts with the membrane phospholipid. We analyzed the curvature profile of the membrane leaflet based on the average position of phosphate atoms of all the membrane phospholipids present in the bilayer. The positions of the phosphate atoms along the membrane normal (z-axis) and along the long axis of the membrane (x-axis) were recorded at intervals of 200 ps.

## Preparation of phospholipid vesicles

The following synthetic lipids were used to prepare mitochondrial inner membrane liposome: 1-palmitoyl-2-oleoyl-sn-glycero-3-phosphocholine (POPC), 1-palmitoyl-2-oleoyl-sn-glycero-3-phosphoethanolamine (POPE), 1-palmitoyl-2-oleoyl-sn-glycero-3-phospho-L-serine (POPS), 1,2-dioctanoyl-sn-glycero-3-phospho-(1′-myo-inositol-4′,5′-bisphosphate) $(PI(4,5)P_2)$, cardiolipin, l-α-phosphatidylethanolamine-N-(lissamine rhodamine B sulfonyl) (RhodaminePE). All lipids were purchased from Avanti Polar Lipids Inc. (Alabaster, AL). The lipid composition was prepared as described[114], POPC: POPE: POPS: $PI(4,5)P_2$: cardiolipin: RhodaminePE = 40:32:3:5:18:2, respectively. Lipids were mixed at desired concentrations and dried under a stream of nitrogen. The lipid mixture was then maintained under reduced pressure for at least 4 h. Subsequently, the dry lipids were hydrated in 20 mM Hepes buffer (pH 7.5) containing 150 mM NaCl to form multilamellar vesicles.

## Co-sedimentation assay

The membrane binding ability of purified proteins was determined by the vesicle co-sedimentation assay. Purified proteins and liposomes were mixed at final concentrations of 2 μM and 500 μM, respectively, in 20 mM Hepes buffer (pH 7.5) with 150 mM NaCl. After incubating at RT for 15 min, the mixture was centrifuged at 436,000 × g for 30 min at 20 °C using a TLA-100 rotor (Beckman Coulter) to separate membrane-bound (pellets, P) and membrane-free fractions (supernatants, S). Equal proportions of supernatants and pellets were loaded onto 12.5% SDS-PAGE. Following electrophoresis, the gels were stained with Coomassie blue, and the intensity of the protein bands was quantified using Fiji. The lipid composition used was POPC: POPE: POPS: $PI(4,5)P_2$: cardiolipin: RhodaminePE = 40:32:3:5:18:2, respectively.

## Membrane remodeling assay by electron microscopy

For electron microscopy experiments, the multilamellar vesicles were extruded through a polycarbonate filter (400 nm pore size) using a mini extruder (Avanti Polar Lipids) to obtain unilamellar vesicles. Samples were negatively stained by mixing 2 μM protein with 500 μM unilamellar vesicles (with a diameter of 400 nm) in 20 mM Hepes buffer (pH 7.5) and 150 mM NaCl at RT for 5 min. The mixture was then applied to the glow-discharged Pioloform (Agar Scientific)- and carbon-coated copper grids and stained with 2% uranyl acetate. Excess solution was removed at each step using filter paper. Membrane morphologies were examined using a JEM-1400 electron microscope (Jeol ltd, Japan) equipped with an Orius SC 1000B bottom-mounted charge-coupled device camera (Gatan). The lipid composition used for EM was POPC: POPE: POPS: $PI(4,5)P_2$: cardiolipin = 40:34:3:5:18, respectively.

## Behavioral tests

All behavioral tests were conducted in rooms between 09:00 and 17:00 during the light phase of the light/dark cycle. Throughout all behavioral tests, the experimenter was blinded to the genotype of mice. All mice were randomly tested. To avoid artifacts brought on by residual smell cues, the equipment was cleaned with 70% ethanol and super hypochlorous after each test. Mice were sacrificed via carbon dioxide inhalation.

## Morris water maze (MWM) test

MWM test was conducted as the reported procedure with minor modifications[115]. Prior to experiments, mice were acclimated in the experimental room for 3 days. The diameter of the testing pool and platform was 150 cm and 12 cm, respectively. The appropriate volume of water was added into the pool to ensure that the platform was submerged 1 cm below the water's surface. The water temperature in the pool was maintained at a constant 23.0 ± 0.5 °C. Mice were trained for five consecutive days and a duration of 90 sec was employed during the training phase. Mice were trained for 4 trials each day with the hidden platform located in the southwest (SW) quadrant and a visual landmark of cues on the walls. The experimenter guided the mice to the platform if the mice did not find the platform within 90 sec of training. The mice were placed on the platform and left there for 15 sec before being taken off. On the 6th day, the hidden platform was removed and a 60 sec probe was performed. The escape latency in reaching the quadrant of the former hidden platform and swimming velocity was recorded, and the percentage of time spent in the target quadrant was also determined. The trajectory of the mouse movement was traced automatically using ANY-maze software (Stoelting CO., LTD, Wood Dale, IL, USA).

## Fear conditioning (FC) test

FC test was conducted as the reported procedure with minor modifications[116]. The apparatus consists of a fear conditioning chamber with a grid floor that could deliver an electric shock and a digital camera mounted above the center location. In brief, on day 1, the mouse was placed in the chamber and allowed to freely explore the chamber for 96 sec, after which a 60 dB white noise cue commenced for 30 sec. The cue co-terminated with a mild foot shock (0.50 mA) for 2 sec. The tone/foot-shock stimuli were repeated another 2 times after 96 sec. After an additional 96 sec, the mouse was returned to its home cage. On day 2, the mouse was placed in the chamber with the same contextual environment without foot shock and cue tone. The mouse behavior was recorded for 180 sec and scored automatically using ANY-maze software (Stoelting CO., LTD, Wood Dale, IL, USA). On day 3, the mouse was placed in the chamber with a changed contextual environment. The mouse behavior was recorded for 480 sec with three repeated white noise cues (60 dB, 30 sec) as on day 1 but without foot shock. Freezing time was defined only when the mouse was immobile for at least 2 sec except for respiration.

## T-maze spontaneous alternation test

T-maze spontaneous alteration test was conducted as the reported procedure with minor modifications[117]. The apparatus consists of three arms (two goal arms: 30 cm × 6 cm × 15 cm; one starting arm: 40 cm × 6 cm × 15 cm, length × width × height). Each mouse was placed at the distal end of the start arm, with its head oriented towards the southern end of the maze, allowing it to choose between one of the two goal arms. Once the mouse chose an arm, the guillotine door was shut to keep the mouse in the chosen arm for 30 sec. At the end of the confinement period, the mouse was gently taken out, and repeat the trial for another 6 times. The mouse chose the same arm explored in the previous trial as a perseveration (error) and chose a different arm as a correct alternation. The percentage of alternation was calculated using the formula: (total number of correct alternations/6) * 100.

## Novel object recognition (NOR) test

NOR test was conducted as the reported procedure with minor modifications[118]. The apparatus consists of a rectangular open field arena (48 cm × 48 cm × 31 cm, length × width × height) with a camera positioned above. The EthoVision 7.0 software (EthoVision 7.0; Noldus Information Technology, Leesburg, VA, USA) was used to automatically trace the exploration behavior. The experiment consisted of three phases: habituation, familiarization, and the test phase. During the habituation phase (on day 1), each mouse was transferred from the cage to the arena without any object. The mouse was allowed to freely explore the empty arena for 10 min and subsequently returned to the cage. During the familiarization phase (on day 2), two identical objects were placed in the arena in two positions (position A and B) before transferring the mouse to the arena. The mouse was then given 10 min to freely explore the arena. To avoid an innate preference for a location, the positions of the two objects were symmetric and fixed between mice. During the test phase (on day 3), the object in position B was replaced by a novel item with a different shape and color. The mouse was then reintroduced to the arena and was allowed to explore for 10 min. The time spent exploring the object in positions A and B was separately quantified. The discrimination index was calculated as the time exploring the new object minus the time exploring the old object and then divided by the total time exploring both objects in the test session.

## Open field test

The open field test was conducted as the reported procedure with minor modifications[119]. In brief, each mouse was transferred from the cage to the middle of the open field chamber (48 cm × 48 cm × 31 cm, length × width × height) with a camera positioned above. The mouse was allowed to freely travel for 15 min and the mouse trajectory was automatically traced with the EthoVision 7.0 software (EthoVision 7.0; Noldus Information Technology, Leesburg, VA, USA).

## Statistical analyses

All investigators were blind to experimental status during the acquisition and analysis of data. Unless otherwise noted, statistical analysis was performed using GraphPad Prism 9 software (version 9). Comparisons between the two groups were conducted using an unpaired Student's $t$-test (two-tailed). Comparisons among multiple groups with one independent factor were done with ordinary one-way ANOVA. Comparisons among multiple groups with two independent factors were performed using two-way ANOVA followed by post hoc analysis with Dunnett's multiple comparisons test or Tukey's multiple comparison test. The survival curves were analyzed using the log-rank test. The $g$-ratio among three groups with or without controlling covariate (axon diameter) was separately conducted with the ANCOVA followed by the Bonferroni post hoc test and Kruskal–Wallis H test using SPSS software (version 27.0, IBM Corporation). In all cases, $p < 0.05$ was set as the cut-off for statistical significance. Details of biological replicates and statistical analysis are described in the corresponding figure legends.

## Reporting summary

Further information on research design is available in the Nature Portfolio Reporting Summary linked to this article.

## Data availability

The crystal structure of FAM92A1 BAR domain generated in this study has been deposited in the Protein Data Bank (PDB) database under accession code 8CEG. The simulation models are openly available on Zenodo [https://zenodo.org/doi/10.5281/zenodo.12647752]. The FIB-SEM data generated in this study have been deposited in the Electron Microscopy Public Image Archive (EMPIAR) database under accession code EMPIAR-12041. The RNA-seq data generated in this study have been deposited in NCBI's Gene Expression Omnibus (GEO) database under accession code GSE264202. Source data are provided with this paper.

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

## Acknowledgements

We deeply acknowledge Mervi Lindma from the Electron Microscopy Unit, HiLIFE—Institute of Biotechnology, (University of Helsinki and Biocenter Finland) for excellent technical assistance on FIB-SEM sample preparation. We gratefully acknowledge the facilities and expertize of the HiLIFE Crystallization unit at the University of Helsinki, a member of FINStruct and Biocenter Finland. We acknowledge the computing resources provided by the CSC—IT Center for Science Ltd. (Espoo, Finland). We express our sincere gratitude to Prof. Pekka Lappalainen and Juha Saarikangas from the University of Helsinki for their constructive comments on the manuscript. We would like to express our gratitude to the Institute of Biophysics, Chinese Academy of Science, for providing us with access to immuno-EM facilities. Additionally, we are thankful to Li Wang for her assistance in Immuno-EM sample preparation. This study was supported by the National Natural Science Foundation of China (grant 32000719 to L.W. and grants 82071494, 81871043, 82371498, and 81272459 to X.C.), 1·3·5 Project for Disciplines of Excellence, West China Hospital, Sichuan University (grant ZYGD23011 to X.C.), the China Postdoctoral Science Foundation (grant 2021M702362 to L.W.), and the Post-Doctor Research project, West China Hospital, Sichuan University (grant 2020HXBH010 to L.W.), the Sichuan Science and Technology Program (grant 23NSFSC2884 to L.W.), the Academy of Finland (grant 323670 to H.Z. and grants 331349, 336234, and 346135 to I.V.), Jane and Aatos Erkko Foundation (to T.K. and H.Z.), the Sigrid Juselius Foundation, Helsinki Institute of Life Science (HiLIFE) Fellow Program, and the Human Frontier Science Program (RGP0059/2019 to I.V.).

## Author contributions

L.W. and Ziyun Y. performed the majority of the experiments and interpretation of the data. Ziyi Y. generated the expression construct of recombinant FAM92A1 BAR protein, and performed protein expression and purification for crystallization. S.F. and T.K. performed the diffraction data collection and crystal structure solution and refinement. X.P. and I.V. performed molecular dynamics simulations. H.V. and E.J. carried out FIB-SEM and 3D remodeling. Yue Z., Y.C., and Q.B. helped with primary neuron culture, transfection, and magnetic resonance imaging. X.H. performed mouse breeding and genotyping. H.L., L.J., F.Q., and N.Z. helped to do mouse behavioral analysis. Ying Z. and M.Q. helped with the analysis of behavioral data. Y.D. helped with the imaging. Yinglan Z. and W.K. revised the manuscript. X.C. and H.Z. conceived the project and wrote the manuscript with contributions from all other authors.

## Competing interests

The authors declare no competing interests.

## Additional information

[1]Mental Health Center and National Chengdu Center for Safety Evaluation of Drugs, State Key Laboratory of Biotherapy, West China Hospital of Sichuan University, Chengdu 610041, China. [2]Faculty of Biological and Environmental Sciences, University of Helsinki, 00014 Helsinki, Finland. [3]Helsinki Institute of Life Science - Institute of Biotechnology, University of Helsinki, Helsinki, Finland. [4]Department of Physics, University of Helsinki, Helsinki, Finland. [5]Helsinki Institute of Life Science (HiLIFE) - Institute of Biotechnology, University of Helsinki, Helsinki, Finland. [6]School of Life Sciences, Guangxi Normal University, Guilin, China. [7]Guangxi Universities Key Laboratory of Stem Cell and Biopharmaceutical Technology, Guangxi Normal University, Guilin 541004, China. [8]Mental Health Center, West China Hospital, Sichuan University, Chengdu, Sichuan 610041, China. [9]These authors contributed equally: Liang Wang, Ziyun Yang. ✉e-mail: hongxia.zhao@helsinki.fi; xbcen@scu.edu.cn

