## [Peer Review File · Nature Communications]

Membrane remodeling by FAM92A1 during brain development regulates neuronal morphology, synaptic function, and cognitionREVIEWER COMMENTS

Reviewer #1 (Remarks to the Author):

As requested by the editor my comments will be mainly on evaluation of mouse MRI and voxel-based morphometry experiments. For this part of the work Wang et al. examined the brain morphology of FAM92A1 knockout and Wilt type mice as a control using volumetric MRI. In summary, authors described a brain swelling and a reduction in gray matter in FAM92A1 KO mice.

Specifically, authors used 7T MRI (Bruker) and did the images acquisition on anesthetized animals. Authors found an increased brain volume in FAM92A1 KO mice. Then, to investigate more in detail this increased of brain volume, authors performed a whole brain voxel-based morphometry (VBM) of the gray matter. The authors find that in general gray matter volume is decreased in FAM92A1 KO mice. These changes seem very localized and subtle.

Pre and post processing of the data are appropriate and well conducted. All together the MRI experiments presented by Wang et al. are well designed with correct statistical analyses.

Minor comments

-As these changes are subtle why author didn't do MRI on fixed brain as classically done for VBM (see: PMID: 25199916 PMID: 22557981) to increase the resolution of the images?

Reviewer #2 (Remarks to the Author):

Wang and colleagues report a comprehensive functional analysis of the BAR-domain containing protein FAM92A1, including the generation and morphological characterization of gene knockout mice, behavioral studies, neuronal activity measurements, experiments probing morphology and function of mitochondria, cell-based assays, such as co-localization and endocytic uptake experiments, EM analyses of synapses, structural biology and molecular dynamics simulations. Based on their analyses, they propose a function of FAM92A1 in the regulation of synaptic plasticity and neural function by controlling endocytosis and mitochondrial membrane remodeling.

This is an interesting and noteworthy study uncovering a role of FAM92A1 in neuronal development and membrane remodeling. However, the wide breadth of different disciplines used to characterize the function of on FAM92A1 comes at the expense that some of the experiments are not conducted in the required detail to be fully conclusive (at least in the second part). My expertise is in structural biology and endocytic trafficking so I will mostly comment on experiments starting from Fig. 6.

Major:

Fig. 6A: Cristae morphology needs to be quantified from several independent micrographs.

Fig. 6M/N: Synaptic area appears to be increased in heterozygous, but not so much in homozygous ko mice. Any explanation for this? While it is an interesting observation worth to report, it seems difficult to conclude from these results that FAM92A1 has a role in regulating synaptic vesicle number/synaptic activity. Please discuss.

Fig. 7A and Fig. 1B: Please provide a specificity control for the FAM92A1 antibody in immunostainings by showing that the respective ko cells and ko brain slices are not stained by the antibody (obviously using the same protocols as for wt). Is only a single FAM92A1 band recognized in the Western blots or are there also non-specific bands? Has the antibody been raised against the BAR domain or some region outside the BAR domain? Please mention the latter in the methods.

Fig. 7A top: The endogenous FAM92A1 staining looks mostly punctate and cytoplasmic with some minor mitochondrial co-localization. I do not see much plasma membrane staining, beyond some random dots. The authors should repeat these experiments using TIRF microscopy and staining of the endogenous FAM92A1 protein, together with a marker for clathrin coated pits (CCPs) and/or markers for clathrin-independent uptake routes. Without a clear co-localization of FAM92A1 to CCPs, it is difficult to judge whether the observed difference in CCP morphology (Fig. 7L) and

endocytic uptake (Figure 7G-H) is a direct effect on clathrin-mediated endocytosis or indirectly caused, for example by defective mitochondria/lipid metabolism.

Fig. 7A bottom: The staining pattern of GFP-tagged FAM92A1 looks different from the endogenous staining, e.g. more cellular structures are stained which could be an over-expression artefact. I would only show the endogenous stain (if it is specific, see point above). Same worries for Fig. 7B. Is this endogenous staining (as stated in the main text) or staining of FAM92A1-GFP (as stated in the figure)? Please show the endogenous FAM92A1 staining for these experiments.

Fig. 8: The structural data for the FAM91A1 BAR domain appear convincing, based on the provided data table and in the absence of a validation report. However, the follow-up evaluation of the structural data is not conclusive:

While the significance of the BAR domain dimer may be inferred by comparison with other BAR domains, the suggested oligomer interface cannot – BAR domain proteins use different oligomerization interfaces for their assembly on membranes. Does FAM91A1 form a dimer in solution and can it be disrupted by mutations in the dimer interface? Are the observed dimer and oligomer interfaces required for the interaction with liposomes and membrane tubulation?

Fig. 8: The authors should discuss or show a scheme how the positive curvature of the BAR domain dimer relates to its localization in the mitochondrial matrix/cristae.

Minor

Fig. 8: Please add the domain architecture of FAM92A1 with the amino acid numbers, also showing the crystallized construct, e.g. even if it is only a BAR domain. The absence of an SH3 domain may hint for a localization other than CCPs, as most BAR domain proteins at CCPs contain an SH3 domain required for their recruitment.

Numbers, errors: Numbers throughout the manuscript should be rounded to the first digit of their error, e.g. $334 \pm 4 \text{ mm}^3$, not $333.5 \pm 3.812 \text{ mm}^3$, $0.04 \pm 0.01 \mu\text{m}^3$, not $0.038 \pm 0.014 \mu\text{m}^3$.

At some sites, PMIDs instead of references are provided.

Reviewer #3 (Remarks to the Author):

Wang et al. investigate the neuronal functions of FAM92A1, a BAR domain protein previously shown to be important for membrane remodeling and proper mitochondrial structure and function. Using a newly generated murine KO model, the authors here report that global deletion of FAM92A1 results in pleiotropic effects in the CNS, ranging from deficits in synapse structure to neuronal network dysfunction and impaired memory. In the second part of the ms, the authors solve the crystal structure of FAM92A1 and perform MD simulations to elucidate its membrane interactions.

General points

(1) The authors conclude that their characterization of CNS effects in FAM92A1 KO mice revealed the “molecular basis of the pathophysiology of FAM92A1-linked neurological disorders in humans” (l. 108). However, which disorder(s) do the authors refer to here? In l. 63, they cite Debost-Legrand et al. (2013) who reported a case of autism associated with 8q21.3 deletion, which may or may not be due to FAM92A1 loss. Even if this were the case, how does the microencephaly found in these patients relate to brain swelling observed in FAM92A1 KO mice? Mutations in FAM92A1 have further been linked to polydactyly in human and mice (Schrauwen et al. 2019 PMID: 30395363), but this phenotype was not explored in the present ms. A related question is whether male infertility in mice as observed in the present ms has been found in patients with mutations in FAM92A1? In sum, the above conclusion is not fully supported by the data provided.

(2) Most investigations were performed at an age when a substantial percentage of KO mice had already died (Fig. 1H). This suggests that the surviving mice were severely sick at the time of investigation, implying that the observed alterations may well be due to systemic factors rather than more direct effects of FAM92A1 in the CNS. Indeed, FAM92A1 seems to be ubiquitously expressed from embryonic stages onward (Figs. 1 and S1). This issue may be particularly relevant for “high-level functions” such as synaptic transmission (Figs. 4-7), neuronal network dynamics

(Fig. 5) or behavior (Fig. 3). As the authors analyzed a conventional KO, FAM92A1 was absent from the onset of embryogenesis, which further makes it impossible to discriminate between acute effects (at the time of investigation) vs. developmental impairments. One way of addressing this question is by inducing a more spatially and temporally restricted KO or KD of FAM92A1. An advanced mechanistic understanding would considerably increase the relevance of the ms. (3) The ms suffers from several shortcomings regarding the reporting of data analysis and statistics, which challenge several of the authors' main conclusions (see below).

Additional points & suggestions

- Statistical details can only be found in the figure legends and are frequently incomplete. To increase the transparency of reporting, I strongly suggest adding a supplementary table containing detailed statistical information for all datasets included.

- Incomplete statistics:

(i) ANOVA in Fig. 1G: Details of post-hoc tests need to be added.

(ii) Fig. 1B, C, D, E: The number of biological replicates (n) is lacking.

(iii) Fig. 1G: "n = 3 replicates/group": Are these biological or technical replicates?

(iv) l. 152 "departure from the expected Mendelian inheritance pattern": Please add test details.

(v) l. 189 "Though the statistical differences were less prominent, a consistent reduction [...]": This conclusion is invalid, as Fig. 2E shows that there is no significant difference in any of the analyzed hippocampal areas!

(vi) Fig. 3G, H: n is lacking.

(vii) Fig. 6A: n is lacking

(viii) Fig. 6G: Please add statistical details.

(ix) Fig. 6H: The use of a standard ANOVA is questionable, as g-ratio depends strongly on axon diameter (Fig. 6G). ANCOVA or similar may be more appropriate. What is n here (mice/slices/axons)?

(x) Fig. 6N: I could not find any statistical details, i.e., the conclusion is not adequately supported by data.

- The relevance and mechanism of brain "swelling" (Fig. 1) remains unclear, as neither changes in CSF nor volume changes of white or grey matter can explain it. This discrepancy should be resolved.

- Fig. 1L, right: 60 mm seems to be incorrect.

- Fig. 2C: The unit should be added.

- Fig. 2G-H: The reported reduction is not evident from the sample images. Also, how does this relate to the volume seen with MRI?

- l. 215 "reduced regional gray matter volume, which may be attributed to neuronal degeneration": This conclusion is not convincing, as no reduction of gray matter volume (MRI) was seen in DG, where Fluoro-Jade staining was performed. It remains unclear to me why these histological analyses have focused on DG rather than those areas in which gray matter volume changes were found by MRI.

- Fig. 3G-J: These analyses are of peripheral interest to the present ms.

- Fig. 4: It remains unclear which neurons were traced in Sholl analyses (l. 278: "hippocampal neurons"). Please add all relevant details to Methods. Again, the brain region analyzed is obviously not motivated by the MRI findings in Figs. 1 and 2. Please explain.

- Fig. 5A-C: Methods related to Ca²⁺ imaging are completely lacking. It is unclear to me how the experiments were conducted and what exactly was measured/quantified. For example, the authors mention spontaneous activity, but Fig. 5B shows trials. Please add all relevant details to Methods.

- Fig. 5D-E and l. 330 ("impact of FAM92A1 on excitatory synaptic transmission is more pronounced at individual synaptic sites"): The data provided do not support such a conclusion! Also note that mEPSC and sEPSC amplitudes are basically the same.

- l. 333 "further supported by the decreased number of glutamate release sites per neuron": A reduction in mEPSC frequency does not necessarily imply a lower number of release sites, as it is also strongly dependent on release probability. Please correct.

- Fig. 5E: Why are the lower traces interrupted? What does the green line in the left trace indicate?

- MEA recordings: The results are very difficult to read, mainly because the definitions of the reported quantities are not properly introduced. For example, "synchrony index" is not defined. Please add all analytical details to Methods and rephrase Results for clarity. What do the gray bars in Fig. 5J indicate?

- l. 424-425 "area of synaptic vesicles (SVs) ... above 5000 nm³": Did the authors analyze areas or volumes?

- l. 89 "structural characteristics have not been elucidated yet": Why is Breugel et al. 2022 (PMID:

35383272) not cited?

- The authors repeatedly use words such "valuable" (l. 42) and "compelling evidence" (l. 254, l. 736) when referring to their own data. This is inadequate and should be corrected.
- l. 53: CIBAR1?
- Refs. 18 and 20 are the same.

Point-by-Point Response to Reviewers' Comments

We deeply appreciate the insightful comments and suggestions provided by the reviewers, which have significantly improved the quality of our work. In response to these comments and suggestions, we have incorporated new data, re-analyzed existing data, and restructured specific sections of the manuscript accordingly. We believe that these additional works significantly improve the quality and comprehensiveness of our manuscript. Below, we present a detailed, point-by-point response to the reviewers' comments, with reviewer remarks shown in black and our responses in blue. Please be aware that the figure citations in our response pertain to the new (post-revision) figures. We have highlighted the corresponding changes within the manuscript in red. We look forward to your feedback and the opportunity to have our work published in *Nature Communications*.

Reviewer #1 (Remarks to the Author):

As requested by the editor my comments will be mainly on evaluation of mouse MRI and voxel-based morphometry experiments. For this part of the work Wang et al. examined the brain morphology of FAM92A1 knockout and Wilt type mice as a control using volumetric MRI. In summary, authors described a brain swelling and a reduction in gray matter in FAM92A1 KO mice.

Specifically, authors used 7T MRI (Bruker) and did the images acquisition on anesthetized animals. Authors found an increased brain volume in FAM92A1 KO mice. Then, to investigate more in detail this increased of brain volume, authors performed a whole brain voxel-based morphometry (VBM) of the gray matter. The authors find that in general gray matter volume is decreased in FAM92A1 KO mice. These changes seem very localized and subtle.

Pre and post processing of the data are appropriate and well conducted. All together the MRI experiments presented by Wang et al. are well designed with correct statistical analyses.

Response: We thank the reviewer for the appreciation of our study and the constructive comments. Following the suggestions, we have conducted additional experiments as the reviewer's suggested method and addressed this reviewer's concerns as described below. We hope that our response sufficiently addresses the concern raised by the reviewer.

Minor comments

-As these changes are subtle why author didn't do MRI on fixed brain as classically done for VBM (see: PMID: 25199916IF: 11.0 Q1 PMID: 22557981IF: 4.7 Q2) to increase the resolution of the images?

Response: We appreciate the reviewer's valuable comments. As suggested, we have acquired new MRI scans with enhanced contrast by utilizing fixed mouse brains perfused with 2 mM ProHance. Due to the super time-consuming nature of scanning the entire brain and the limited available homozygous mice suitable for perfusion, we were only able to scan three mice from each wild-type and FAM92A1^{+/-} group for analysis. Despite the small number of mice used, the deformation-based morphometry (DBM) analysis of the volumetric morphology of the mouse brain and the

voxel-based morphometry (VBM) analysis of gray matter revealed a noticeable decrease in volume outside of the lateral ventricles (LVs), accompanied by a reduced gray matter density within the diminished cortical area (**Supplementary Fig. 2e**). These data suggest that the suggested method can enhance sensitivity to detect subtle differences in hippocampal subregions. We thank the reviewer for suggested methods, we have now used and cited this work.

We have supplemented test details and new data in our revised manuscript. Changes in corresponding *Materials* (Lines 997-1001 and 1029-1041), *Results* descriptions (Lines 203-226), and *Figure legends* (Supplementary file, Lines 62-69) are marked in red.

Reviewer #2 (Remarks to the Author):

Wang and colleagues report a comprehensive functional analysis of the BAR-domain containing protein FAM92A1, including the generation and morphological characterization of gene knockout mice, behavioral studies, neuronal activity measurements, experiments probing morphology and function of mitochondria, cell-based assays, such as co-localization and endocytic uptake experiments, EM analyses of synapses, structural biology and molecular dynamics simulations. Based on their analyses, they propose a function of FAM92A1 in the regulation of synaptic plasticity and neural function by controlling endocytosis and mitochondrial membrane remodeling.

This is an interesting and noteworthy study uncovering a role of FAM92A1 in neuronal development and membrane remodeling. However, the wide breadth of different disciplines used to characterize the function of on FAM92A1 comes at the expense that some of the experiments are not conducted in the required detail to be fully conclusive (at least in the second part). My expertise is in structural biology and endocytic trafficking so I will mostly comment on experiments starting from Fig. 6.

Response: We thank the reviewer for the appreciation of our study and the constructive comments. Following the suggestions, we have conducted additional experiments and addressed the reviewer's major concerns as described below. We appreciate the invaluable comments provided during the review process. This included the incorporation of new MRI data and a reanalysis of previous MRI data to comprehensively understand the cause of brain swelling. Based on the latest findings, we have meticulously revised the description and succinctly summarized the results in the manuscript. We believe that the quality of our study has been significantly improved by addressing these concerns. We hope that our response adequately addresses the concerns raised by the reviewer.

Major:

Fig. 6A: Cristae morphology needs to be quantified from several independent micrographs.

Response: We appreciate the reviewer's comments. The impact of FAM92A1 depletion on mitochondrial morphology,

including mitochondrial diameter, perimeter, and the number of cristae per mitochondrial perimeter, has been assessed using electron micrographs as we previously reported. In line with our earlier findings, FAM92A1 depletion results in swollen mitochondria with reduced or even absent cristae (**Fig. 6b**).

We have supplemented these new data in our revised manuscript. Changes in corresponding **Results** (Lines 438-444) descriptions and **Figure legends** (Lines 2134-2135) are marked in red.

Fig. 6M/N: Synaptic area appears to be increased in heterozygous, but not so much in homozygous ko mice. Any explanation for this? While it is an interesting observation worth to report, it seems difficult to conclude from these results that FAM92A1 has a role in regulating synaptic vesicle number/synaptic activity. Please discuss.

Response: We appreciate the reviewer's critical comments. As pointed out by the reviewer, the changes in the average area of SVs in the FAM92A1^{+/-} group were more pronounced than those in the FAM92A1^{-/-} group when compared to the SV area in the FAM92A1^{+/+} group. In the quantification of the synaptic area in previous Fig. 6M, we analyzed data from a similar number of synaptic vesicles (SVs), rather than SVs from the same number of synapses. In fact, through quantification of SVs number (**Fig. 6b**) by a stereological approach, the number of SVs was remarkably increased in the FAM92A1^{-/-} synapses in contrast to FAM92A1^{+/+} synapses. Hence, to get a similar number of SVs (FAM92A1^{+/+}, n = 361; FAM92A1^{+/-}, n = 351; FAM92A1^{-/-}, n = 346) for statistical analysis, a smaller number of synapses were indeed incorporated in FAM92A1^{-/-} group compared to the FAM92A1^{+/+} group in the previous Fig. 6M. Due to most of the SVs among all groups with a diameter < 60 nm (**Fig. 6p**), the average area of SVs was not evident in the FAM92A1^{-/-} group.

To address this concern, SVs from the same number of synapses (n = 15 synapses per group) were used for quantification in the revised manuscript (**Supplementary Fig. 5e, f**). Compared to the SV area in the FAM92A1^{+/+} group, changes in the average area of SVs in the FAM92A1^{+/-} group were still more evident than that in the FAM92A1^{-/-} group. Loss of FAM92A1 not only resulted in decreased endocytosis but also impaired mitochondrial inner membrane architecture and energy production (**Fig. 6a-f**).

Neuronal activity, including neurotransmitter release and synaptic vesicle membrane retrieval at synapses, relies heavily on energy. Depletion of FAM92A1 not only impaired the FAM92A1-involved step of synaptic vesicle membrane retrieval but also inhibited all energy-dependent endocytic processes of synaptic vesicle membrane (*Li et al. 2021. PMID: 34782781; Devine et al. 2018. PMID: 29348666*). Compared to FAM92A1 homozygous mice, both mitochondrial architecture and function were less affected in the FAM92A1 heterozygous mice (**Fig. 6a-f**), indicating less suffering from the energy shortage. As a result, in contrast to the FAM92A1 homozygous, the endocytic process of the synaptic vesicle membrane might occur more frequently in the FAM92A1 heterozygous. Consequently, more

SVs might exhibit enlarged areas due to defects in membrane curvature during endocytosis of the synaptic vesicle membrane. Therefore, we speculated that the aberrant structure of SVs and decreased synaptic activity result from the combined effects of impaired mitochondrial energy provision and the endocytic process of the synaptic vesicle membrane.

Moreover, to observe the changes in SVs, both changes in synaptic number and area have been quantified using the FIB-SEM electron micrographs in the revised manuscript. Synapses displaying distinguishable pre- and post-synaptic areas were measured on a micrograph with dimensions of 2209×205 pixels. Quantification was conducted on every 50th slice, with a total of 18 slices used for analysis (Fig. 6n). Quantification results revealed that FAM92A1 depletion caused an increased ratio of SVs with a diameter larger than 80 nm. The average area of SVs was found to be increased in the FAM92A1-depleted synapses (Fig. 6o-q). Notably, the heterozygous mouse data were not included in this analysis due to the large dataset. This 3D EM data will be deposited in EMPIAR and made publicly visible.

We have supplemented these new data in our revised manuscript. Changes in corresponding *Materials* (Lines 1157-1162), *Results* (Lines 490-498) descriptions, and *Figure legends* (Lines 2153-2155) are marked in red.

Fig. 7A and Fig. 1B: Please provide a specificity control for the FAM92A1 antibody in immuno-stainings by showing that the respective ko cells and ko brain slices are not stained by the antibody (obviously using the same protocols as for wt). Is only a single FAM92A1 band recognized in the Western blots or are there also non-specific bands? Has the antibody been raised against the BAR domain or some region outside the BAR domain? Please mention the latter in

the methods.

Response: We appreciate the reviewer's valuable and critical comments. Both Fig. 1b and 7a depict the localization of FAM92A1 in the brain and HT22 cells, respectively. We tested two commercial FAM92A1 antibodies purchased from Sigma (CAT#HPA034760) and Proteintech (CAT#24803-1-AP) in different experiments. Our test results revealed that both antibodies have their limitations, as is common with technical challenges associated with antibodies. The antigen sequences used for raising these antibodies have been indicated in **Supplementary Fig. 1e**. The FAM92A1 antibody from Proteintech recognized multiple non-specific bands across all samples, including primary fibroblast and brain tissue (**Response Fig. 1b**). The antibody from Sigma displayed fewer non-specific bands.

The two guide RNAs were separately targeted to the upstream of *Fam92a1* exon 1 and downstream of *Fam92a1* exon 6, resulting in the microdeletion of the first six exons (**Supplementary Fig. 1c**). According to the resolved crystal structure of FAM92A1 BAR dimer, the N-terminus residues 2-211 encoded by genes spanning exon 1 to exon 7 is the FAM92A1 BAR domain. Both FAM92A1 antibodies can recognize the part of the sequence of the BAR domain. The FAM92A1 antibody from Sigma recognizes amino acids encoded by the *Fam92a1* gene spanning exon 7 to exon 8, and the FAM92A1 antibody from Proteintech recognizes amino acids encoded by the *Fam92a1* gene spanning exon 4 to exon 8 (**Supplementary Fig. 1e**). Due to the deletion of the first six exons in the FAM92A1 knockout mouse, we utilized the antibody from Proteintech for the knockout efficiency identification through western blot, except for immunostaining. This decision was made because the Proteintech antibody did not perform well in immunostaining experiments, exhibiting non-specific diffused localization in cells due to its recognition of multiple non-specific bands (**Response Fig. 1a, c**). The antibody from Sigma was used to detect the localization of endogenous FAM92A1. However, although the FAM92A1 antibody from Sigma displayed fewer non-specific bands, it exhibited a minor weak non-specific background signal in KO cells and slices in immunostaining experiments (**Response Fig. 1a**). We chose not to include these data in the manuscript due to the potential for misleading information caused by antibody limitations.

We have supplemented these new data in our revised manuscript. Changes in corresponding *Results* (Lines 146-157) descriptions and *Figure legends* (Supplementary file, Lines 37-46) are marked in red.

Fig. 7A top: The endogenous FAM92A1 staining looks mostly punctate and cytoplasmic with some minor mitochondrial co-localization. I do not see much plasma membrane staining, beyond some random dots. The authors should repeat these experiments using TIRF microscopy and staining of the endogenous FAM92A1 protein, together with a marker for clathrin coated pits (CCPs) and/or markers for clathrin-independent uptake routes. Without a clear co-localization of FAM92A1 to CCPs, it is difficult to judge whether the observed difference in CCP morphology (Fig. 7L) and endocytic uptake (Figure 7G-H) is a direct effect on clathrin-mediated endocytosis or indirectly caused, for example by defective mitochondria/lipid metabolism.

Response: We appreciate the reviewer's valuable comments. As suggested, we utilized TIRF microscopy to determine

the localization of endogenous FAM92A1 on the plasma membrane and its co-localization with clathrin and caveolin. The results revealed that FAM92A1 resides in the plasma membrane, although its abundance there was relatively lower compared to its enriched abundance in mitochondria. Moreover, it was found to co-localize with both clathrin and caveolin (**Fig. 7p**). These findings suggest the involvement of FAM92A1 in both endocytic pathways. This parallels another BAR domain protein, endophilinA2, which is also implicated in membrane remodeling during both clathrin-mediated endocytosis (CME) and clathrin-independent endocytosis (CIE) (Boucrot *et al.* 2014. PMID: 29425511 Bertot *et al.* 2018. PMID: 25517094). Consistently, our new immuno-electron microscopy data confirmed the localization of endogenous FAM92A1 on the plasma membrane (The detailed explanation and results were shown in the next response). Collectively, these data indicate that FAM92A1 localizes to the plasma membrane and is required for endocytosis.

We have supplemented these new data in our revised manuscript. Changes in corresponding *Materials* (Lines 1109-1122), *Results* (Lines 573-580) descriptions, and *Figure legends* (Lines 2126- 2128) are marked in red.

Fig. 7A bottom: The staining pattern of GFP-tagged FAM92A1 looks different from the endogenous staining, e.g. more cellular structures are stained which could be an over-expression artefact. I would only show the endogenous stain (if it is specific, see point above). Same worries for Fig. 7B. Is this endogenous staining (as stated in the main text) or staining of FAM92A1-GFP (as stated in the figure)? Please show the endogenous FAM92A1 staining for these experiments.

Response: We appreciate the reviewer's critical comments. As suggested, we have used the FAM92A1 antibody to visualize the subcellular localization of endogenous FAM92A1 both in the HT22 cells and primary hippocampal neurons through immunofluorescence and Immuno-electron microscopy (Immuno-EM). The immunofluorescent images presented that FAM92A1 localized to mitochondria and the plasma membrane in HT22 cells (**Fig. 7a**). The Immuno-EM also revealed that FAM92A1 was located inside mitochondria and at or near the periphery of the plasma membrane (**Fig. 7b**). Similar results were visualized in the primary hippocampal neurons, FAM92A1 was highly expressed in neurons, with partial localization to both the mitochondrial protein VDAC and synaptic proteins SV2, vGlut1, and PSD95 (**Fig. 7c, d and Supplementary Fig. 6d**). We completely understand the concern of the reviewer

regarding the potential artifact of overexpressed FAM92A1-GFP. Therefore, we moved all FAM92A1-GFP overexpressed results into the supplementary figures (**Supplementary Fig. 6c-d**).

We have supplemented these new data in our revised manuscript. Changes in corresponding *Materials* (Lines 1080-1097), *Results* (Lines 514-529) descriptions, and *Figure legends* (Lines 2162-2167 and Supplementary file, Lines 119-128) are marked in red.

Fig. 8: The structural data for the FAM91A1 BAR domain appear convincing, based on the provided data table and in the absence of a validation report. However, the follow-up evaluation of the structural data is not conclusive: While the significance of the BAR domain dimer may be inferred by comparison with other BAR domains, the suggested oligomer interface cannot – BAR domain proteins use different oligomerization interfaces for their assembly on membranes. Does FAM91A1 form a dimer in solution and can it be disrupted by mutations in the dimer interface? Are the observed dimer and oligomer interfaces required for the interaction with liposomes and membrane tubulation?

Response: We appreciate the reviewer's valuable comments. As suggested, we simultaneously mutated three hydrophobic residues at the dimer interface observed in the structure to negatively charged residues, generating a mutant construct (FAM92A1 Leu70Glu-Phe73Glu-Phe184Glu, abbreviated as mutant) (**Supplementary Fig. 7e**). These mutations aimed to disrupt the dimer interface packing. Compared to the dimerized FAM92A1 wild-type BAR domain protein in equilibrium (Wang *et al.* 2019. PMID: 30404948), the purified mutant protein existed as a monomer in equilibrium, with minor aggregates (**Supplementary Fig. 7f**). As shown in the SEC-MALS profile, the major peak in the chromatogram corresponds to a monomeric species, not seen for the WT (Wang *et al.* 2019. PMID: 30404948). This result indicates the importance of these residues for formation of the FAM92A1 BAR dimer

Although the interface mutant lost its capacity to form dimers compared to WT, co-sedimentation assay revealed that the mutant protein preserved its membrane binding ability with a minor decrease compared to the WT protein (**Supplementary Fig. 7g**), while mutation of the membrane binding residues of FAM92A1 BAR (as we previously detected Mut3 and Mut5) have been demonstrated to impair its membrane affinity (*Wang et al. 2019. PMID: 30404948*). Despite retaining the affinity of liposomes, the mutant protein abolished its ability to sculpt the spherical liposomes into narrow tubules, suggesting the absence of membrane remodeling activity (**Supplementary Fig. 7h**). These data demonstrate that the dimerization of the FAM92A1 BAR domain is necessary for membrane sculpting, but not dispensable for binding to the membrane interaction.

The oligomerization interface involves a head-to-head interaction in the crystal lattice (Supplementary Fig. 8b, c). We mutated some residues involved in the oligomerization interface (FAM92A1 Leu127Glu-Leu151Glu-Glu148Ala). However, we noticed non-specific aggregates exist in all the samples, making it difficult to define the oligomerization mutants. The role of protein oligomerization in membrane-interaction and -remodeling is thus challenging to study. We have discussed this in the revised manuscript.

We have supplemented these new data in our revised manuscript. Changes in corresponding **Materials** (Lines 1472-1492 and 1570-1591), **Results** (Lines 617-613) descriptions and **Figure legends** (Supplementary file, lines 147-155) are marked in red

Fig. 8: The authors should discuss or show a scheme how the positive curvature of the BAR domain dimer relates to its localization in the mitochondrial matrix/cristae.

Response: We appreciate the reviewer's insightful comments. The crystal structure of the FAM92A1 BAR domain demonstrates that its curvature closely resembles that of classical BAR domains, which are banana-shaped. Similar to other BAR domain proteins, the concave surface of the FAM92A1 BAR domain generates positive membrane curvature, as we have shown using molecular simulations in this study (Fig. 7h, j) and also our previous giant vesicle model (Wang et al., 2019). Additionally, our previous Immuno-EM data showed that endogenous FAM92A1 resides in the mitochondrial matrix in close proximity to the inner membrane and clusters mostly along the cristae (Figure 1E in Wang et al., 2019). Some localization at the base and tip of the cristae was also observed. Combining these observations with the molecular simulation data, it suggest that FAM92A1 interacts with the mitochondrial inner membrane to generate positive curvature, initiating membrane invaginations to form cristae, and maintains the cristae structure by binding to the inner membrane. We have discussed this in the revised manuscript, and changes in the corresponding *Discussion* (Lines 819-934) descriptions are marked in red.

Minor

Fig. 8: Please add the domain architecture of FAM92A1 with the amino acid numbers, also showing the crystallized construct, e.g. even if it is only a BAR domain. The absence of an SH3 domain may hint for a localization other than CCPs, as most BAR domain proteins at CCPs contain an SH3 domain required for their recruitment.

Response: We appreciate the reviewer's valuable comments. As suggested, we have added the aligned amino acid sequences of human and mouse FAM92A1, the domain architecture of FAM92A1 with amino acid numbering, and the details of the construct used for crystallization of the FAM92A1 BAR domain in the revised manuscript (**Supplementary Fig. 1e and 7a**). At the beginning of crystallizing the FAM92A1 BAR domain, according to the secondary structure of human FAM92A1 and the implied BAR domain from Uniprot, we predicted the N-terminal residues 1-219 constituted the BAR domain (Wang et al. 2019. PMID: 30404948). Therefore, we expressed and purified the N-terminal residues 1-219 (**Supplementary Fig. 7a**). In fact, based on the resolved crystal structure of the human FAM92A1 BAR domain, the N-terminal residues 2-211 is the BAR domain.

FAM92A1 predominantly localizes to mitochondria, with some localization on the plasma membrane. Our new data revealed that FAM92A1 co-localized with the endocytic markers clathrin and caveolin (**Fig. 7p**), suggesting its involvement in both endocytic pathways. FAM92A1 likely plays a role in generating positive curvature of the plasma membrane during the endocytosis process, rather than interacting with other proteins such as N-WASP through an SH3 domain, to coordinate with the actin cytoskeleton during the endocytic process (Defne et al. 2007. PMID: 17609109; Ringstad et al. 1999. PMID: 10677033). However, the detailed molecular mechanism requires further studies in the future. We have discussed this in the revised manuscript.

We have supplemented these new data in our revised manuscript. Changes in corresponding *Materials* (Lines 684-685), *Results* (Lines 146-151 and 588-589) descriptions and *Figure legends* (Supplementary file, Lines 139-140, and 41-46) are marked in red.

Supplementary Fig. 1

Supplementary Fig. 7a

Numbers, errors: Numbers throughout the manuscript should be rounded to the first digit of their error, e.g. 334 ± 4 mm³, not 333.5 ± 3.812 mm³, 0.04 ± 0.01 μ m³, not 0.038 ± 0.014 μ m³.

Response: We appreciate the reviewer for this helpful suggestion. As suggested, we have thoroughly reviewed the entire article, and these kinds of problems in the numbers and errors have been revised in the revised manuscript.

At some sites, PMIDs instead of references are provided.

Response: We thank the reviewer's kind suggestion. We have thoroughly reviewed the entire article and revised these, and the PMIDs have been replaced with references in some sites of the main text.

Reviewer #3 (Remarks to the Author):

Wang et al. investigate the neuronal functions of FAM92A1, a BAR domain protein previously shown to be important for membrane remodeling and proper mitochondrial structure and function. Using a newly generated murine KO model, the authors here report that global deletion of FAM92A1 results in pleiotropic effects in the CNS, ranging from deficits in synapse structure to neuronal network dysfunction and impaired memory. In the second part of the ms, the authors solve the crystal structure of FAMP92A1 and perform MD simulations to elucidate its membrane interactions.

Response: We thank the reviewer for the appreciation of our study and the constructive comments. Following the suggestions, we have conducted additional experiments and addressed this reviewer's major concerns as described

below. We believe that the quality of our study is significantly improved by addressing these concerns and we hope that our response sufficiently addresses the concerns raised by the reviewer.

General points

(1) The authors conclude that their characterization of CNS effects in FAM92A1 KO mice revealed the “molecular basis of the pathophysiology of FAM92A1-linked neurological disorders in humans” (l. 108). However, which disorder(s) do the authors refer to here? In l. 63, they cite Debost-Legrand et al. (2013) who reported a case of autism associated with 8q21.3 deletion, which may or may not be due to FAMP92A1 loss. Even if this were the case, how does the microencephaly found in these patients relate to brain swelling observed in FAM92A1 KO mice? Mutations in FAM92A1 have further been linked to polydactyly in human and mice (Schrauwen et al. 2019 PMID: 30395363IF: 6.2 Q1), but this phenotype was not explored in the present ms. A related question is whether male infertility in mice as observed in the present ms has been found in patients with mutations in FAM92A1? In sum, the above conclusion is not fully supported by the data provided. (2) Most investigations were performed at an age when a substantial percentage of KO mice had already died (Fig. 1H). This suggests that that the surviving mice were severely sick at the time of investigation, implying that the observed alterations may well be due to systemic factors rather than more direct effects of FAM92A1 in the CNS. Indeed, FAM92A1 seems to be ubiquitously expressed from embryonic stages onward (Figs. 1 and S1). This issue may be particularly relevant for “high-level functions” such as synaptic transmission (Figs. 4-7), neuronal network dynamics (Fig. 5) or behavior (Fig. 3). As the authors analyzed a conventional KO, FAM92A1 was absent from the onset of embryogenesis, which further makes it impossible to discriminate between acute effects (at the time of investigation) vs. developmental impairments. One way of addressing this question is by inducing a more spatially and temporally restricted KO or KD of FAM92A1. An advanced mechanistic understanding would considerably increase the relevance of the ms. (3) The ms suffers from several shortcomings regarding the reporting of data analysis and statistics, which challenge several of the authors’ main conclusions (see below).

Response: We appreciate the reviewer’s comments.

(1) We agree with the reviewer that the confirmed neurological disorders associated with FAM92A1, including male infertility in humans as well as experimental data, are currently unknown. The confirmed disease related to the FAM92A1 mutation is polydactyly (Schrauwen et al. 2019. PMID: 30395363). Consistently, during mouse breeding in this study, four of 26 FAM92A1 knockout mice were found to have visible polysyndactyly. For example, mouse with polysyndactyly was observed on the right hind paw (**Response Fig. 2**). Because this manuscript aimed to investigate the potential role of FAM92A1 in the central nervous system. The reported abnormal digit morphology in the *Fam92a*^{-/-} mice, like osteoma (Schrauwen et al. 2019. PMID: 30395363), which is not visible through the eye but is detectable using X-ray. Anyway, the visible polysyndactyly in the FAM92A1^{-/-} mice further confirmed the role of FAM92A1 in maintaining the normal skeleton morphology.

As for the reported case of an autism patient with an 8q21.3 microdeletion, it remains uncertain whether it is merely attributable to FAM92A1 loss. Although the three patients with polydactyly did not exhibit evident neurological problems, neurological disorders may become apparent with age progression, according to our behavioral result of age-associated memory decline and cognitive deficits (Fig. 4 and Supplementary Fig. 4). Hence, the direct link of FAM92A1 mutation with the neurological disorder in humans is hardly known at present. We aimed to elucidate potential associations through IPA analysis of differentially expressed genes following FAM92A1 loss. This bioinformatic investigation revealed a clustering of neurological disorders, including dementia and Alzheimer's disease (AD) (Fig. 3r). These analyses collectively imply that the loss of FAM92A1 expression heightens the susceptibility to neurological disorders. However, the exact neurological disease associated with FAM92A1 requires further investigation.

According to our previous study and current study, we revealed that membrane-remodeling mediated by FAM92A1 is crucial from embryonic development onward. Consequently, the absence of FAM92A1-caused abnormal inner membrane architecture leads to mitochondrial dysfunction, including energy shortage, oxidative stress, and probably metabolic dysfunction. These factors collectively contribute to neurodevelopmental anomalies, a hallmark pathology observed in microcephaly. Notably, an increasing number of patients or infants with mutations in mitochondrial proteins exhibit microcephaly (*Nathalie et al. 2020. PMID: 32294449; Ruth et al. 2016. PMID: 26992161; José-Mario et al. 2015. PMID: 25650066*). For instance, a patient harboring a homozygous variant in *NDUFA8*, affecting mitochondrial complex I, also manifests developmental delay and microcephaly (*Yukiko et al. 2020. PMID: PMID: 32385911*). Additionally, the malformation of brain morphology caused by FAM92A1 deficiency may worsen with decreased neuronal plasticity (Fig. 4 and 5). Moreover, brain swelling in the FAM92A1-deficient mice may reversely lead to increased pressure inside the skull, potentially causing further damage to brain tissue and impeding brain growth. Taken together, according to our experimental results, the observed global developmental delay, autism, and microcephaly in the patient with microdeletions around the *FAM92A1* loci may represent a combined consequence of systemic neurodevelopmental defects.

For male infertility, the reported study did not report whether three men with the homozygous *Fam92a* mutation

have problems with male infertility. And the pedigree did not show linkage analysis for the offspring of these three men (*Schrauwen et al. 2019. PMID: 30395363*). Hence, we are unable to determine whether the male infertility of FAM92A1-depleted mice also occurs in patients with mutations in FAM92A1. Because the critical role of mitochondria in sperm function is widely studied, extending beyond energy production to encompass various functions throughout gamete production and reproduction (*Singh et al. 2020. PMID: 32534048*). Therefore, it is not surprising that FAM92A1 knockout mice exhibit male infertility. Importantly, this phenotype further implies the pivotal role of FAM92A1 in mitochondria. Because the founded male infertility is beyond the scope of this study. We included this phenomenon, which has not been reported in other studies to date, in the manuscript to provide a comprehensive overview of the symptoms observed in FAM92A1 knockout mice. We hope that future studies or case reports will explore this aspect in more detail.

(2) We acknowledge the Reviewer's insightful advice. According to the reviewer's comments, we investigated the role of FAM92A1 in the central nervous system (CNS) using AAV-mediated spatially and temporally knockdown neuronal FAM92A1 in the mouse hippocampus. A miR-30-based short hairpin RNA (shRNA) technique was applied to specifically knockdown the FAM92A1 transcript within the hippocampus. Two siRNA sequences targeting FAM92A1 (FAM92A1 siRNA #1 and #4) (**Supplementary Fig. 3a, b**) were individually cloned into a neurotropic AAV vector (pAAV-hSyn-EGFP-3xFlag), generating pAAV-hSyn-EGFP-3xFlag-miR30shRNA (*Fam92a1*) (abbreviated as sh*Fam92a1*) (**Fig. 3a**). Three weeks after AAV injection into the hippocampus (**Fig. 3b-d**), a series of learning and memory-involved behavioral tests were conducted to assess the acute effects of FAM92A1 depletion on mouse behavior.

Consistent with the memory defects of FAM92A1 knockout mice, mice with FAM92A1 knockdown in the hippocampal neuron presented an attenuated performance in memory tasks, particularly in tasks involving spatial and contextual memory retrieval (**Fig. 3g, i, k, n, and Supplementary Fig. 4b, e, f, i**). In contrast to the control group, mice subjected to FAM92A1 knockdown exhibited decreased accuracy and spent less time in the target quadrant (SW) during the Morris water maze task (**Fig. 3g, i, k**). Additionally, they demonstrated impaired recall of fear memory when reintroduced to the same contextual environment in the fear conditioning test (**Fig. 3n**), and decreased ratio of correct choice in T-maze (**Supplementary Fig. 4i**). Similar to FAM92A1 knockout mice, FAM92A1 knockdown has less affection on the recognition memory of mice (**Supplementary Fig. 4l**). These data collectively demonstrate the role of FAM92A1 in preserving spatial and contextual memory.

In addition to the behavioral tests, our previous study has investigated changes in neuronal morphology following the knockdown of FAM92A1 in cultured hippocampal neurons. In contrast to the enriched presence of mushroom-like spines, FAM92A1 knockdown led to a noticeable reduction in spine density and the absence of mushroom-like morphology (**Fig. 4j**). For the affection of FAM92A1 knockdown on neuronal function, we previously recorded the spontaneous electrophysiological activity of cultured hippocampal neural networks using the microelectrode arrays (MEA) technology. Due to the low probability of visible FAM92A1^{-/-} embryos, we failed to identify the visible FAM92A1^{-/-} embryos even after four attempts. Consequently, we were unable to compare the activity difference between FAM92A1 knockout and knockdown neurons.

We have supplemented these new data in our revised manuscript. Changes in corresponding *Materials* (Lines 1322-1341), *Results* (Lines 252-305) descriptions and *Figure legends* (Lines 2068-2081 and Supplementary file, lines 88-101) are marked in red.

(3) We have comprehensively revised the manuscript. In addition to revising data analysis and statistics, we included new MRI data and reanalyzed previous MRI data to gain a comprehensive understanding of the cause of brain swelling. Furthermore, we carefully revised the description of electrophysiological activity obtained from the microelectrode array test in the revised manuscript. All revisions have been highlighted in red.

Additional points & suggestions

- Statistical details can only be found in the figure legends and are frequently incomplete. To increase the transparency of reporting, I strongly suggest adding a supplementary table containing detailed statistical information for all datasets included.

Response: Thanks for the reviewer's kind suggestion. We have compiled all the raw data used for making graph, statistical information, and exact p values for the datasets, into a table of **Source Data file**.

- Incomplete statistics:

(i) ANOVA in Fig. 1G: Details of post-hoc tests need to be added.

Response: Thanks for the reviewer's kind suggestion. Due to the word limitation of the figure legend, only the statistical methods and the number of samples used for statistical analysis are listed in the Figure legends. Details of statistical analysis, including the analysis methods, post-hoc tests, as well as original data, for all figures have been summarized in the **Source Data file**.

(ii) Fig. 1B, C, D, E: The number of biological replicates (n) is lacking.

Response: We appreciate the reviewer's comments. For Fig. 1b, the distribution of FAM92A1 in the mouse brain was assessed across three mice (**Response Fig. 3**, lower panel). In Fig. 1c-1e, the protein level of FAM92A1 in different brain areas and developmental stages was conducted in three biological replicates, respectively (**Fig. 1c-e**, **Supplementary Fig. 1a, b**, and **Response Fig. 3**, the top three panel).

We have supplemented these quantified data in our revised manuscript. Changes in corresponding **Results** (Lines 124-133) descriptions and **Figure legends** (Lines 2019-2026 and Supplementary file, lines 35-37) are marked in red. The raw data these graphs have compiled into Source data file.

(iii) Fig. 1G: “n = 3 replicates/group”: Are these biological or technical replicates?

Response: Thanks for reviewer’s the careful check. Statistical analysis in Fig. 1g was performed using three technical replicates over one independent experiment (one-way ANOVA), indicating that 'n' represents the number of technical replicates over one independent experiment. We have included the term "technical" in the Figure legend of Fig. 1g to clarify (Lines 2029-2031).

(iv) l. 152 “departure from the expected Mendelian inheritance pattern”: Please add test details.

Response: Thanks for the reviewer’s kind suggestion. According to the Mendelian laws of inheritance, the theoretical frequencies of FAM92A1^{+/+} (q²), FAM92A1^{+/-} (2pq), and FAM92A1^{-/-} (p²) offspring from single breeding of two heterozygous animals are expected to be 0.25, 0.5, and 0.25, respectively (Supplementary Fig. 1g). We quantified the genotype of 235 offspring from 25 litters and found that the observed frequencies of +/+, +/-, and -/- were 0.33, 0.64, and 0.08, respectively (Supplementary Fig. 1h). It's noteworthy that some litters exhibited reduced or no viable FAM92A1^{-/-} offspring (Supplementary Fig. 1i). This observation suggests a departure from the expected Mendelian inheritance pattern for the birth of FAM92A1 homozygotes.

We have supplemented test details and new data in our revised manuscript. Changes in corresponding *Materials* (Lines 906-911), *Results* descriptions (Lines 164-172), and *Figure legends* (Supplementary file, Lines 46-51) are marked in red.

(v) l. 189 “Though the statistical differences were less prominent, a consistent reduction [...]”: This conclusion is invalid, as Fig. 2E shows that there is no significant difference in any of the analyzed hippocampal areas!

Response: Thanks for Reviewer’s the critical comments. We fully comprehend the concern raised by the reviewer. Although a noticeable neuronal loss was observed in the hippocampal DG area of FAM92A1-depleted mice (Fig. 2g, h), the alteration of regional gray matter volume (rGMV) within the hippocampus was relatively subtle across the three groups. Several factors may correlate with this difference: 1) we utilized 7T MRI scanner to acquire T2-weighted MRI data. In comparison to MRI instruments with higher magnetic field strengths, such as 9.4T MRI, the magnetic field strength in 7T MRI is relatively lower, resulting in data acquisition with lower resolution and signal quality. This lower resolution and signal quality may limit the ability to effectively capture potential differences; 2) All images were acquired from anesthetized mice without using any contrast medium to enhance MRI image contrast. This might further lead to diminished differences among the groups; 3) The morphometric difference we observed currently in the hippocampus is characterized by the presence of vacuoles and the resulting reduced thickness of the dentate gyrus (DG). Although the subsequent decrease in neuronal complexity and spine density was also detected in this region (Fig. 4), MRI-based morphology analysis may be insufficient to capture these types of neuronal degeneration-induced changes in gray matter.

To address these issues, we had indeed planned to acquire new MRI scans with enhanced contrast by utilizing fixed mouse brains perfused with a contrast medium. However, due to the super time-consuming nature for scanning the entire brain (around 4 h for one mouse brain) and the limited available homozygous mice suitable for perfusion, we could only scan three mice from each wild-type and FAM92A1^{+/-} mouse brain for analysis. Despite the small number of mice used, FAM92A1 heterozygotes exhibited a noticeable decrease in volume outside of the lateral ventricles (LVs), accompanied by a reduced gray matter density in the diminished cortical area (**Supplementary Fig. 2e**). These data suggest that the modified methods may enhance sensitivity to detect subtle differences in hippocampal subregions. We will to conduct further experiments when a sufficient number of FAM92A1 homozygotes become available in the future.

Although the reduction in gray matter density did not reach statistical significance, the difference in the changed ratio of gray matter density was aggravated with the loss of FAM92A1, primarily manifesting as decreased GMD in

hippocampal subregions (**Fig. 2i**). In conjunction with neuronal degeneration and loss in this region, these data may reliably reflect changes in gray matter density following the loss of FAM92A1.

We have supplemented test details and new data in our revised manuscript. Changes in corresponding *Materials* (Lines 997-1001 and 1029-1041), *Results* descriptions (Lines 203-226), and *Figure legends* (Supplementary file, Lines 62-69) are marked in red.

(vi) Fig. 3G, H: n is lacking.

Response: Thanks for the reviewer’s kind suggestion. For previous Fig. 3g (currently Fig.3o, p), change in FAM92A1 protein level was visualized and measured in different areas from two biological wild-type and APP/PS1 mice brains (**Fig. 3p**). For previous Fig. 3h (currently Fig.3q, r), changes in FAM92A1 with the increase of age were detected by western blots from four biological mice for each group. The quantified result showed a decreased FAM92A1 level in old mice (**Fig. 3r**).

For **fig.3p**, n = 60–62 measurements from two biologically independent experiments; unpaired two-tailed Student’s *t* test. For **fig.3r**, Data represent mean ± SEM of four biologically independent experiments; unpaired two-tailed Student’s *t* test

We have supplemented quantified data in our revised manuscript. Changes in corresponding *Results* descriptions (Lines 310-311), and *Figure legends* (Lines 2081-2087) are marked in red.

(vii) Fig. 6A: n is lacking

Response: We appreciate the reviewer's comments. The observation of changes in mitochondrial morphology was obtained from four mice in each group. Additionally, the impact of FAM92A1 depletion on mitochondrial morphology, including mitochondrial diameter, perimeter, and the number of cristae per mitochondrial perimeter, has been assessed using electron micrographs as we previously reported. In line with our earlier findings, FAM92A1 depletion results in swollen mitochondria with reduced or even absent cristae (**Fig. 6b**). (Data present mean \pm SEM; n = 168–177 mitochondria per group; one-way ANOVA).

We have supplemented these new data in our revised manuscript. Changes in corresponding **Results** (Lines 438-444) descriptions and **Figure legends** (Lines 2134-2135) are marked in red.

(viii) Fig. 6G: Please add statistical details.

Response: We appreciate the reviewer's critical comments. We apologize for the oversight in omitting the statistical information in the figure legend. Specifically, the Kruskal–Wallis H test for independent samples was employed for the statistical analysis (**Response Fig. 4**). We appreciate the reviewer's comments. Following the recommended ANCOVA analysis, the required sample size for analyzing changes in g-ratio was determined using G*Power 3.1. The detailed explanation and results of the required sample size are shown in the next response. In comparison to the previous version, we have additionally included 15 axons for each group. Compared to the control, the loss of FAM92A1 resulted in profound changes in the g-ratio (**Fig. 6h**).

We have updated these data in our revised manuscript. Changes in corresponding **Results** (Lines 461-463) descriptions and **Figure legends** (Lines 2143- 2145) are marked in red.

(ix) Fig. 6H: The use of a standard ANOVA is questionable, as g-ratio depends strongly on axon diameter (Fig. 6G). ANCOVA or similar may be more appropriate. What is n here (mice/slices/axons)?

Response: We appreciate the reviewer's insightful comments and agree with the reviewer's advice. We applied ANCOVA (Analysis of Covariance) to compare differences among three groups while controlling for covariates. Before conducting the ANCOVA analysis, the required sample size was determined using G*Power 3.1, which indicated that a total of 190 samples are needed for the three groups (**Response Fig. 5a**). Therefore, each group is represented by n = 65 axons from 3–5 mice per group (**Fig. 6i**). ANCOVA followed by the Bonferroni post hoc test (**Response Fig. 5b**) revealed a significantly decreased g-ratio in the FAM92A1 heterozygous mice, indicating thicker myelin sheath (**Fig. 6i**). A decreased g-ratio implies a decrease in axon caliber or an increase in the amount of myelin surrounding each axon. To further understand the causes of decreased g-ratio, we further analyzed the thickness of myeline sheath using one-way ANOVA. Compared to the myelin sheath in wild-type group, the myelin thickness was increased in FAM92A1 heterozygous mice (**Fig. 6i**). These data indicating the aberrant membrane of the myelin sheath in FAM92A1 heterozygous mice. Compared to the heterozygous mice, the abnormalities of myelin sheaths in FAM92A1 homozygous mice were more evident in a reduced axon diameter (Fig. 6h, inset). Combined with the sequential changes in the axonal morphology during the process of degeneration, from swelling to eventual breaking at the thinned part (*Weil et al. 2016. PMID: 27346352*), these findings suggest that neuronal damage may be more severe in the FAM92A1 knockout mice.

We have supplemented these new data in our revised manuscript. Changes in corresponding **Results** (Lines 463-473) descriptions and **Figure legends** (Lines 2143- 2146) are marked in red.

(x) Fig. 6N: I could not find any statistical details, i.e., the conclusion is not adequately supported by data.

Response: Thanks for the reviewer's kind suggestion. Changes in the number and area of synaptic vesicles have been quantified using the FIB-SEM electron micrographs. All synapses displaying distinguishable pre- and post-synaptic areas were measured on a micrograph with dimensions of 2209 × 205 pixels. Quantification was conducted on every 50th slice, with a total of 18 slices used for analysis (**Fig. 6n**). Quantification results revealed that FAM92A1 depletion caused an increased ratio of SVs with a diameter larger than 80 nm. The average area of SVs was found to be increased

in the FAM92A1 depleted synapses (Fig. 6o-q).

We have supplemented these new data in our revised manuscript. Changes in corresponding *Materials* (Lines 1157-1162), *Results* (Lines 490-498) descriptions, and *Figure legends* (Lines 2153-2155) are marked in red.

- The relevance and mechanism of brain “swelling” (Fig. 1) remains unclear, as neither changes in CSF nor volume changes of white or grey matter can explain it. This discrepancy should be resolved.

Response: We appreciate the reviewer’s comments. We only visualized the MRI images without conducting any quantification. Due to the challenge of visualizing subtle changes in the ventricle area without quantification, we incorrectly reported no alterations in ventricle cerebrospinal fluid (CSF). To precisely investigate the cause of brain swelling, we measured the ventricle areas, comparing both the total and individual ventricle area per brain area. The results revealed that the total ventricle areas were enlarged in the FAM92A1-depleted mice. Specifically, the enlargement in the ventricle area was primarily due to the expansion of lateral ventricles (LVs) (Fig. 2a, b and Supplementary Fig. 2c, d). These quantified results suggest that the enlarged LVs may contribute to brain swelling in the caudal area of the FAM92A1-deficient mouse brain.

In addition to its role in mitochondrial inner membrane remodeling, FAM92A1 is also involved in ciliogenesis (*Li et al. 2016. PMID: 27528616*). Cilia on ependymal cells lining the brain’s ventricles are essential for directing the flow of CSF through the ventricular system. Therefore, the loss of FAM92A1 could potentially impair ependymal cell cilia, disrupting CSF flow dynamics and resulting in the accumulation of CSF within the ventricles. Notably, if the brain swelling in the FAM92A1-deficient mouse brain only correlates with the enlarged LVs, ventricular dilatation was expected to squeeze its surrounding brain area, resulting in reduced volumetric morphology and consequently increased gray matter density. Although a deformation-based morphometry (DBM) analysis of the volumetric morphology of the mouse brain indeed revealed a decreased volume outside of the LVs (Supplementary Fig. 2e), the voxel-based morphometry (VBM) analysis of gray matter identified reduced gray matter density (Fig. 2c-d and Supplementary Fig. 2e). These findings suggest that the brain swelling observed in FAM92A1-deficient mice may

result from a combination of factors, including enlarged ventricular dilatation, neuronal degeneration, and volumetric differences in the brain caused by neuronal loss.

We have supplemented these new data in our revised manuscript. Changes in corresponding **Results** (Lines 198-224) descriptions and **Figure legends** (Lines 2043-2044 and Supplementary file, lines 57-69) are marked in red.

- Fig. 1L, right: 60 mm seems to be incorrect.

Response: Thanks for Reviewer's the careful check. The "decimal point" of the original value was lost for unknown reasons. We have corrected the ticks of the Y-axis of **Fig. 1I** in the revised manuscript.

- Fig. 2C: The unit should be added.

Response: Thanks for reviewer's the careful check. The previously indicated average regional gray matter volume (rGMV) should be corrected. It is regional gray matter density (GMD). rGMV means the total volume of gray matter present in a particular brain region, while GMD represents the concentration of gray matter in this area (*Takeuchi H et al. 2011. PMID: 20740644*). The Y-axes in our manuscript represent the density of gray matter (GMD), which reflects the changes in the concentration of gray matter. Since gray matter density is a dimensionless measure, and it does not have a unit. We have revised the scale label of **Fig. 2d** and the title of Y-axes of **Fig. 2e** (previous Fig.2c) to the density of gray matter in the revised manuscript.

- Fig. 2G-H: The reported reduction is not evident from the sample images. Also, how does this relate to the volume seen with MRI?

Response: We appreciate the reviewer's comments. Compared to the FAM92A1^{+/+} mice, cresyl violet (CV) staining revealed vacuolation in the hippocampus of the FAM92A1-deficient mouse brain, particularly in the subgranular zone (SGZ). Therefore, the decreased thickness in the DG area is primarily attributed to the presence of vacuoles. To enhance the visualization of differences among the three groups, the rotated CA1, CA3, and DG areas are horizontally presented in the revised manuscript (**Fig. 2g**).

Through quantifying the ventricle area using MRI images, we revealed that the enlargement of lateral ventricles (LVs) area might contribute to the enlargement of brain gross morphology in the FAM92A1 knockout mice (**Fig. 2a, b and Supplementary Fig. 2a-d**). In addition to our reported role of FAM92A1 in maintaining mitochondrial inner membrane architecture, FAM92A1 also plays a role in ciliogenesis (*Li et al. 2016. PMID: 27528616*). Therefore, the loss of FAM92A1 might also result in defects in ependymal cilia, leading to reduced dynamics of cerebrospinal fluid and consequently contributing to the development of hydrocephalus (*Jiang et al. 2020. PMID: 32229724*). If the brain swelling in the FAM92A1 deficient mouse brain only correlates with the enlarged LVs, ventricular dilatation was expected to squeeze its surrounding brain area, resulting in reduced volumetric morphology and consequently increased gray matter density. In fact, a deformation-based morphometry (DBM) analysis of the volumetric morphology of the mouse brain revealed a decreased volume outside of the LVs (**Supplementary Fig. 2e**), while voxel-based morphometry (VBM) analysis of gray matter identified reduced gray matter density, rather than the expected increase in GMD (**Fig. 2c-d and Supplementary Fig. 2e**). These findings suggest that, beside ventricular dilatation, neuronal degeneration and loss caused morphometric difference may also contribute to brain swelling in FAM92A1-depleted mice.

In contrast to the two areas with noticeable reductions in gray matter density, the gray matter density in the hippocampal region seemed to be less affected, despite the presence of numerous vacuoles indicating potential neuronal damage or loss in this area (**Fig. 2g-i**). We guess it might be attributed to the following reasons: 1) due to the lower magnetic field strength of the MRI instrument and the absence of contrast medium to enhance contrast during data acquisition, the used MRI data may not be sufficiently sensitive to detect all abnormalities; 2) The morphometric difference we observed currently in the hippocampus is characterized by the presence of vacuoles and the resulting reduced thickness of the dentate gyrus (DG). Although the subsequent decrease in neuronal complexity and spine

density was also detected in this region (Fig. 4), MRI-based morphology analysis may be insufficient to capture these types of neuronal degeneration-induced changes in gray matter. Considering the intricate intrinsic connectivity between the entorhinal cortex and the hippocampus across species, coupled with the evident neuronal loss observed in CV staining, we thus focus on the hippocampal region to explore the role of FAM92A1 in the CNS. We have discussed it our revised manuscript (Lines 750-785).

- l. 215 “reduced regional gray matter volume, which may be attributed to neuronal degeneration”: This conclusion is not convincing, as no reduction of gray matter volume (MRI) was seen in DG, where Fluoro-Jade staining was performed. It remains unclear to me why these histological analyses have focused on DG rather than those areas in which gray matter volume changes were found by MRI.

Response: We appreciate the reviewer’s comments. In the current study, the voxel-based morphometry (VBM) analysis of T2-weighted MRI data revealed a significant decrease in gray matter volume in two specific areas, alongside a slight reduction in the hippocampal region. All observed alterations, such as CV, Fluoro-Jade B, Fluoro-Jade C staining, Golgi staining, synapses ultrastructure, and so on, were concentrated in the hippocampal area but not the area with evident reduced gray matter volume. This shift primarily attributed to the following reasons: 1) FAM92A1 exhibits relatively higher expression levels in the hippocampus, as indicated by in situ hybridization (ISH) data sourced from the Allen Brain Atlas database (Fig. 1a); 2) our preliminary behavioral tests revealed that FAM92A1 knockout mice exhibited memory deficits, particularly poor performance in spatial memory tasks (Fig. 3a-n and Supplementary Fig. 4). 3) the profound decrease in gray matter density (GMD) in the entorhinal area.

In both human and rodent brains, grid cells in the medial entorhinal cortex and place cells in the hippocampus form a well-known entorhinal–hippocampal neural circuit, which is critical for spatial navigation and memory (Donato et al. 2017. PMID: 28154241; Aronov et al. 2017. PMID: 28358077). Consequently, FAM92A1 deficient mice also exhibited decreased memory in spatial-involved contextual memory tasks but not tone-involved cued memory tasks (Fig. 3l-n). Considering the results of the significant decrease in gray matter density (GMD) within the entorhinal cortex and the well-established intrinsic connectivity between the entorhinal cortex and the hippocampus across various species, we additionally conduct a VBM analysis focusing on the hippocampal region. The results showed that the impact on gray matter density was exacerbated along with the loss of FAM92A1, primarily manifesting as decreased GMD in hippocampal subregions (Fig. 2i). In conjunction with neuronal degeneration and loss in this region, these data may reliably reflect changes in gray matter density following the loss of FAM92A1. Therefore, we focused on the hippocampal region to investigate the function of FAM92A1.

Moreover, due to the lower magnetic field strength of the MRI instrument and the absence of a contrast medium to enhance contrast during data acquisition, the current MRI data may not be sufficiently sensitive to detect all abnormalities. We fully acknowledge the reviewer's concerns. We will scan new MRI images with increased contrast by using the fixed mouse brain in the future, which is perfused with the contrast medium.

We have revised the description of result and discussed in the revised manuscript. Changes in corresponding **Results** (Lines 195-247) and **Discussion** (Lines 767-785).

- Fig. 3G-J: These analyses are of peripheral interest to the present ms.

Response: We appreciate the reviewer's comments. We agree with the reviewer's perspective. These data look peripheral to the main focus of this manuscript. We hope to report these data primarily due to the following reasons. The current knowledge of FAM92A1 in the CNS system is quite limited. There is one clinical case report associating *FAM92A1* microdeletion in the human genome with brain abnormalities and autism disorder. Yet, experimental validation of these findings remains elusive. Despite multiple behavioral tests in this study revealing memory defects in FAM92A1-depleted mice, it remains inconclusive to definitively link FAM92A1 defects with potential neurological diseases. To comprehensively understand the roles of FAM92A1 in the CNS, we thus explored the relationship of FAM92A1 level with both aging and AD disease, and the potential diseases following the absence of FAM92A1 by clustering the differentially expressed genes. All of these data imply the association of FAM92A1 defects with multiple neurological disorders, and loss of FAM92A1 heightens the susceptibility to neurological disorders. We hope to incorporate these data into the manuscript to emphasize the potential connection between FAM92A1 and the molecular pathology of neurological diseases.

- Fig. 4: It remains unclear which neurons were traced in Sholl analyses (l. 278: "hippocampal neurons"). Please add all relevant details to Methods. Again, the brain region analyzed is obviously not motivated by the MRI findings in Figs. 1 and 2. Please explain.

Response: We appreciate the reviewer's for pointing this out. We apologize for missing this detailed method in the previous manuscript. Due to the challenge of identifying individual neurons or dendrites amidst the dense population of neurons, we primarily analyzed granule neurons in the hippocampal dentate gyrus (DG) region. The details methods and criteria of neuron for sholl analysis have been added into the *Materials* (Lines 1200-1212) as follows:

Hippocampal granule neurons in the DG region included in the analysis satisfied the following criteria: a) the presence of soma within the hippocampal DG region; b) morphologically, by the presence of multiple short dendrites extending outward in a radial pattern as characteristic of hippocampal granule neurons; c) an isolated cell body with a clear relationship of the primary dendrite to the soma; d) complete impregnation of the neuron along the entire length of the dendritic arbor without truncated dendrites; and e) minimal overlap or relative non-overlap with surrounding impregnated cells.

The reason we focus on the hippocampus to investigate the FAM92A1 roles has been discussed in the revised manuscript as follows: 1) FAM92A1 exhibits relatively higher expression levels in the hippocampus, as indicated by in situ hybridization (ISH) data sourced from the Allen Brain Atlas database (**Fig. 1a**); 2) our preliminary behavioral tests revealed that FAM92A1 knockout mice exhibited memory deficits, particularly poor performance in spatial memory tasks (**Fig. 3a-n and Supplementary Fig. 4**). 3) the profound decrease in regional gray matter density (GMD) in the entorhinal area.

In both human and rodent brains, grid cells in the medial entorhinal cortex and place cells in the hippocampus form a well-known entorhinal–hippocampal neural circuit, which is critical for spatial navigation and memory

(Donato et al. 2017. PMID: 28154241; Aronov et al. 2017. PMID: 28358077). Consequently, FAM92A1 deficient mice also exhibited decreased memory in spatial-involved contextual memory tasks but not tone-involved cued memory tasks (Fig. 3I-n). Considering the results of the significant decrease in gray matter density (GMD) within the entorhinal cortex and the well-established intrinsic connectivity between the entorhinal cortex and the hippocampus across various species, we additionally conduct a VBM analysis focusing on the hippocampal region. The results showed that the impact on gray matter density was exacerbated along with the loss of FAM92A1, primarily manifesting as decreased GMD in hippocampal subregions (Fig. 2i). In conjunction with neuronal degeneration and loss in this region, these data may reliably reflect changes in gray matter density following the loss of FAM92A1. Therefore, we focused on the hippocampal region to investigate the function of FAM92A1. In the discussion section, we elucidated the underlying cause of brain swelling and explained the reason for a particular focus on the hippocampus.

- Fig. 5A-C: Methods related to Ca²⁺ imaging are completely lacking. It is unclear to me how the experiments were conducted and what exactly was measured/quantified. For example, the authors mention spontaneous activity, but Fig. 5B shows trials. Please add all relevant details to Methods.

Response: We really appreciate for pointing this out. The detailed methods for optical fiber-based Ca²⁺ recordings and photometry data analysis have been included in the *Materials* (Lines 1330-1371) section of the revised manuscript.

- Fig. 5D-E and l. 330 (“impact of FAM92A1 on excitatory synaptic transmission is more pronounced at individual synaptic sites”): The data provided do not support such a conclusion! Also note that mEPSC and sEPSC amplitudes are basically the same.

Response: We really appreciate the reviewer’s critical comments. The original conclusion as well as the description of our results (Lines 379-389) has been corrected as follows:

The absence of FAM92A1 resulted in a reduction in both sEPSC and mEPSC frequencies, indicating the impaired excitatory synaptic inputs following loss of FAM92A1. Additionally, the amplitude of mEPSC was diminished upon FAM92A1 depletion, whereas the amplitude of sEPSC remained less affected (Fig. 5d, e). These data suggest that FAM92A1 knockout might affect the efficacy of neurotransmitter release during miniature synaptic events while having a lesser effect on the overall strength of synaptic transmission onto the postsynaptic neuron. The reduction in the frequency of sEPSC indicates that FAM92A1 depletion dampens overall synaptic activity (Fig. 5d), further supported by the decreased probability of glutamate release sites per neuron and the weakened strength of each site (as inferred from the frequency and amplitude of mEPSCs) (Fig. 5e). Overall, these results suggest that FAM92A1 knockout results in the dysfunction of excitatory synaptic transmission.

For the amplitude of mEPSCs and sEPSCs, we have carefully re-checked our raw data. The mean values of mEPSC and sEPSC amplitudes of control neurons are slightly different (sEPSC: 14.06 and mEPSC: 15.44). As the reviewer mentioned, the amplitude of mEPSCs and sEPSCs is typically similar under most conditions, but this may not always be the case. Sometimes, the amplitude of mEPSCs and sEPSCs can vary (Barbara et al. 2022. PMID: 36137051; Mao

et al. 2015. PMID: 25816842; Sebastian et al. 2014. PMID: 25406064; Acharjee et al. 2018. PMID: 30185466; Cai et al. 2002. PMID: 12149253), as observed in our data. We speculate that the difference in the current data may be attributed to the presence of numerous small-amplitude responses during the spontaneous EPSCs test. In fact, we had planned to repeat this experiment. Unfortunately, due to mortality and the very low frequency of FAM92A1^{-/-} offspring, we really do not have available FAM92A1 knockout mice to use for this experiment at present.

- l. 333 “further supported by the decreased number of glutamate release sites per neuron”: A reduction in mEPSC frequency does not necessarily imply a lower number of release sites, as it is also strongly dependent on release probability. Please correct.

Response: We appreciate the reviewer for pointing this out. We agree with the reviewer’s comments. The amplitudes of mEPSC reflect the number and conductance of subsynaptic AMPARs, whereas the frequency of mEPSC reflects the probability of vesicle release (p) and/or the number of synapses. Hence, "probability" is the correct description. We have revised the relevant description in the revised manuscript

- Fig. 5E: Why are the lower traces interrupted? What does the green line in the left trace indicate?

Response: We really appreciate the reviewer for this observation. It's possible that the line was inadvertently shifted when creating the images. Specifically, the green line present in the left panel is the missing line in the right panel. We have revised it in the revised Fig. 5e.

- MEA recordings: The results are very difficult to read, mainly because the definitions of the reported quantities are not properly introduced. For example, “synchrony index” is not defined. Please add all analytical details to Methods and rephrase Results for clarity. What do the gray bars in Fig. 5J indicate?

Response: Thanks to the reviewer for pointing this out. As suggested, we have rephrased the MEA results (Lines 391-430), and the criteria of the parameters used, along with their definitions, have been added to the section on MEA recordings and data analysis in the *Materials* (Lines 1309-1317). Synchrony represented a measure of similarity between two spike trains and the synchrony index was estimated by the area under the normalized cross-correlogram for a time window of 20 ms. All gray bars on the raster plots in previous Fig.5j (currently Fig. 5l) represent the detected network bursts.

Data analysis was performed off-line by using the manufacturer’s standalone tool Neural Metric Tool (Axion Biosystems). A spike detection was computed with an adaptive threshold of six standard deviations of the estimated noise for each electrode. The electrodes that detected at least 5 spikes per minute were classified as active electrodes. Bursts were identified using an inter-spike interval (ISI) threshold requiring a minimum number of five spikes with a maximum ISI of 100 ms. Network bursts were identified as bursts of more than 50 spikes that occurred in more than 35% of the active electrodes in the well, with a maximum ISI of 100 ms. Synchrony represented a measure of similarity between two spike trains and the synchrony index was estimated by the area under the normalized cross-correlogram for a time window of 20 ms.

- l. 424-425 “area of synaptic vesicles (SVs) ... above 5000 nm³”: Did the authors analyze areas or volumes?

Response: Thanks to the reviewer for pointing this out. This unit was incorrectly described in the results. It should be the area (mm²) but not the volume (mm³) as shown in **Supplementary Fig. 5f** (previous Fig. 6l). We have corrected this mistake in the revised manuscript.

- l. 89 “structural characteristics have not been elucidated yet”: Why is Breugel et al. 2022 (PMID: 35383272IF: 5.9 Q1) not cited? - The authors repeatedly use words such “valuable” (l. 42) and “compelling evidence” (l. 254, l. 736) when referring to their own data. This is inadequate and should be corrected.

Response: We thank the reviewer for highlighting the study on computationally predicted structure of FAM92A1. We have now mentioned and cited this work. Furthermore, based on the reviewer's suggestion, we have tempered down the statement regarding our results and conclusion.

- l. 53: CIBAR1?

Response: Thanks to the reviewer for pointing this out. The alias of FAM92A1 is CIBAR1. We have fixed this mistake in the revised manuscript.

- Refs. 18 and 20 are the same.

Response: Thanks for this careful check. All references have been checked and the repeated references have been deleted in the revised manuscript.

In summary, we sincerely hope that the reviewer finds the addition of new data, revisions, and our responses to the critiques are satisfactory and support the publication of our study.

REVIEWER COMMENTS

Reviewer #1 (Remarks to the Author):

The authors added the experiments that I suggested. This experiment allows to increase the sensitivity and resolution of MRI data and detect subtle changes. The MRI part is fully satisfactory for publication.

Reviewer #2 (Remarks to the Author):

I greatly appreciate the efforts of the authors for extensively revising their manuscript in response to the referees' comments. The revisions have improved their comprehensive study spanning physiological, cell biological, structural and computational analysis to explore the role of FAM92A1.

From my side, there is one major problem left, which is the use of the Sigma antibody HPA034760 for immunofluorescence staining. As the authors acknowledge in their response letter, this antibody is not fully specific for FAM92A1 in Western Blot analyses and shows a background signal even in FAM92A1-deficient tissue (I assume the second and fourth row in Response Fig. 1a represent two different stainings of ko slices?). This sheds serious doubts on the suspiciously broad reported cellular localization of FAM92A1 on CCPs, caveolae and mitochondria. It is not an option to omit such important control experiments from the manuscript. Together with the data for the ProteinTech antibody, the same western blots and cellular localization experiments as shown in Response Fig. S1a, S1b and c must be conducted and reported as SI for the Sigma Antibody using wt and ko cells. If there is a significant background staining for the Sigma antibody in FAM92A1 ko cells, in particular at cellular membranes, I see three possibilities to address this issue (in the order of preference):

- 1) The authors further seek to affinity purify the Sigma antibody and improve the specificity. There should be no membrane structure labelled in the FAM92A1 ko cells with this antibody since the membrane-binding BAR domain has been deleted in the ko cells.
- 2) The authors omit all immunohistochemistry/-fluorescence data from the manuscript using this antibody (Fig. 1, Fig. 7).
- 3) The authors carefully analyze the background signal of the Sigma antibody in the FAM92A1-deficient cells and try to dissect specific from non-specific signal.

Minor

Line 623: In comparison to the dimerized wild-type of the FAM92A1 BAR domain (abbreviated as WT), the mutant primarily existed as a monomer in equilibrium (Supplementary Fig. 7f): Sorry, I missed this in the last review. The wt control must be included in the gelfiltration analyses, which is otherwise meaningless. Is the wt really a dimer in gelfiltration? Most other BAR domains are monomers at concentrations used in gelfiltration analyses.

Revised text

Line 591: Analysis of the α -helix structure revealed that it originated from residue 2 (partially within the helix) and extended to residue 211 (also partially within the helix). Thus, the region spanning residues 2–211 was designated as the FAM92A1 BAR domain (Supplementary Fig. 1e). More simple: Based on secondary structure analysis, the region spanning residues 2–211 was designated as the FAM92A1 BAR domain (Supplementary Fig. 1e). (the exact description of secondary structure elements follows anyhow afterwards).

Line 594: The BAR domain structure of FAM92A1 reveals a characteristic "banana-shaped" homo-dimeric state, with a central dimeric six-helical bundle and peripheral arms formed by the three extended alpha-helices from each monomer (Fig. 8a).

Better: In the asymmetric unit of the crystals, two FAM92A1 BAR domain monomers assembled via a two-fold symmetric interface, yielding a banana-shaped homo-dimer characteristic for BAR-domain proteins. (and then the description).

Supplemental Fig. 8a, oligomerization of the FAM92A1 BAR domain

I had a quick look into the determined pdb file 8ceg available in the pdb database. Next to the reported tip-to-tip dimer, there is another BAR domain dimer in a parallel orientation in the crystal

lattice which could reveal another oligomerization site. This could also be reported in the SI as potential oligomerization interface. Since none of the oligomerization interfaces were experimentally validated, one should, however, be cautious with the interpretation:
Instead line 792: Taken together, the BAR domain of FAM92A1 has the capability to form dimers and oligomers
Better: Taken together, the BAR domain of FAM92A1 has the capability to form dimers and, possibly, oligomers ...

Reviewer #3 (Remarks to the Author):

The authors have made a notable effort to address my previous criticisms, resulting in significant improvements to the manuscript. The revised manuscript provides a comprehensive evaluation of the function of FAM92A1 in neuronal structure and function. However, the manuscript remains very difficult to read, posing a tour de force for the reader. In particular, the Results section could substantially benefit from thorough editing to consolidate critical numerical/statistical information into the main text, rather than dispersing it across Results, figure legends, supplementary figures, and additional supplementary files. I also recommend a comprehensive language edit to enhance clarity and readability, making the content more accessible to readers.

Point-by-Point Response to Reviewers' Comments

Reviewer #1 (Remarks to the Author):

The authors added the experiments that I suggested. This experiment allows to increase the sensitivity and resolution of MRI data and detect subtle changes. The MRI part is fully satisfactory for publication.

Response: We thank the reviewer for the valuable comments to improve the quality of the manuscript.

Reviewer #2 (Remarks to the Author):

I greatly appreciate the efforts of the authors for extensively revising their manuscript in response to the referees' comments. The revisions have improved their comprehensive study spanning physiological, cell biological, structural and computational analysis to explore the role of on FAM92A1.

Response: We thank the reviewer for the valuable comments to improve the quality of the manuscript.

From my side, there is one major problem left, which is the use of the Sigma antibody HPA034760 for immunofluorescence staining. As the authors acknowledge in their response letter, this antibody is not fully specific for FAM92A1 in Western Blot analyses and shows a background signal even in FAM92A1-deficient tissue (I assume the second and fourth row in Response Fig. 1a represent two different stainings of ko slices?). This sheds serious doubts on the suspiciously broad reported cellular localization of FAM92A1 on CCPs, caveolae and mitochondria.

It is not an option to omit such important control experiments from the manuscript. Together with the data for the ProteinTech antibody, the same western blots and cellular localization experiments as shown in Response Fig. S1a, S1b and c must be conducted and reported as SI for the Sigma Antibody using wt and ko cells. If there is a significant background staining for the Sigma antibody in FAM92A1 ko cells, in particular at cellular membranes, I see three possibilities to address this issue (in the order of preference):

- 1) The authors further seek to affinity purify the Sigma antibody and improve the specificity. There should be no membrane structure labelled in the FAM92A1 ko cells with this antibody since the membrane-binding BAR domain has been deleted in the ko cells.
- 2) The authors omit all immunohistochemistry/-fluorescence data from the manuscript using this antibody (Fig. 1, Fig. 7).
- 3) The authors carefully analyze the background signal of the Sigma antibody in the FAM92A1-deficient cells and try to dissect specific from non-specific signal.

Response: We appreciate the reviewer's valuable and critical comments. We have assessed the knockout efficiency of FAM92A1 in mouse brain slices and primary fibroblast cells using anti-FAM92A1 antibodies. The Sigma antibody works well in the immunostaining experiments. The fluorescent signal of FAM92A1, detected by the Sigma antibody (HPA034760), was nearly completely depleted in the homozygous FAM92A1 knockout cells and slices, with only a minor, weak non-specific background signal (**Supplementary Fig. 1g, h**, left panel). In contrast, the FAM92A1 antibody from Proteintech (24803-1-AP), which recognized multiple non-specific bands (**Supplementary Fig. 1i**) in addition to the FAM92A1 protein band, displayed less pronounced reduction in the fluorescent signal of FAM92A1 compared to the wild-type cells and brain slices (**Supplementary Fig. 1g, h**, right panel). These

immunofluorescent data indicate that the localization of FAM92A1 indicated using the Sigma antibody are reliable. Thanks to the reviewer's comments, we have added these data in the Supplementary Fig. 1.

We have supplemented these new data in our revised manuscript. Changes in corresponding **Results** (Lines 154-158) descriptions and **Figure legends** (Supplementary file, Lines 48-52) are marked in red.

Minor

Line 623: In comparison to the dimerized wild-type of the FAM92A1 BAR domain (abbreviated as WT), the mutant primarily existed as a monomer in equilibrium (Supplementary Fig. 7f):

Sorry, I missed this in the last review. The wt control must be included in the gelfiltration analyses, which is otherwise meaningless. Is the wt really a dimer in gelfiltration? Most other BAR domains are monomers at concentrations used in gelfiltration analyses.

Response: We appreciate the reviewer's valuable comments. The equilibrium state of purified wild-type FAM92A1 BAR domain protein, as determined using SEC-MALS, was previously reported in our published article (Figure 4B) (Wang *et al.* 2019, PMID: 30404948). The results showed that the purified wild-type FAM92A1 BAR domain exists primarily as dimers, with a minor quantity present as oligomers. In the current manuscript, we assessed the equilibrium state of purified FAM92A1 BAR dimer interface mutant protein under the same conditions and using the same instruments. Compared to the wild-type protein, which exists mainly as dimer in equilibrium, the purified mutant protein predominantly existed as a monomer, with minor aggregates. Since only one sample can be analyzed per experiment in SEC-MALS, the wild-type and mutant proteins cannot be compared simultaneously. The equilibrium state of the wild-type FAM92A1 BAR domain protein is consistent with our previously reported data. To avoid

repeating published data in the current manuscript, we only present the equilibrium state of the FAM92A1 BAR mutant protein in our previous revision. Thanks to the reviewer's suggestion, we have added the equilibrium state of the wild-type FAM92A1 BAR protein in the revised **Supplementary Fig. 7f**. In contrast to the monomeric state of mutant proteins, the wild-type FAM92A1 BAR domain proteins primarily existed as a dimeric state.

Size-exclusion chromatography (SEC), is a highly effective method utilized for the separation of molecules based on their size. When coupled with multi-angle light scattering (MALS), it evolves into a robust technique for evaluating the molecular weight and structural characteristics of proteins in their native state. In contrast, analysis using SDS-PAGE after gel filtration reveals the monomeric state of BAR domain proteins due to denaturation of proteins in SDS treatment. By using SEC-MALS, several BAR domain proteins, such as the PX-BAR protein Mvp1 (*Sun et al. 2020, PMID: 32198400*), SNX-BAR domain protein FCHSD2 (*Almeida-Souza et al. 2018, PMID: 29887380*), F-BAR domain protein SRGAP2 (*Sporny et al. 2017, PMID: 28333212*), and BAR domain protein EndophilinA (*Bademosi et al. 2023, PMID: 36827984*), are detected as dimers and even tetramers. Taken together, these studies indicate that the purified BAR domain proteins can indeed exist as dimers in their native state.

We have supplemented these new data in our revised manuscript. Changes in corresponding **Materials and Methods** (Lines 1486-1489), **Results** (Lines 625-629) descriptions, and **Figure legends** (Supplementary file, Lines 154-156) are marked in red.

Revised text

Line 591: Analysis of the α -helix structure revealed that it originated from residue 2 (partially within the helix) and extended to residue 211 (also partially within the helix). Thus, the region spanning residues 2–211 was designated as the FAM92A1 BAR domain (Supplementary Fig. 1e).

More simple: Based on secondary structure analysis, the region spanning residues 2–211 was designated as the FAM92A1 BAR domain (Supplementary Fig. 1e). (the exact description of secondary structure elements follows anyhow afterwards).

Response: We appreciate the reviewer's constructive comments. As suggested, the description of the FAM92A1 BAR domain has been revised in the manuscript as follows (Lines 595-596):

“Based on the crystal structure, the region spanning residues 2–211 was coiled-coil composed of three α helices, designated as the FAM92A1 BAR domain.”

Line 594: The BAR domain structure of FAM92A1 reveals a characteristic "banana-shaped" homo-dimeric state, with a central dimeric six-helical bundle and peripheral arms formed by the three extended alpha-helices from each monomer (Fig. 8a).

Better: In the asymmetric unit of the crystals, two FAM92A1 BAR domain monomers assembled via a two-fold symmetric interface, yielding a banana-shaped homo-dimer characteristic for BAR-domain proteins. (and then the description).

Response: We appreciate the reviewer's constructive comments. The description of the crystal structure of the FAM92A1 BAR dimer has been revised in the manuscript according to the reviewer's suggestions (Lines 596-599).

Supplemental Fig. 8a, oligomerization of the FMA92A1 BAR domain

I had a quick look into the determined pdb file 8ceg available in the pdb database. Next to the reported tip-to-tip dimer, there is another BAR domain dimer in a parallel orientation in the crystal lattice which could reveal another oligomerization site. This could also be reported in the SI as potential oligomerization interface. Since none of the oligomerization interfaces were experimentally validated, one should, however, be cautious with the interpretation:

Instead line 792: Taken together, the BAR domain of FAM92A1 has the capability to form dimers and oligomers

Better: Taken together, the BAR domain of FAM92A1 has the capability to form dimers and, possibly, oligomers ...

Response: We appreciate the reviewer's valuable and critical comments. As suggest, the other potential lateral/parallel oligomerization interface has been marked with a black arrow in **Supplementary Fig. 8a**. And the corresponding image of the potential lateral oligomerization interface has also been added to **Supplementary Fig. 8d**. It scores poorly in PDBePISA server (insignificant and with very poor free energy (ΔG) (**Response Fig. 1a, c**), unlike the tip-to-tip oligomerization interface (**Response Fig. 1b, c**). Despite its poor score, since it is the other major packing surface, we have included it in the file of Supplementary information and briefly mentioned it in the text (Lines 665-672). It is also discussed in the legend for Supplementary Fig. 8a (Supplementary file, Lines 165-172). Throughout the text, the words "potential" or "possible" have been added to refer to the oligomerization interfaces in the revised manuscript.

We thank the reviewer valuable comments. The mentioned text on line 792 in previous manuscript has also been revised as suggested to: "Taken together, the BAR domain of FAM92A1 has the capability to form dimers and possibly oligomers" (Lines 725-726).

We have supplemented these new data in our revised manuscript. Changes in corresponding **Results** (Lines 665-672) descriptions, and **Figure legends** (Supplementary file, Lines 165-172 and Lines 181-183) are marked in red.

Reviewer #3 (Remarks to the Author):

The authors have made a notable effort to address my previous criticisms, resulting in significant improvements to the manuscript. The revised manuscript provides a comprehensive evaluation of the function of FAM92A1 in neuronal structure and function. However, the manuscript remains very difficult to read, posing a tour de force for the reader. In particular, the Results section could substantially benefit from thorough editing to consolidate critical numerical/statistical information into the main text, rather than dispersing it across Results, figure legends, supplementary figures, and additional supplementary files. I also recommend a comprehensive language edit to enhance clarity and readability, making the content more accessible to readers.

Response: We thank the reviewer for the valuable comments to improve the manuscript. As suggested, the manuscript has been thoroughly revised. Additionally, some critical numerical and statistical information have been added to better describe the results. We hope the revised manuscript offers improved clarity and readability for readers.

REVIEWERS' COMMENTS

Reviewer #2 (Remarks to the Author):

Thank you very much. My concerns have been addressed in the revised manuscript and I congratulate the authors on their detailed work on a membrane-remodelling machine.

Point-by-Point Response to Reviewers' Comments

Reviewer #2 (Remarks to the Author):

Thank you very much. My concerns have been addressed in the revised manuscript and I congratulate the authors on their detailed work on a membrane-remodelling machine.

Response: We thank the reviewer for the valuable comments to improve the quality of the manuscript.